# Structure and mechanism of a mycobacterial isoniazid efflux pump *Ms*Rv1273c/72c with a degenerate nucleotide-binding site

Jing Yu[1,7], Yuhui Lan[1,7], Chen Zhu[1], Zhendong Chen[1], Junyi Pan[1], Yanfeng Shi[1,2], Lan Yang[1,2], Tianyu Hu[1], Yan Gao [1], Yao Zhao [2], Xiaobo Chen[1], Xiuna Yang [1], Shuihua Lu[2], Luke W. Guddat [3], Haitao Yang [1] ✉, Zihe Rao [1,2,4,5,6] ✉ & Jun Li [1] ✉

Heterodimeric ATP-binding cassette (ABC) transporters containing one catalytically impaired degenerate nucleotide-binding site (NBS) have a mechanism different from those with two active NBSs. However, the structural basis of their transport mechanism remains to be explained. Here, we determine mycobacterial *Ms*Rv1273c/72c to be an isoniazid efflux pump and determine several structures by cryo-electron microscopy showing specific asymmetrical features including an N-terminal extending loop and a periplasmic helical hairpin only found in *Ms*Rv1272c. In addition, we capture three distinct asymmetric states where the nucleotide-binding domains are partially dimerized at the degenerate site. Using these intermediate states, the D-WalkerB loop and X-signature loop of *Ms*Rv1272c modulate and couple the function of both NBSs through conformational changes. Thus, these data provide insights into the mechanism of this heterodimeric ABC transporter containing a degenerate NBS. The structures also provide a framework for the rational design of anti-tuberculosis drugs targeting this drug-efflux pump.

Tuberculosis (TB), caused by the pathogenic bacteria, *Mycobacterium tuberculosis* (*Mtb*), kills around 1.7 million people each year[1]. The current recommended treatment for TB involves a combination of four antibiotics: isoniazid (INH; targeting InhA[2]), rifampicin (RIF; targeting RNA polymerase[3]), pyrazinamide (PZA; target unknown) and ethambutol (EMB; targeting EmbA/B/C[4]), which were all discovered nearly 70 years ago[5]. However, these drugs are becoming less effective against *Mtb* where resistance has developed, either by site of action changes, or the emergence of metabolism modifying enzymes or efflux pumps. Hence, new targets and drugs to treat TB are urgently needed, or where drug resistance has developed a better understanding of the

molecular basis for this needs to be developed. One possible class of proteins important for drug resistance are the ATP-binding cassette (ABC) transporters and these are found in all kingdoms of life[6]. However, they are structurally and functionally diverse and have not been fully characterized. These transporters use the energy of ATP binding and hydrolysis to transport an astonishing variety of substrates including small ions, lipids, peptides, and proteins across membranes[7–10]. All ABC transporters are built from common structural modules comprising two nucleotide-binding domains (NBDs), which bind and hydrolyze ATP, and two transmembrane domains (TMDs) that facilitate substrate translocation[11,12]. The binding and hydrolysis of

[1]Shanghai Institute for Advanced Immunochemical Studies and School of Life Science and Technology, ShanghaiTech University, Shanghai 201210, China. [2]National Clinical Research Center for Infectious Disease, Shenzhen Third People's Hospital, Shenzhen 518112, China. [3]School of Chemistry and Molecular Biosciences, The University of Queensland, Brisbane, QLD 4072, Australia. [4]State Key Laboratory of Medicinal Chemical Biology, Nankai University, Tianjin 300353, China. [5]Laboratory of Structural Biology, Tsinghua University, 100084 Beijing, China. [6]Innovative Center for Pathogen Research, Guangzhou Laboratory, Guangzhou 510005, China. [7]These authors contributed equally: Jing Yu, Yuhui Lan. ✉e-mail: yanght@shanghaitech.edu.cn; raozh@mail.tsinghua.edu.cn; lijun1@shanghaitech.edu.cn

ATP induces association and dissociation between the NBDs. This results in conformational changes of the TMDs that are likely transmitted through the coupling helices (CpHs)[13]. Consequently, the substrate binding cavity (central cavity) between the two TMDs will open either towards the cytoplasmic side (inward facing, **IF**) or towards the periplasmic side (outward facing, **OF**) of the cell membrane, or be occluded (**Occ**) from both sides.

Transporters with two active nucleotide-binding sites (NBSs) are referred to as canonical transporters, while heterodimeric ABC exporters frequently have both a degenerate NBS and a consensus NBS. Compared to the consensus site with active ATPase activity, the degenerate site contains non-canonical residues that strongly impair ATP hydrolysis[14]. Thus, the proposed models of transport cycle for the canonical transporters are not compatible with a 'degenerate' transporter, because these models (a) require the normal hydrolysis of two ATPs for one cycle, (b) an obligate alternation of ATP hydrolysis between the two NBSs, or (c) impose complete NBD separation[14]. These 'degenerate' transporters must therefore have established a distinct mechanism, in which the degenerate NBS sustains active substrate translocation across a biological membrane[14]. Indirect evidence suggested that ATP might act as a glue in the degenerate site, keeping this NBS in a closed conformation over several iterations of the transport cycles[15–19]. Mutagenesis data suggested that ATP hydrolysis in the consensus site is sufficient for all necessary conformational changes to complete a transport cycle, but subsequent hydrolysis of ATP in the degenerate site would favor NBD opening[20]. Several biochemical and structural investigations have been undertaken on this type of ABC exporters including TM278/288[21–23], TmrAB[24,25], TAP[26] and CFTR[27–29]. However, the detailed mechanism and why these transporters have an impaired NBS remains unclear. Therefore, more structural evidence is required to explain the exact role of the degenerate site during the process of transport. The degenerate sites of different transporters often have unique residue substitutions[30], thus, this site may prove to be a good target for the design of highly specific drugs[20].

The genes encoding ABC transporters occupy about 2.5% of the genome of *Mtb*[31]. Among them, *rv1273c* and *rv1272c* are arranged in tandem. Their coding proteins, Rv1273c and Rv1272c, are TMD-NBD fusion proteins and they are predicted to form a heterodimeric ABC exporter[31]. However, Rv1272c is hypothesized to function as a homodimer. Interestingly, it has been shown to enhance the transport of long-chain fatty acids when expressed in *Escherichia coli* (*E. coli*)[32]. Rv1273c has 28% sequence identity with Rv1272c[32] and is capable of modulating cell wall lipid composition, promoting mycobacterial survival within macrophages and in affecting the host cell immune response[33]. It has also been proposed as a contributor to biofilm formation[34]. However, whether Rv1273c interacts with Rv1272c to function together as a transporter has remained unknown. Based on their sequences, Rv1273c and Rv1272c have been annotated as multidrug transporters[31]. Expression and mutation analysis of drug resistant clinical isolates of *Mtb* showed that Rv1273c is associated with resistance to drugs such as isoniazid[35], ethambutol[36] and bedaquiline[37]. An analysis of extensively drug-resistant (XDR) clinical isolates of *Mtb* revealed common single nucleotide polymorphisms (SNPs) in Rv1273c such as S118G and I175T, which are believed to be associated with drug resistance caused by Rv1273c[38]. Another cell-based study also indicated that Rv1273c acts as a multidrug efflux pump targeting a wide range of antibiotics[34]. Given their critical importance, Rv1273c and Rv1272c are thus potential targets for the treatment of TB and for mitigating the effects of drug resistance. However, currently, little is known about the molecular mechanism as to how Rv1273c and Rv1272c function as transporter proteins.

*Ms*Rv1273c (MSMEG_5008) and *Ms*Rv1272c (MSMEG_5009) from *Mycobacterium smegmatis* (*Msm*) are close homologs of Rv1273c and Rv1272c, respectively, sharing 69% and 75% sequence identities with their *Mtb* counterparts. We focus on the *Msm* homologs because they

could be successfully expressed and purified with good quality while this is not possible for the *Mtb* proteins. Here, we purify the active *Ms*Rv1273c/72c (complex of *Ms*Rv1273c and *Ms*Rv1272c) transporter as a heterodimer, confirm its role as a drug efflux pump of isoniazid, and determine its cryo-electron microscopy (cryo-EM) structures in the apo / AMPPNP-bound / ADP-bound / ATP | ADP-bound **IF** states and the ATP-bound / ATP | ADP+Vi-bound **Occ** states. Importantly, we capture three types of asymmetric conformations as intermediate states that have not been previously reported for any of the ABC superfamily. With these data, the mechanism of degenerate NBS mediated transport by a heterodimeric ABC exporter is proposed.

## Results

### Characterization and functional analysis of *Ms*Rv1273c/72c

Rv1273c and Rv1272c, as well as *Ms*Rv1273c and *Ms*Rv1272c, were tandemly expressed in *Msm* cells and then purified in digitonin as a complex with a Flag-tag or 6×His-tag fused at the C-terminus of Rv1272c and *Ms*Rv1272c, respectively. Gel filtration and SDS-PAGE showed that the two subunits form a heterodimer with 1:1 stoichiometry (Supplementary Fig. 1a, b). Since the sample quality of Rv1273c/72c was poor due to low yields and impurities, it was difficult to perform biochemical and structural studies. We therefore focused on *Ms*Rv1273c/72c. To prove that *Ms*Rv1273c/72c can undertake ATP hydrolysis, an ATPase activity assay was performed. The results showed that it has basal ATPase activity of $113.3 \pm 13.9$ nM $P_i$ min$^{-1}$ mg$^{-1}$ (mean $\pm$ S.E.M., $n = 3$), $181.2 \pm 2.3$ nM $P_i$ min$^{-1}$ mg$^{-1}$ and $147.9 \pm 2.4$ nM $P_i$ min$^{-1}$ mg$^{-1}$ in detergent, proteoliposomes and peptidiscs, respectively (Fig. 1a). These data show that *Ms*Rv1273c and *Ms*Rv1272c are able to form an active heterodimeric transporter.

Sequence alignment of NBDs of *Ms*Rv1273c/72c with other ABC transporters showed that Glu553 in the Walker B motif of *Ms*Rv1272c is a conserved catalytic residue, which is responsible for activating the attacking water molecule during ATP hydrolysis[39]. Another conserved residue, His584, in the Switch-loop of *Ms*Rv1272c plays a central role in stabilizing the transition state of the reaction[40]. However, the corresponding residues in *Ms*Rv1273c are replaced by Asp497 and Gln528, respectively (Fig. 1b). Thus, *Ms*Rv1273c should have a degenerate site impaired for ATPase activity while *Ms*Rv1272c has a functional consensus site with the classic enzymatic motifs. Based on this analysis, we performed a mutagenesis study and used an ATPase activity assay to verify the roles of the expected catalytic residues (e.g., E553 acts as a catalytic base?). The result shows that the E553Q mutation at the consensus site significantly diminishes ATP hydrolysis, consistent with mutagenesis experiments for other ABC transporters where E has been changed to Q[21,41]. This confirms that Glu553 is crucial for ATPase activity in *Ms*Rv1273c/72c. Note that this mutant has some residual activity (Fig. 1c), this is likely because Asp497 at the degenerate site can still slowly hydrolyze ATP. Since the side chain of Asp497 is shorter than Glu, but can still act as a base, it may not be ideal for optimal catalysis. In contrast, the D497N mutant did not show a significant loss of ATPase activity compared to the wild-type transporter (Fig. 1c). This is because Glu553 is still fully functional at the consensus site. Confirmation that this is correct was achieved by mutating both E553Q and D497N, creating the double mutant, which has little activity (Fig. 1c). Together, these results indicate that the consensus site of *Ms*Rv1273c/72c is functional with normal ATPase activity while its degenerate site exhibits very low activity.

Since Rv1273c and Rv1272c could potentially play a part in the resistance mechanism for drugs such as isoniazid and ethambutol, we first tested their effects on ATP hydrolysis using the *Ms*Rv1273c/72c complex. In this experiment, neither drug affected ATPase activity significantly (Supplementary Fig. 1c, d), this may be due to their weak binding, which is commonly observed for a non-physiological substrate. Next, we established a transport assay to determine whether isoniazid or ethambutol could be imported into proteoliposomes

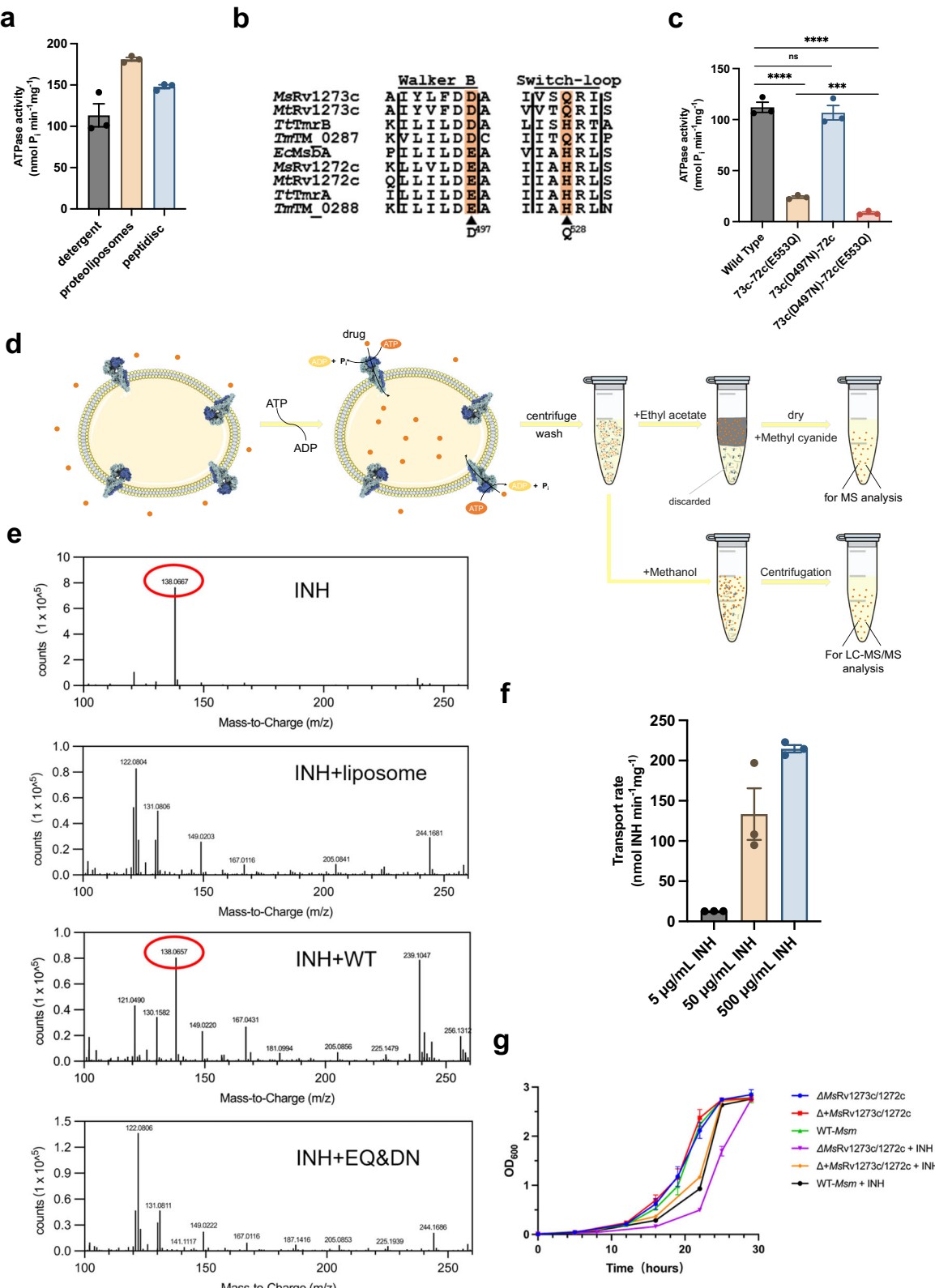

containing *Ms*Rv1273c/72c (Fig. 1d). Mass spectrometry detected isoniazid inside the liposomes when the wildtype transporter was inserted into the liposomes, while there was no detectable isoniazid inside the liposomes when the ATPase inactive mutant was used (Fig. 1e). When ethambutol was tested, it was not detected inside the liposomes in any of the experiments (Supplementary Fig. 1g). These results confirmed that *Ms*Rv1273c/72c is able to transport isoniazid but not

ethambutol. Further analysis by Liquid Chromatography-Tandem Mass Spectrometry (LC-MS/MS) showed that isoniazid is transported by *Ms*Rv1273c/72c in a dose-dependent manner (Fig. 1d and f). In agreement with this, *Msm* culture growth experiments showed that knocking out of *Ms*Rv1273c/72c resulted in an increase in growth inhibition when isoniazid was added, while complementation of *Ms*Rv1273c/72c in the knockout strain eliminated such a growth

**Fig. 1 | Characterization and functional analysis of *Ms*Rv1273c/72c. a** The ATPase activity of *Ms*Rv1273c/72c in detergent, proteoliposomes and peptidiscs. Data are presented as the mean ± S.E.M., calculated from three biologically independent experiments ($n = 3$). **b** Sequence alignment of the Walker B motif and Switch-loop for Rv1272c, Rv1273c from *Mtb* (*Mt*) and *Msm* (*Ms*), and other representative ABC transporters including TmrAB from *Thermus thermophilus* (*Tt*), TM_0287/88 from *Thermotoga maritima* (*Tm*) and MsbA from *E. coli* (*Ec*). The crucial sequence variations between the degenerate and consensus sites are highlighted in orange. **c** The ATPase activity of *Ms*Rv1273c/72c and mutants. Data are presented as the mean ± S.E.M., calculated from three biologically independent experiments ($n = 3$). *P* values were calculated using an unpaired two-sided *t* test. ***, $P < 0.001$; ****, $P < 0.0001$; ns, not significant. Wild Type vs D497N, $P = 0.57$. **d** A scheme of proteoliposomes-based transport assay after which the contents inside the liposomes were isolated and analyzed by MS or LC-MS/MS method. **e** Mass spectrometry was used to determine the contents inside the liposomes with wildtype *Ms*Rv1273c/72c (WT), or with the E553/D497N double mutant (EQ&DN) inserted, and liposomes without any protein added. Pure isoniazid (INH) was also measured as a standard. The red circle indicates the expected mass/charge of the drug. **f** The transport rate of varying concentrations of isoniazid supplied outside the proteoliposomes. Data are presented as the mean ± S.E.M., calculated from three biologically independent experiments ($n = 3$). **g** Growth curves of wild type *Msm* (WT-*Msm*), *Ms*Rv1273c/72c knockout strain (Δ*Ms*Rv1273c/72c), and complemented strain containing pMV261-*Ms*Rv1273c/72c (Δ+*Ms*Rv1273c/72c) in the presence or absence of isoniazid. Data are presented as the mean ± S.E.M., calculated from three biologically independent experiments ($n = 3$).

difference in the presence of isoniazid (Fig. 1g). These results confirm that *Ms*Rv1273c/72c is involved in resistance to isoniazid. However, no growth difference was observed in all these strains in the presence of ethambutol (Supplementary Fig. 1h), which suggests that *Ms*Rv1273c/72c is not sensitive to ethambutol. Thus, *Ms*Rv1273c/72c is a drug efflux pump for isoniazid, but not ethambutol. Therefore, its *Mtb* counterpart is a likely factor contributing to isoniazid resistance in *Mtb*.

### Structure in the IF$^{apo}$ state

Using single-particle 3D reconstruction, the cryo-EM structure of *Ms*Rv1273c/72c was determined at 3.1 Å resolution in the IF$^{apo}$ state (Supplementary Fig. 2 and 11a). The overall fold of the *Ms*Rv1273c/72c complex (Fig. 2a) belongs to the type IV family of ABC transporters (or type I ABC exporter) with a seven-type classification[42]. The central cavity is located between the two TMDs and it opens towards the cytoplasm (Fig. 2b). The two NBDs are separate at both NBSs (Fig. 2c). However, interactions are maintained between the two NBDs at their C-terminal helices (Fig. 2a). In the heterodimer, *Ms*Rv1272c and *Ms*Rv1273c share a similar fold (Fig. 2d). Superposition shows that the *r.m.s.d.* between the two subunits is 2.59 Å for 413 pairs of Cα atoms, suggesting there are significant differences in the two subunits. We observe that the gap between TM4 and TM6 of *Ms*Rv1272c (12.9 Å between Val243 and Gln358) is larger than that of *Ms*Rv1273c (5.0 Å between Thr184 and Pro299), so that the N-terminal extending loop of *Ms*Rv1272c could insert into the gap between Val243 and Gln358 (Fig. 2d). This is because the bending point of the kinked TM6 (Ala351) is at a higher position compared to its equivalent in *Ms*Rv1273c (Pro299) (Fig. 2d). A small relative rotation of NBDs is also observed by superposition (Fig. 2d), thus the NBD of *Ms*Rv1272c holds the CpH more deeply and tightly in its cleft.

Interestingly, the structure of the transporter is asymmetrical, and in particular there are two additional structural features found in *Ms*Rv1272c which might be function related: *(1) The N-terminal extending loop that inserts into the substrate binding cavity.* The N-terminal extending loop (includes residues 1-16, but only 10-16 are observed in the IF$^{apo}$ structure) of *Ms*Rv1272c wraps around TM6 by forming a right angle with the elbow helix (EH) and then interacts with the sidechain of Trp246 in TM4, which is conserved among the mycobacterial Rv1272c homologs (Supplementary Fig. 12c). After that, the extending loop inserts into the gap between TM4 and TM6 (Fig. 2e and Supplementary Fig. 12a). We cannot further trace its structure inside the central cavity due to flexibility (residue 1-9 is missing). Since the extending loop is inserted and points into the central cavity in the IF$^{apo}$ state (Fig. 2e), we hypothesize that this loop might help the substrate enter the cavity. It remains unknown why the length and residue composition of the extending loop varies in different mycobacterial homologs of Rv1272c (Supplementary Fig. 12d). (2) *The periplasmic helical hairpin structure that caps the exit of substrate-binding cavity.* The extra-cellular domain (ECD) is formed by the extracellular loop 1 (ECL1, residue 62-109) between TM1 and TM2. It contains two short horizontal helices (H1 and H2) forming a helical hairpin structure

and the protruding helix extending from TM1 (Fig. 2f). H1 interacts with ECL3 of *Ms*Rv1272c and ECL1 of *Ms*Rv1273c and H2 stacks above H1. The location of helical hairpin structure is on the path of the substrate releasing from the central cavity (Fig. 2f). Thus, the role of the ECD in this transporter is likely to help substrate release.

### Structure in the ATP-bound Occ state

An approach to obtain the **Occ** state of ABC transporter is to inactivate the ATPase activity by making a glutamate to glutamine (EtoQ) mutation in the Walker B motif of NBD. This allows the trapping of the transporter in a pre-hydrolytic state when adding ATP[43]. Based on this, we purified the E553Q mutant which had a significantly diminished ATPase activity and then added ATP before data collection. We then solved the cryo-EM structure of E553Q mutant in the presence of ATP at 3.1 Å resolution, which showed an ATP-bound **Occ** state (Fig. 3a, Supplementary Fig. 3, 11h). In another experiment, vanadate (Vi) was added in an attempt to trap the transporter in a conformation immediately after ATP hydrolysis[43]. In such a state, ADP together with vanadate should stabilize the NBD dimer and trap an **Occ** state. After treatment with ATP and vanadate, the cryo-EM structure of *Ms*Rv1273c/72c was solved and showed an ATP | ADP+Vi-bound **Occ** **(Vi)** state at 2.7 Å resolution (Supplementary Figs. 10, 11i, 16b and 16c). Since both structures adopt a similar conformation, only the ATP-bound **Occ** state was analyzed and compared here.

In the **Occ** structure, the central cavity in the TM region is sealed at both the cytoplasmic and periplasmic sides (Fig. 3b). The N-terminal extending loop is squeezed out from the cavity (Supplementary Fig. 12b). The two NBDs are fully closed and interact with each other in an antiparallel manner (Fig. 3c). ATP is clamped between the dimer interface at the consensus and degenerate NBSs. The binding mode of ATP is similar at both sites and is classical amongst the ABC transporter family[42]. The binding and hydrolysis of ATP involves several conserved motifs around the binding site including Walker A motif, Walker B motif, A-loop, Q-loop, Switch-motif of one NBD and Signature motif of another (Fig. 3d, e). Taking ATP binding at the degenerate site for example, the α- and β-phosphate group of ATP binds in the groove formed by Walker A motif of *Ms*Rv1273c, while the γ-phosphate group is stabilized by Mg$^{2+}$ and helix α6 of *Ms*Rv1272c. The adenosine group of ATP is sandwiched by the A-loop and Signature motif *Ms*Rv1273c (Fig. 3f).

### Structure in the AMPPNP-bound IF$^{asym-1}$ state

To capture a pre-hydrolytic state occurring before the **Occ** state, we treated the cryo-EM sample with AMPPNP, since it is a non-hydrolysable analog of ATP. The cryo-EM structure of *Ms*Rv1273c/72c in the AMPPNP-bound IF$^{asym-1}$ state was then obtained (Fig. 4a, Supplementary Figs. 4, 11b). Similar to the IF$^{apo}$ state, the central cavity still opens towards the cytoplasm and the extending loop also inserts into the gap between TM4 and TM6 of *Ms*Rv1272c (Supplementary Fig. 12e). However, in this state, the overall conformation is asymmetric, especially at the NBDs. Though both NBSs bind AMPPNP, the NBDs are

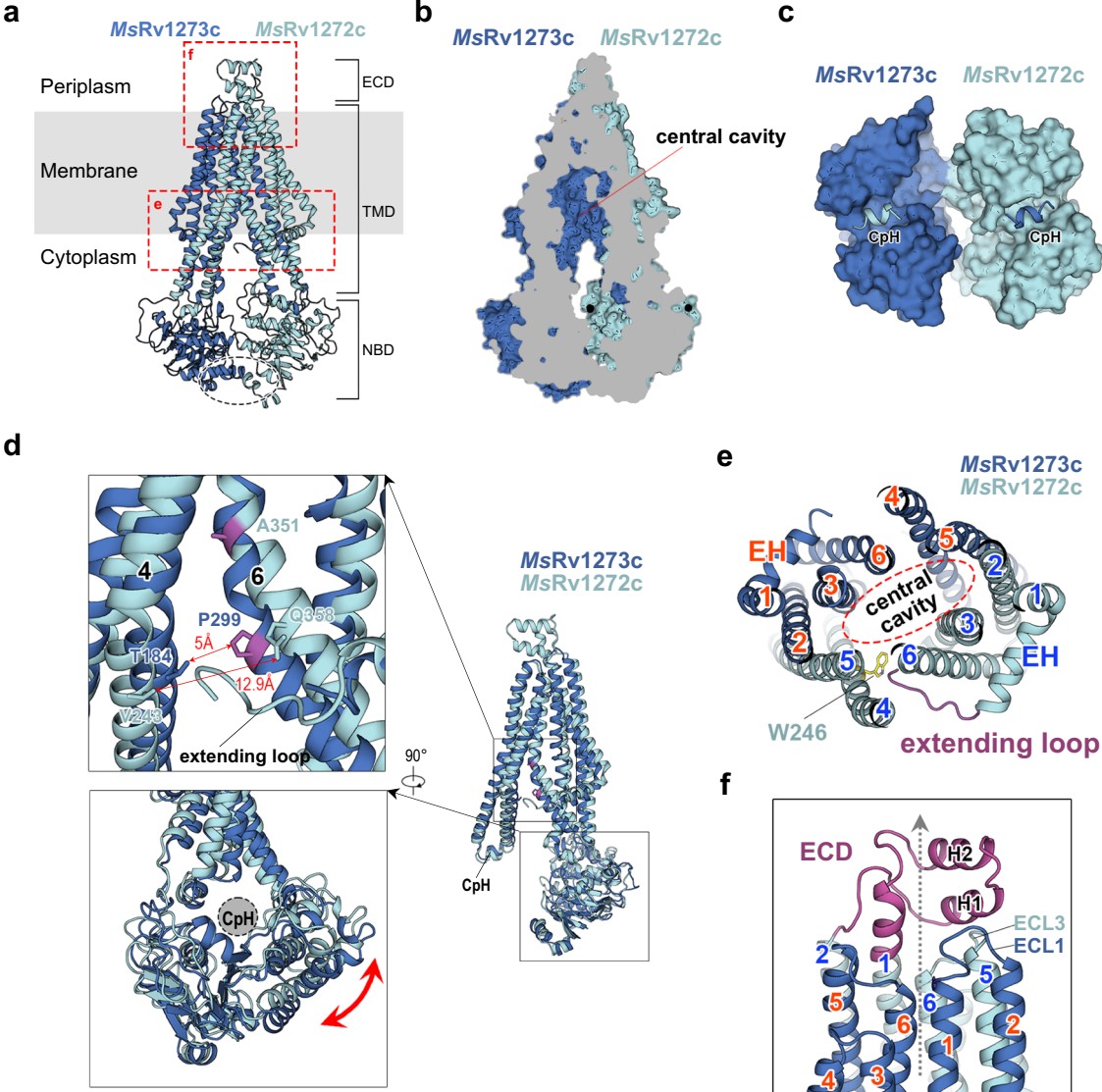

**Fig. 2 | Structure of *Ms*Rv1273c/72c in the IF$^{apo}$ state. a** Overall view of the cryo-EM structure of *Ms*Rv1273c/72c in the *apo* form **IF$^{apo}$**. The black dashed circle indicates that there are dimer interactions between the two NBDs. The structure in the dashed boxes is shown in detail in panel **e** and **f**. TMD, transmembrane domain; NBD, nucleotide-binding domain; ECD, extra-cellular domain. **b** Clipped view of structure to show the central cavity in the TM region which opens towards the cytoplasmic side. **c** Surface representation of NBDs, viewed from the periplasm. CpH, coupling helix. **d** Superposition between *Ms*Rv1272c and *Ms*Rv1273c. Zoom-in views are shown in the inlets. Pro299 and Ala351 (magenta) are the hinge points of TM6 in the two subunits. The distance between TM4 and TM6 in both structures is provided. The rotation between NBDs is marked with a red double headed arrow. **e** Clipped view of the TMDs looking from the periplasm. The extending loop is colored in magenta. Trp246 is shown as yellow sticks. The central cavity is indicated by the red dashed circle. EH, elbow helix. **f** Close-up view of ECD (magenta) in *Ms*Rv1272c. The gray dotted arrow indicates the substrate releasing path when the TM region is separated into two wings in the **OF** state. ECL, extracellular loop; H, helix.

partially dimerized at the degenerate site whereas there is a gap between NBDs at the consensus site (Fig. 4b). Thus, the ability of AMPPNP to mediate NBD dimerization at consensus site is reduced compared with that at degenerate site. The binding mode of AMPPNP at the consensus site in the **IF$^{asym-1}$** structure superposes well with that of ATP in the **Occ** structure while the nucleotide binds distinctly at the degenerate site (Supplementary Fig. 13j, k). In the **IF$^{asym-1}$** state, we observed that the whole NBD of *Ms*Rv1272c binds to the other NBD with a rotation angle of 8.1° around the degenerate site (Fig. 4b). Due to this rotation, helix α5 moves closer to the A-loop of *Ms*Rv1273c (Fig. 4c) and as a result the nucleotide is more buried in the degenerate site compared to the ATP-bound **Occ** structure (Fig. 3f). Helix α6 is pointing away from the γ-phosphate group, and therefore cannot stabilize it by using its dipole. Moreover, the Signature motif moves away from the degenerate site following the shift of helix α6 and is not able to clamp the adenosine ring of the nucleotide together with Tyr343 of A-loop (Fig. 4c). These observations indicate that AMPPNP binds weakly at the degenerate site in the **IF$^{asym-1}$** state.

**Structure in the ADP-bound IF$^{asym-3}$ state**

To capture more post-hydrolytic states after the **Occ** state, we prepared cryo-EM samples with different treatments of nucleotides. We obtained two ADP-bound cryo-EM structures after samples were treated with ATP at 37 °C or ADP at 4 °C (Supplementary Fig. 6, 7, 11d, 11e). Since the two structures show a similar asymmetric conformation and the same bound nucleotide (Supplementary Fig. 16a), they can be considered as one state, the **IF$^{asym-3}$** state. We then used the ATP (37 °C) treated **IF$^{asym-3}$** structure for further analysis. In this structure, ADP

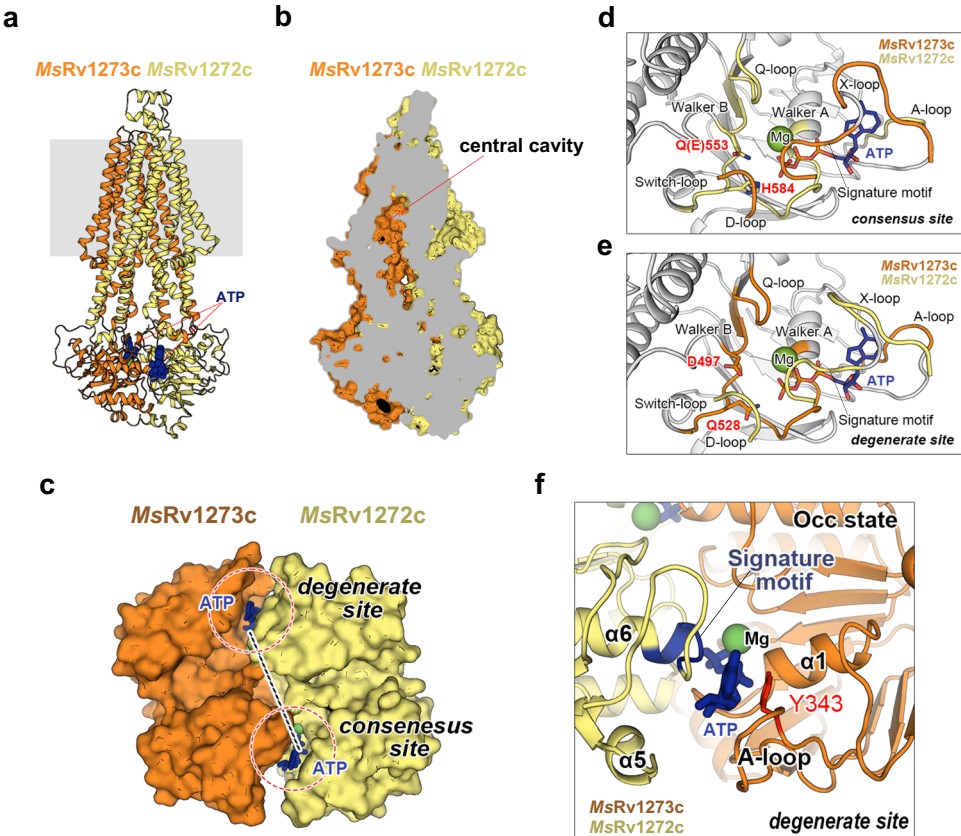

**Fig. 3 | Structure in the ATP-bound Occ state. a** Overall view of the cryo-EM structure of *Ms*Rv1273c/72c in the ATP-bound **Occ** state. ATP is shown as blue spheres. **b** Clipped view of the structure in the **Occ** state. The central cavity in the TM region is sealed at the periplasmic and cytoplasmic sides. **c** Surface representation of NBD dimer in the ATP-bound **Occ** structure, viewed from the periplasm. The degenerate site and consensus site are marked with red dashed circles. Nucleotides are shown as sticks. The dashed line linking β-phosphorus atoms at both sites serves as the reference line for measuring NBD rotation in other states. **d** The consensus nucleotide-binding site. The Q-loop, Walker B motif, Switch-loop, Walker A motif and A-loop in *Ms*Rv1272c are highlighted in yellow, while the D-loop,

Signature motif and X-loop in *Ms*Rv1273c are highlighted in orange. Glu553 (here mutated to Gln) and His584 of *Ms*Rv1272c are shown as sticks. ATP and Mg²⁺ are represented as blue sticks and green spheres. **e** The degenerate nucleotide-binding site. The Q-loop, Walker B motif, Switch-loop, Walker A motif and A-loop in *Ms*Rv1273c are highlighted in orange, while the D-loop, Signature motif and X-loop in *Ms*Rv1272c are highlighted in yellow. Asp497, Gln528 of *Ms*Rv1273c are shown as sticks. **f** Close-up view of the degenerate site in the ATP-bound **Occ** structure. The Signature motif and Tyr343 of the A-loop are highlighted in blue and red, respectively.

binds at the consensus site (Fig. 4d) as a hydrolyzed product after ATP hydrolysis. Interestingly, there is no nucleotide at the degenerate site even though we added ATP or ADP in the sample. The NBDs are also partially dimerized (Fig. 4e), but the conformation is different from the **IF^{asym-1}** state. We found the gap at the consensus site is larger in this **IF^{asym-3}** structure. We also observed that the whole NBD of *Ms*Rv1272c binds to the other NBD with a rotation angle of 23.8° around the degenerate site (Fig. 4e). The dimerization is only based on direct interactions between the two NBDs at the degenerate site (Fig. 4f). Due to a large rotation of the NBD in *Ms*Rv1272c, helix α5 inserts into the inner side of the A-loop of *Ms*Rv1273c and occupies the binding position of the adenosine group of ATP together with the side chain of Tyr343. The direction of helix α6 also significantly deviates from the phosphate binding site of ATP resulting in the gap between the Signature motif and Tyr343 of the A-loop being too small to clamp the adenosine group of ATP (Fig. 4f). Thus, such a distorted degenerate site does not allow the binding of ATP.

### Structure in the ATP|ADP-bound IF^{asym-2} state
When the sample was treated with ATP at 4 °C, we obtained a cryo-EM structure of *Ms*Rv1273c/72c in the **IF^{asym-2}** state (Fig. 4g, Supplementary Fig. 5, 11c). In this structure, the conformation is also asymmetric but is different from the **IF^{asym-1}** and **IF^{asym-3}** states. The NBDs are partially

dimerized at the degenerate site with a larger rotation angle of 32.8° (Fig. 4h). The consensus site is bound with an ADP, which originates from the rapid hydrolysis of ATP, while the degenerate site is bound with an ATP, which has not been hydrolyzed. The structure of the ATP bound site is also distorted (Fig. 4i). The direction of helix α6 deviates further compared to the **IF^{asym-3}** state and it is not able to interact with the γ-phosphate group of the bound ATP. The Signature motif linking to α6 is still able to weakly clamp ATP with Tyr343 in the A-loop (Fig. 4i). Considering the sample preparation condition, this **IF^{asym-2}** structure may represent a state before **IF^{asym-3}** and after **Occ**.

To support this proposal, we performed molecular dynamics (MD) simulation starting from the **IF^{asym-2}** structure by removing ATP in the degenerate site, then we inspected the conformations and analyzed the changes in the distance between the two NBDs close to the consensus site during 100 ns simulation (Supplementary Fig. 14d). We found that the NBDs keep interactions at the degenerate site and **IF^{asym-3}** liked states with distance similar to the **IF^{asym-3}** state could be induced from the **IF^{asym-2}** state if ATP in the degenerate site is released.

### Conformational changes between different states
Next, we analyzed conformational changes by comparing the different states of *Ms*Rv1273c/72c. In the **IF^{apo}** structure, the NBDs are separated both at consensus and degenerate sites by 8.4 Å and 10.7 Å,

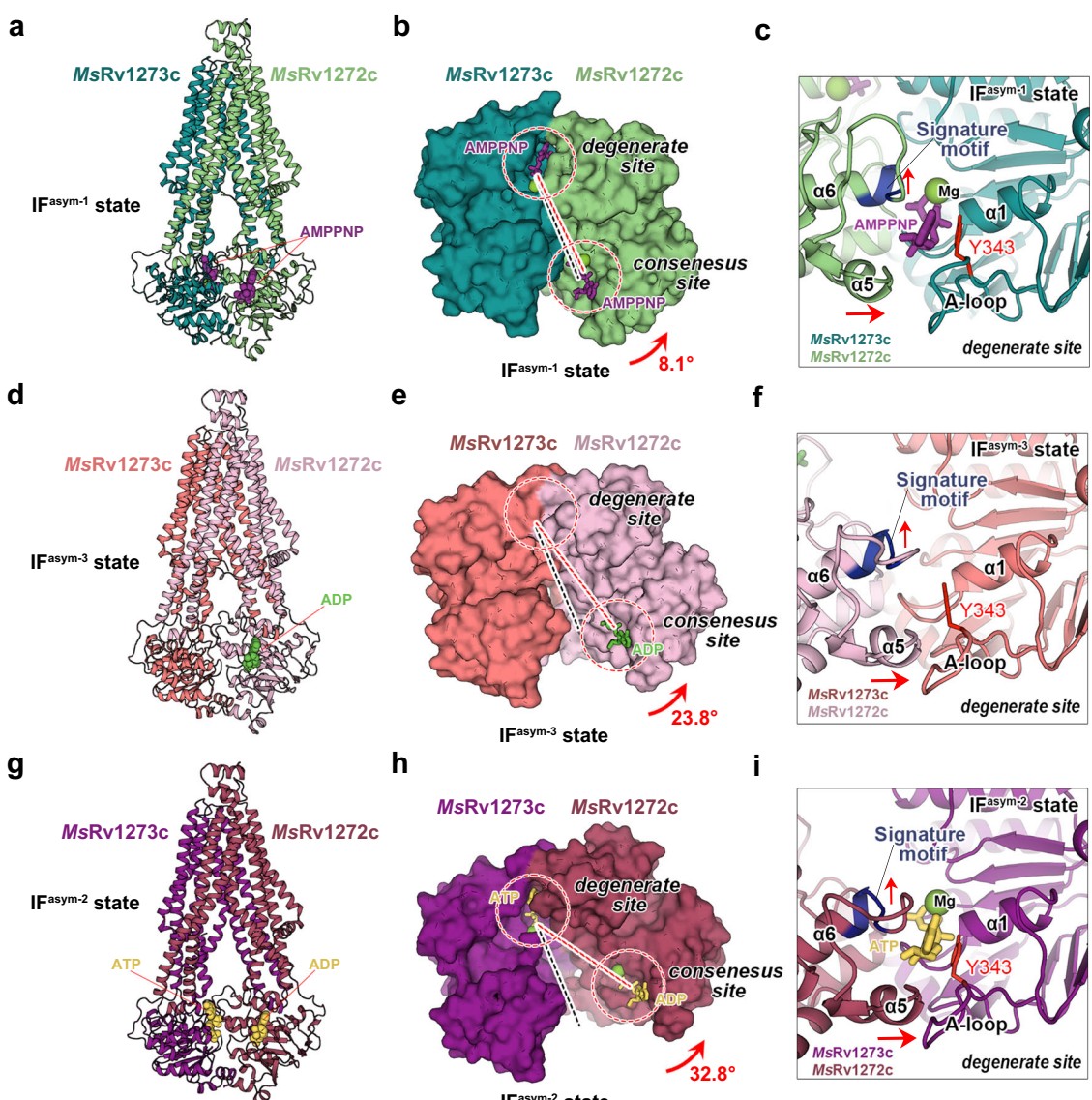

**Fig. 4 | Structures in the IF^asym states. a** Overall view of the cryo-EM structure of *Ms*Rv1273c/72c in the AMPPNP-bound **IF^asym−1** state. AMPPNP is shown as purple spheres. **b** Surface representation of the NBD dimer in the **IF^asym−1** structure, viewed from the periplasm. The NBD of *Ms*Rv1273c is fixed as in Fig. 2g and the rotation (red arrow) of NBD of *Ms*Rv1272c is measured based on the position of the β-phosphorus atom of the nucleotide in the NBS (the angle between the two dashed lines linking β-phosphorus atoms at both sites). The degenerate site and consensus site are marked with red dashed circles. AMPPNP is shown as sticks. **c** Close-up view of the degenerate site in the **IF^asym−1** structure. The Signature motif and Tyr343 of

the A-loop are highlighted in blue and red, respectively. The shifts of α5 and α6 relative to the **Occ** structure in Fig. 3f are marked with red arrows. **d** Overall view of the cryo-EM structure of *Ms*Rv1273c/72c in the ADP-bound **IF^asym−3** state. ADP is shown as green spheres. **e** Surface representation of NBD dimer in the **IF^asym−3** structure. **f** Close-up view of the degenerate site in the **IF^asym−3** structure. **g** Overall view of the cryo-EM structure of *Ms*Rv1273c/72c in the ATP | ADP-bound **IF^asym−2** state. ATP and ADP are shown as yellow spheres. **h** Surface representation of NBD dimer in the **IF^asym−2** structure. **i** Close-up view of the degenerate site in the **IF^asym−2** structure.

respectively (Fig. 5a). NBDs are partially dimerized at the degenerate site in the three asymmetric states. The corresponding distances are 9.6 Å and 7.9 Å for the **IF^asym−1** structure, 17.0 Å and 7.6 Å for the **IF^asym−2** structure, 13.5 Å and 8.2 Å for the **IF^asym−3** structure. In the ATP-bound **Occ** structure, the NBDs are fully dimerized with distances of 6.8 Å and 6.1 Å at the two NBSs. During NBD closure, the two CpHs in the TMDs move closer to each other with the distance reducing from 23.5 Å to 12.5 Å (Supplementary Fig. 14a). Superposition of the TMDs for each subunit shows that TM4-5 shifts towards the helix bundle composed of TM2-3 and TM6 at the cytoplasmic side (Supplementary Fig. 14b, c). These changes lead to an opening and closing of the central cavity as the structure moves between the **IF** and **Occ** states. Thus, the three **IF^asym** states adopt intermediate conformations.

Besides the rigid-body movement of NBDs during NBD closure, local structural rearrangements are also observed in the NBD of *Ms*Rv1272c. For example, the region containing a part of the D-loop, a part of the Walker B motif and the connecting residues (554AT555), which we assigned as the D-WalkerB loop (552DEATSSVD559), adopts a classical S-shaped conformation when the NBD binds nucleotide (Fig. 5b and Supplementary Fig. 13h). However, the loop is adjusted to form an L-shaped conformation in the **IF^apo** state (Fig. 5b and Supplementary Fig. 13g). In this state, the catalytic residue, Glu553, points in the opposite direction to the consensus NBS. Meanwhile, Asp552 of the Walker B motif also shifts away from the NBS (Fig. 5c). On the other hand, the locations of α7 and α8 have changed and a gap is created to accommodate the side chain of Glu553. Glu553 is further stabilized

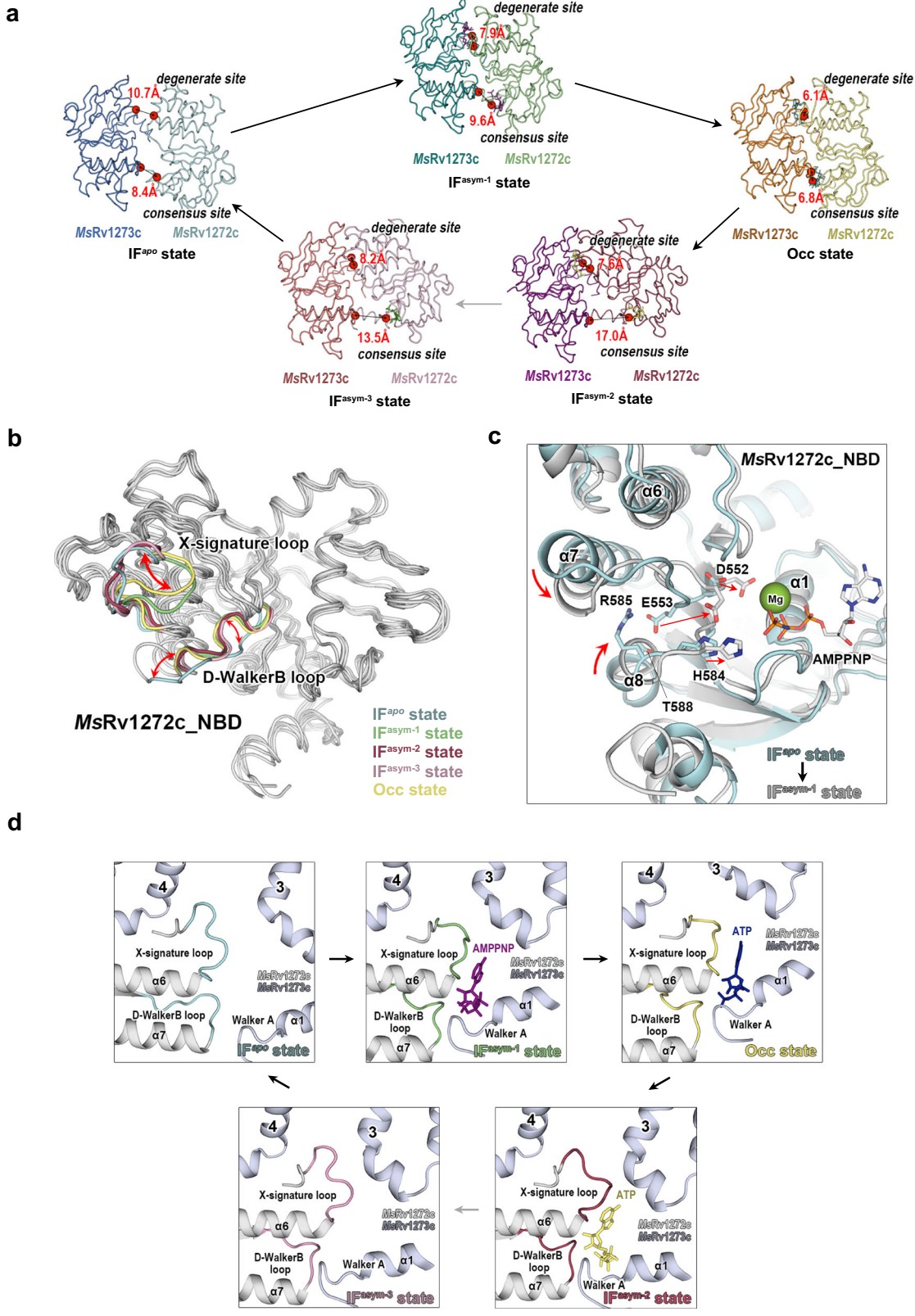

**Fig. 5 | Conformational changes of NBDs in different states. a** Changes in distance between the two NBDs in the five states. The distances at the degenerate site are measured between Gly371 of *Ms*Rv1273c and Ser529 of *Ms*Rv1272c, and the distances at the consensus site are measured between Ser473 of *Ms*Rv1273c and Gly427 of *Ms*Rv1272c. The position of all the measured residues is indicated by red spheres. **b** Superposition of NBDs of *Ms*Rv1272c in the five states. The shifts of

D-WalkerB loop and X-signature loop are marked with red double headed arrows. **c** Conformational changes of NBD of *Ms*Rv1272c from IF*apo* state (cyan) to IF*asym-1* state (gray). Important residues are shown as sticks. Shifts of helices and residues are marked with red arrows. **d** Separate views of the degenerate site for the five states show the conformational changes of the D-WalkerB loop and X-signature loop.

through interactions with Arg585 and Thr588. Meanwhile, His584 of the Switch-loop linking to α8 is also pulled away from the NBS (Fig. 5c). Thus, the consensus NBS is distorted in the **IF**$^{apo}$ state. So, how can structural rearrangements occur to activate the consensus site? We found that the partial dimerization at the degenerate site in the intermediate state (**IF**$^{asym-1}$) is indispensable to trigger the activation. When the two NBDs approach each other, α7 and the D-loop of *Ms*Rv1272c will clash with the NBD of *Ms*Rv1273c at the degenerate site (Supplementary Fig. 13i). Thus, under the packing force, α7 will be squeezed backwards and towards α8 (Fig. 5c, d). The D-WalkerB loop will be crimped to the "S" conformation and the whole consensus NBS will be positioned to bind and hydrolyze ATP (Fig. 5c, d). Thus, the classical and active form of NBD of *Ms*Rv1272c is only stabilized upon NBD closure at the degenerate site. This could prevent futile consumption of ATP at the consensus site when NBD dimerization at the degenerate site does not happen. These analyses show that the D-WalkerB loop of *Ms*Rv1272c plays a crucial role in coordinating the allosteric coupling between the degenerate site and consensus site through conformational changes. Note that similar "L" and "S" conformations of D-WalkerB loop have also been observed in the **IF** and **OF** structures of TM287/288[21,22]. However, no structural evidence for how and when they transit between each other under the intermediate states has been provided previously.

Another structural rearrangement is in the fragment we assigned as the X-signature loop ($^{522}$DDDGGAIS$^{529}$) which contains a part of the X-loop, a part of the Signature motif and the connecting residues ($^{526}$GA$^{527}$) (Fig. 5b). It is stretched into the gap between the two CpHs in the **IF**$^{apo}$ and **IF**$^{asym-3}$ states (Fig. 5d) so that the Signature motif at the degenerate site cannot bind ATP (Supplementary Fig. 13i). In the **Occ** state and **IF**$^{asym-1}$ state, the X-signature loop is clamped by the two approaching CpHs. As a result, the Signature motif could be re-shaped in a favorable position to interact with ATP at the degenerate site (Fig. 5d). From this point of view, the **IF**$^{asym-3}$ state is closer to the **IF**$^{apo}$ state while the **IF**$^{asym-1}$ state is more similar to the **Occ** state.

Altogether, the above analysis suggests that through the intermediate **IF**$^{asym}$ states, conformational changes in the D-WalkerB loop and the X-signature loop of *Ms*Rv1272c occur. These changes modulate and couple the function of both the consensus and degenerate NBSs. Therefore, visualization of the intermediate states has explained how the transporter transits between the **IF** state and **Occ** state, which are necessary for the function of this transporter.

## Discussion

In the structures of the heterodimeric transporter of *Ms*Rv1273c/72c, there are several asymmetrical features in comparing the two subunits. *Ms*Rv1273c bears the degenerate site and the classical fold of half an ABC transporter, while *Ms*Rv1272c contains more structural features compared to the canonical type IV ABC transporter. Strikingly, an additional extending loop at the N-terminus of *Ms*Rv1272c is observed. A similar N-terminal extending loop is also observed in TM287/288 (Supplementary Fig. 12e). In that structure (PDB code: 3QF4), it is proposed that this loop restricts the movement of ICL4 (and of NBD1 connected to it) with respect to the other TM helices and might therefore further strengthen the interaction between the NBDs[21]. In the structure of the heme transporter, *Ec*CydDC (PDB code: 8IPS), an additional helix extends from EH of CydD (Supplementary Fig. 12e) and it is responsible for stabilizing the substrate-loaded conformation by interacting with TM4[41]. However, our analysis showed that the extending loop in *Ms*Rv1272c appears more likely to play a role in helping the substrate enter the cavity. The helical hairpin structure of ECD is another additional feature in *Ms*Rv1272c which is also found in several other ABC transporters but with different folds. In BmrCD, the β-stranded architecture of ECD is also formed by ECL1 of the consensus subunit (Supplementary Fig. 12f), and it is thought that translocation of substrate through the TMD is facilitated via an interaction with ECD.

The ECD contributes to the transport cycle by influencing substrate positioning or by promoting formation of the low-affinity site for substrate release[44]. In the ABCA family of exporters, the large and complicated structure of ECDs (Supplementary Fig. 12f) may serve as a temporary storage space or a delivery passage for lipid substrates by forming a hydrophobic tunnel[45]. In LptB$_2$FG, both the β-jellyroll-like ECDs of LptF and LptG (Supplementary Fig. 12f) are involved in LPS transport[46]. These examples suggest that the diverse ECDs of ABC transporters may have a common function to facilitate substrate translocation. For *Ms*Rv1273c/72c, further evidence will be required in future studies to understand the precise roles of the extending loop and helical hairpin. The nucleotide binding position and orientation at the consensus site of *Ms*Rv1273c/72c is classical amongst ABC transporters and is similar within our structures (Supplementary Fig. 13j). However, its binding mode is variable at the degenerate site among the different states (Supplementary Fig. 13j and 13k). Besides these asymmetric features, the D-WalkerB loop and X-signature loop of *Ms*Rv1272c undergo conformational changes to affect ATP binding and hydrolysis at the consensus and degenerate sites and in coordinating the allosteric coupling between the two sites. These analyses help us to understand the unique structural and functional asymmetry in the heterodimeric ABC transporters that are missing in homodimeric transporters.

In this study, we expected to capture the **OF** state by using AMPPNP, vanadate or the EtoQ mutant to induce NBD closure. However, the **IF**$^{asym}$ states and **Occ** state were observed instead of the **OF** state. In the study of TmrAB, the authors suggest that in the resulting **OF** state, the transporter can transit between **OF** and **Occ** conformations until the release of P$_i$ from the canonical site weakens the interactions between the NBDs[25]. Our results imply that the transporter is more stable at the **Occ** state than the **OF** state when the NBDs are fully closed. Addition of substrate could potentially stabilize the **OF** state and move the equilibrium from **Occ** towards **OF** state.

The ADP (4 °C) treated **IF**$^{asym-3}$ state indicates that ADP can bind to the consensus site but cannot bind to the distorted degenerate site. The ATP (37 °C) treated **IF**$^{asym-3}$ state suggests that it may represent a state after ATP hydrolysis. In this structure, ADP binds at the consensus site as a hydrolyzed product, however, it is interesting that NBDs dimerized at the degenerate site without nucleotide binding. Since ATP is prone to induce NBD dimerization and it is not easy to release from NBS, it is possible to be hydrolyzed to ADP and then release from the distorted degenerate site with weak binding affinity. However, more studies are required to determine how the nucleotide is released from this site. The nucleotides (ATP/ADP) are either substrates or products of the transporter, and either induces physiologically relevant conformations. An artificial state is unlikely to be induced by these functional molecules. Thus, we believe that the **IF**$^{asym-3}$ state represents a nucleotide-induced physiological conformation. On the other hand, it is possible that there are contacts between NBDs without nucleotide being present in the NBS. For example, both the C-terminal end of NBDs in our **IF**$^{apo}$ structure (Fig. 2a) and the *apo* structure of TM287/288 (PDB code: 4Q4H)[23] are involved in NBD partial dimerization. To rule out the effect of detergent on the induction of this state, we reconstituted the protein sample in a peptidisc[47] and then solved structures with the same treatments (adding ATP at 37 °C or ADP at 4 °C). These new structures (here we named **IF**$^{asym-3}$ **(peptidisc)** (Supplementary Fig. 8, 9, 11f–g)) adopt the same conformation as the **IF**$^{asym-3}$ state (Supplementary Fig. 16a). This suggests that this conformation is induced by the nucleotide and it is not an artifact of the detergent used.

For the heterodimeric ABC exporters, **IF**, **Occ** and **OF** states have been reported[14]. However, in only a few cases have asymmetrical conformations with NBDs partially dimerized at one NBS been reported. For example, in the structure of heme transporter, CydDC (PDB code: 7ZDA and 7ZDK)[48], the NBDs are partially dimerized at the

functional consensus site and widely separated at the presumed degenerate site (Supplementary Fig. 13d). Note that the dimerized site is contrary to what is observed in the *Ms*Rv1273c/72c structure. In addition, the nucleotide at this site does not participate in dimer formation as is observed in other NBDs. An elexacaftor-bound ABC transporter CFTR Δ508 mutant (PDB code: 8EIG)[49] also has a "cracked-open" NBD dimer structure partially held together by the presence of ATP at the consensus site (Supplementary Fig. 13e). This contrasts with the *Ms*Rv1273c/72c structures that dimerize at the degenerate site. Because elexacaftor is not endogenous to the cell, this is not a conformation likely to exist physiologically. For the heterodimeric ABC exporter TmrAB, conformation space under turnover conditions has been reported. In the captured asymmetrical **IF** structures (PDB code: 6RAM and 6RAL)[25], ATP at the degenerate site mediates the NBD dimerization while a fine crack is observed at the ADP-bound consensus site (Supplementary Fig. 13f). This state is nearly identical to the **Occ** state of TmrAB (PDB code: 6RAI and 6RAK). Here, our structures in the **IF**^asym-1, **IF**^asym-2 and **IF**^asym-3 states (Supplementary Fig. 13a–c) are distinct from previously reported asymmetrical conformations. These **IF**^asym states may represent obligatory intermediate conformations during transport that have never been captured for heterodimeric ABC exporters and help to explain how conformational transition occurs when a degenerate site is present.

Based on the above analysis of cryo-EM structures in the **IF**^apo state, ATP-bound **Occ** state, AMPPNP-bound **IF**^asym-1 state, ATP | ADP-bound **IF**^asym-2 state and ADP-bound **IF**^asym-3 state, as well as previous knowledge of the alternating-access transport model of ABC transporters[50], a detailed mechanism for the *Ms*Rv1273c/72c transporter with a degenerate NBS can now be proposed (Fig. 6): (1) Substrate enters the central cavity with the help of the extending loop in the **IF**^apo state or the later **IF**^asym-1 state. (2) ATP binds to both NBSs and acts as a glue for the partial dimerization at the degenerate site. In the meantime, the X-signature loop of *Ms*Rv1272c is induced to interact with ATP. Upon dimerization, conformational changes of the consensus NBD are triggered, especially for the formation of the "S" conformation of the D-WalkerB loop of *Ms*Rv1272c, and then the

distorted consensus site is activated. The transporter is in the ATP-bound **IF**^asym-1 state. (3) The NBDs are fully dimerized to induce shifts of the TM helices. As a result, the central cavity is occluded and the extending loop is squeezed out. The transporter is in the ATP-bound **Occ** state. (4) The central cavity opens towards the periplasm and substrate is released with the help of ECD. Note that the ATP-bound **OF** state is unstable and it may exist only transiently[25]. (5) ATP hydrolysis occurs at the consensus site during the **Occ** or **OF** stage. (6) Phosphate is released at the consensus site, the NBDs are separated with a large gap at the ADP-bound consensus site while they are still in contact at the ATP-bound degenerate site, accompanied by the changes of the X-signature loop of *Ms*Rv1272c. So, the structure returns to the ATP | ADP-bound **IF**^asym-2 state. (7) ATP is then released from the distorted degenerate site by an unknown mechanism, possibly in the form of ADP after slow hydrolysis. The NBDs are still dimerized at the empty degenerate site. The transporter is then in the ADP-bound **IF**^asym-3 state. (8) NBDs are further separated at the degenerate site, this allows the D-WalkerB loop of *Ms*Rv1272c to stay in the relaxed "L" conformation and ADP is released from the distorted consensus site. Finally, the transporter returns to the **IF**^apo state. This completes the major cycle of transport. Possibly, there is also a minor cycle branch after step (6). The transporter in the ATP | ADP-bound **IF**^asym-2 state is able to load a new substrate (with help of the extending loop) and exchange ADP with a new ATP molecule at the consensus site. At the degenerate site, ATP does not have time to hydrolyze but still binds there. This allows the transporter to directly enter step (2) for a new cycle. The minor cycle may represent a highly effective mode of transport. To complete the full suite of structures to explain the mechanism requires additional intermediate states to be determined. Further evidence is needed to explain how step (7) occurs to induce the conversion from the **IF**^asym-2 state to the **IF**^asym-3 state (Fig. 6).

There are a large family of heterodimeric ABC exporters with a degenerate site, and these are typically found in higher eukaryotes. For example, about half of the human heterodimeric ABC exporters have a degenerate site[21]. Because of the existence of a degenerate site, the heterodimeric ABC transporters have a mechanism of transport that is

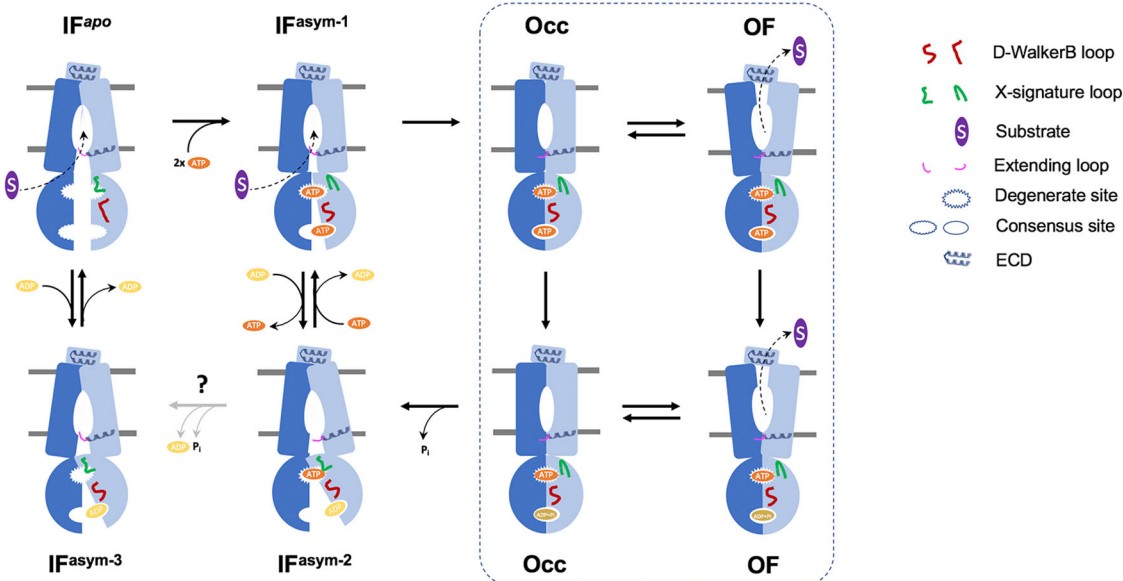

**Fig. 6 | A model for *Ms*Rv1273c/72c assisted transport.** Entry and release of substrate is proposed to require the extending loop and ECD of *Ms*Rv1272c. NBD partial dimerization is mediated by the degenerate site in the three types of intermediate asymmetric states. Conformational changes of NBD in *Ms*Rv1272c, especially the D-WalkerB loop and X-signature loop, are necessary for the function of both NBSs, including activation of the consensus site and release of nucleotide from the degenerate site. To validate how ATP is released to induce the conversion from **IF**^asym-2 to **IF**^asym-3 will require the determination of the structure of more intermediate states (marked with a question mark). *Ms*Rv1273c and *Ms*Rv1272c are shown in blue and light blue, respectively. The dashed box indicates states with NBDs fully closed.

different from the homodimeric counterparts. We suggest that the partial dimer of NBDs mediated by the degenerate site is indispensable for the transition between the **IF**$^{apo}$ state and Occ state in *Ms*Rv1273c/72c. Thus, the degenerate site is essential for the functioning of the transporter. The number of reported structures of ABC exporters with a degenerate site is limited, most of which are in the **IF**, **Occ**, and **OF** states. Though the conformational differences between these states have been analyzed, the detailed transition process has never been elucidated. This is due to a lack of information on the key intermediate steps, even though the conformational space has been delineated by the study of the TM287/288 transporter[25]. The findings from this study, including the intermediate states and the structural basis of degenerate-site mediated transport cycle improve our understanding of the mechanism for heterodimeric ABC transporters where a degenerate site is present.

Since Rv1273c/72c is functionally essential for lipid construction of the *Mtb* cell wall in being able to transport long-chain fatty acids[32,33], we hypothesize that the physiological substrate is a lipid precursor for the cell wall. In our **IF** state cryo-EM maps, *i.e.*, the **IF**$^{apo}$ state, we cannot accurately identify densities that are found in the central cavity (Supplementary Fig. 15a). They are surrounded by residues including Phe183, Gln238, Phe293 of *Ms*Rv1273c and Asn186, Gln194 of *Ms*Rv1272c (Supplementary Fig. 15a). These densities could belong to the substrate or the N-terminus of the extending loop. However, we favor substrate, since their position is similar to the substrate binding site reported in other ABC transporters such as BmrCD (PDB code: 7M33)[44], Atm3 (PDB code: 7N59)[51] and ABCB6 (PDB code: 7DNY)[52] (Supplementary Fig. 15b). Further studies will be required to definitively identify the physiological substrate of Rv1273c/72c.

Rv1273c/72c is a good target for new drugs to treat TB. This is because its role is critical in cell wall biosynthesis, critical to pathogen survival in the host cell and the immune response, and in drug resistance. In particular, we have shown that *Ms*Rv1273c/72c is a drug efflux pump for isoniazid. *Ms*Rv1273c/72c shares 71% overall sequence identity with *Mtb* Rv1273c/72c. Even though the localized environments could be different between the *Mtb* and *Msm* heterodimers, the overall folds, intermediate conformations and functional sites are expected to be conserved. In addition, we can generate models of *Mtb* Rv1273c/72c in different intermediate states and use these as templates for inhibitor design. Thus, our structures solved here represent accurate templates for anti-TB drug design. Though there are many ATP-binding transporters with different functions, they are also different in structures, conformations and mechanisms of action. Potentially, the unique structural features in Rv1273c/72c such as the extending loop and ECD could be blocked by designed inhibitors, not affecting other ATP-binding transporters. Alternatively, the conformational changes important for transport could be inhibited by targeting the D-WalkerB loop and X-signature loop of Rv1272c, especially in the distorted degenerate and consensus sites. The residues in the surrounding space of the ATP binding site are different from other ATP-binding transporters and could also be used for the development of specific inhibitors and drugs. Thus, our structures provide a solid framework for the development of anti-TB drugs.

## Methods

### Protein expression and purification

The cluster of *rv1273c-rv1272c* genes from *Mtb* H37Rv strain genome or the cluster of *msmeg_5008-msmeg_5009* genes from *Msm mc² 155* strain genome were cloned into the engineered pMV261 vector fused with a Flag-tag or a 6×His-tag attached to the C-terminus of Rv1272c or its *Msm* homolog, under the control of the acetamide promoter. The sequences of primers used for cloning were provided in Supplementary Data 1. The *Msm mc² 155* competent cells were transformed by electroporation of the resultant plasmid. The cells were cultivated at 37 °C in Luria-Bertani (LB) broth liquid media supplemented with 50 µg mL$^{-1}$ kanamycin, 20 µg mL$^{-1}$ carbenicillin, and 0.1% (w/v) Tween80 until the optical density at 600 nm reached 1.0. Over-expression of the recombinant protein was induced by 0.2% (w/v) acetamide at 16 °C. After four days, cells were harvested by centrifugation at 4000 × *g* for 20 min. Cell pellets were resuspended in Buffer A containing 50 mM Tris (pH 8.0), 500 mM NaCl and then lysed by passing through a high-pressure homogenizer at 1100 bar. Cell debris was removed by centrifugation at 20,000 × *g* for 10 min at 4 °C. The supernatant was collected and ultra-centrifuged at 150,000 × *g* for 1.5 h. The membrane fraction was solubilized in Buffer A with the addition of 1% (w/v) N-dodecyl-b-D-maltoside (DDM; Anatrace) at 4 °C for 1.5 h. The suspension was ultra-centrifuged and the supernatant was supplemented with 10 mM imidazole and then loaded onto a nickel-nitrilotriacetic acid (Ni-NTA) agarose beads (Qiagen) affinity column. The beads were rinsed with Buffer A supplemented with 0.02% (w/v) DDM, and 30 mM imidazole, and the recombinant protein complex was eluted from the beads with Buffer A supplemented with 0.06% (w/v) Digitonin (Biosynth Carbosynth) and 500 mM imidazole. The eluted sample was concentrated and applied to a size exclusion chromatography column (Superose 6 Increase, GE Healthcare) pre-equilibrated with Buffer A supplemented with 5 mM DTT, and 0.06% (w/v) Digitonin. Finally, the main peak fractions were pooled and concentrated to 10 mg mL$^{-1}$ for further studies or stored at −80 °C.

All mutants of *Ms*Rv1273c/72c were generated by the TaKaRa MutanBEST Kit using the DNA sequence of the wild-type protein as the template. The sequences of primers used for cloning were provided in Supplementary Data 1. Expression plasmids were prepared in the same way as for the wild-type transporter. Mutated proteins were expressed and purified following the same protocol as the wild-type protein except that DDM was used during the entire purification procedure for the *Ms*Rv1273c/72c$^{E553Q}$ mutant.

To remove detergent and prepare protein reconstructed in peptidisc, lyophilized NSPr (Nter-FAEKFKEAVKDYFAKFWDPAAEKLK-EAVKDYFAKLWD-Cter) (Genscript, purity >98%) was solubilized in Buffer A at room temperature to a final concentration of 1 mg mL$^{-1}$ (Buffer B) and kept on ice for use[47]. After rinsing the Ni-NTA beads with Buffer A containing 0.02% (w/v) DDM and 30 mM imidazole, 10 mL Buffer B was added to the beads and incubated for 10 min on ice. The recombinant protein complex was eluted from the beads with Buffer B supplemented with 500 mM imidazole. The eluted sample was concentrated and applied to a Superose 6 column pre-equilibrated with Buffer B supplemented with 5 mM DTT. Finally, the main peak fractions were pooled and concentrated to 2 mg mL$^{-1}$ for further studies or stored at −80 °C.

### Cryo-EM grid preparation and data collection

For cryo-EM grid preparation, aliquots (3 µL) of fresh protein samples at 10 mg mL$^{-1}$ (2 mg mL$^{-1}$ for samples in the peptidisc) after purification were immediately applied to H$_2$/O$_2$ glow-discharged holey carbon grids (Quantifoil Cu R1.2/1.3). Grids were blotted for 3.0 s and flash-frozen in liquid ethane cooled by liquid nitrogen using an FEI Vitrobot Mark IV (Thermo Fisher) with the environmental chamber set to 100% humidity and 8 °C. For the AMPPNP-bound **IF**$^{asym-1}$ state, the protein was incubated with 5 mM AMPPNP and 4 mM MgCl$_2$ at 4 °C for 30 min before grid preparation. For the ATP-bound **Occ** state, the *Ms*Rv1273c/72c$^{E553Q}$ mutant was incubated with 4 mM ATP and 4 mM MgCl$_2$ at 4 °C for 30 min before grid preparation. For the ATP|ADP-bound **IF**$^{asym-2}$ state, the protein was incubated with 10 mM ATP and 4 mM MgCl$_2$ at 4 °C for 30 min before grid preparation. For the ADP-bound **IF**$^{asym-3}$ state and **IF**$^{asym-3}$ (peptidisc) state, the protein was incubated with 10 mM ATP and 4 mM MgCl$_2$ at 37 °C for 5 min before grid preparation. Alternatively, 10 mM ADP and 4 mM MgCl$_2$ were incubated with the protein at 4 °C for 30 min before grid preparation. For the ATP|ADP+Vi-bound **Occ (Vi)** state, 10 mM ATP, 4 mM MgCl$_2$ and 20 mM Na$_3$VO$_4$ were incubated with the protein at 37 °C for 5 min before grid preparation.

Cryo-EM data were collected on a FEI Titan Krios electron microscope operated at 300 keV with a Gatan K3 camera at a nominal magnification of ×105,000 in super-resolution mode and binned to a pixel size of 0.832 Å (or 0.96 Å). Automated single-particle data acquisition was performed with the automated data collection program Serial EM[53]. Each movie was recorded in 40 frames with a total exposure time of 6.4 s and a total exposure dose of 60 e⁻/Å². The nominal defocus range was set to −1.8 to −1.2 μm. The sample preparation conditions and data collection parameters for all the structures are provided in Supplementary Table 1.

## Cryo-EM image processing

All dose-fractionated image stacks were motion-corrected and dose-weighted using MotionCorr2 software[54]. CTF estimation was performed using the "Patch CTF Estimation" program in cryoSPARC (v4.0.1)[55]. For the dataset of $IF^{apo}$ state, 4863 micrographs were selected after those images that exhibited defects in the Thon rings due to excessive drift, ice contamination, or astigmatism were discarded. 3,566,457 particles were automatically extracted using a box size of 320 pixels and subjected to several rounds of reference-free 2D classification to discard bad particles, yielding a stack of 378,470 particles. They were used for Ab-Initio reconstruction to generate 3D models as references to perform heterogeneous refinement. The particles belonging to the correct initial map were subjected to several rounds of heterogeneous refinement. After that, 139,918 particles belonging to the map with best resolution were then refined using non-uniform (NU) refinement to generate the final cryo-EM map with an estimated average resolution of 3.1 Å according to the gold-standard Fourier shell correlation cutoff of 0.143[56]. No mask was used for all the map refinement processes. All the other datasets were processed in the same pipeline. Local resolution ranges were also analyzed using cryoSPARC for each map.

## Model building and refinement

The AlphaFold2 predicted models of *Ms*Rv1273c (AF-A0R271-F1) and *Ms*Rv1272c (AF-A0R272-F1) were used as the initial models, and they were fitted as rigid bodies into the cryo-EM map of *Ms*Rv1273c/72c complex in the $IF^{apo}$ state and adjusted manually using Coot[57]. The structure was refined in real space using PHENIX[58] with secondary structure and geometry restraints to prevent over-fitting. The final atomic model was evaluated using MolProbity[59]. The structure in the $IF^{apo}$ state was further used as the initial model for other states. The processes of model building and refinement for these states are similar to the $IF^{apo}$ state. ATP, ADP, AMPPNP, $Mg^{2+}$ and Vi were fitted into the corresponding cryo-EM map according to the additional non-protein density. The refinement statistics of all final models are shown in Supplementary Table 1. The composition of all the models is summarized in Supplementary Table 2. All the figures were generated using PyMOL (The PyMOL molecular graphics system, Schrödinger, LLC.) or UCSF Chimera (http://www.rbvi.ucsf.edu/chimera).

## Preparation of proteoliposomes

Proteoliposomes were prepared according to a standard protocol described previously[60]. Briefly, 20 mg of *E. coli* polar lipids containing 67% (w/w) PE, 23.2% PG and 9.9% CA (Avanti) were dissolved sequentially in chloroform and ether and dried with a rotary evaporator at 37 °C to produce a thin lipid film. The film was suspended with 1 mL buffer containing 50 mM HEPES (pH 7.0), 150 mM NaCl and then sonicated. Next, the lipid suspension was frozen and thawed three times. Subsequently, the lipids were extruded through a 0.2 μm polycarbonate filter using a mini-extruder (Avanti). This step was performed 11 times. The liposomes produced were destabilized by 15 additions of 10 μL aliquots of 10% (w/v) Triton X-100 so that the solution becomes optically transparent. After that, the protein was mixed with the liposomes at a ratio of 1:100 (w/w) at 4 °C. It was then diluted

to 4 mL with Buffer A (50 mM Tris (pH 8.0), 500 mM NaCl). 80 mg of Bio-Beads SM-2 was added to the mixture and incubated overnight to remove the detergent. Proteoliposomes were collected by centrifugation at 230,000 × g for 30 min. Then, they were suspended in Buffer A and the lipid concentration was adjusted to 20 mg mL⁻¹ for further studies or stored at −80 °C. The integrity of vesicles was checked by negative stain EM to guarantee that all the proteoliposomes were integral and no protein aggregation was formed (Supplementary Fig. 1f). The efficacy of reconstitution was evaluated by SDS-PAGE showing about 97% of the added protein became incorporated into the proteoliposomes (Supplementary Fig. 1e). This ratio was used to adjust the amount of protein in the ATPase and transport assays. The concentration of the liposome-reconstituted transporter was further adjusted by considering the 50-50% distribution of two configurations (outside-out and inside-out) in the proteoliposomes[61]. Only the activity of NBDs in the inside-out configuration can be measured.

## ATPase activity assay

The ATPase activity assay was performed as described previously[41]. The released $P_i$ from ATP hydrolysis reacts with malachite green reagent to form a stable dark green complex, whose presence is detected by measuring absorbance at 620 nm. The intensity of the color is directly proportional to the amount of $P_i$ generated, and thus, to the ATPase activity in the sample. For the assay, 4.68 μg of purified protein complex or the mutants was incubated in a 20 μL reaction volume containing 50 mM Tris (pH 8.0), 500 mM NaCl, 1 mM ATP and 4 mM $MgCl_2$ for 10 min at 37 °C. The reaction was stopped by mixing 100 μL aliquots with activated malachite green ammonium molybdate for 2 min. Samples were subsequently incubated at room temperature for 30 min with 24% (w/v) sodium citrate after which the absorbance at 620 nm was measured using a SpectraMax iD3 multifunction reader (Molecular Devices). Samples without protein were used as the negative controls and subtracted as background. ATPase activity is represented as the amount of $P_i$ produced by 1 mg of protein per minute. All experiments were performed in triplicate.

## Liposome-based transport assay

Wild type protein complex or the mutant was inserted into proteoliposomes as described above. Liposomes without inserted proteins were used as the negative control. 20 mg mL⁻¹ lipid concentration of proteoliposomes or control liposomes, 50 μM of isoniazid or ethambutol, 5 mM ATP and 4 mM $MgCl_2$ were mixed to prepare a reaction system with a total volume of 100 μL. The mixture was incubated at 37 °C for 30 min. Liposomes were then collected by centrifugation at 230,000 × g for 30 min. The supernatant was discarded, and the pellet was resuspended in Buffer A, followed by another centrifugation at 230,000 × g for 30 min. This washing step was repeated three times. The pellet was resuspended in 200 μL Buffer A mixed with 200 μL ethyl acetate. The mixture was vortexed and then allowed to stand for 3 min. The upper layer was aspirated and evaporated to dryness under a stream of warm air. The residue was reconstituted in 200 μL methyl cyanide and filtered through a 0.22 μm filter for mass spectrometry. 0.1 μg mL⁻¹ of either isoniazid (0.73 μM) or ethambutol (0.36 μM) were separately tested by mass spectrometry, which was used as the standard reference.

To quantify the amount of isoniazid transported by proteoliposomes, isoniazid standards were dissolved in methanol at concentrations of 0.02, 0.2, 1, 10, and 50 μg mL⁻¹. The calibration curve was generated using Liquid Chromatography-Tandem Mass Spectrometry (LC-MS/MS). Next, 20 mg mL⁻¹ lipid concentration of proteoliposomes, 5 or 50 or 500 μg mL⁻¹ isoniazid, 5 mM ATP and 4 mM $MgCl_2$ were mixed to prepare a reaction system with a total volume of 100 μL. The mixture was incubated at 37 °C for 30 min. After that, the proteoliposomes were washed three times and the pellet was suspended

in 100 μL of buffer A. Next, 900 μL of methanol was added to disrupt the proteoliposomes. The supernatant was collected by centrifugation at $13,000 \times g$ for 10 min before being passed through a 0.22 μm filter to prepare the samples for LC-MS/MS analysis. The transport rate is the amount of isoniazid transported by 1 mg of protein per minute. All experiments were performed in triplicate.

## Mass spectrometry
Mass spectrometry (MS) analysis was performed using an Agilent 6230 mass spectrometer (Agilent Technologies, Santa Clara, CA, USA) equipped with an electrospray ionization (ESI) source. The sample injection volume was 3 μL. The mobile phase consisted of 0.1% methanol in water (A) and 0.1% formic acid in methanol (B), mixed at a ratio of 30% A and 70% B. The flow rate was set to 0.2 mL min$^{-1}$, with a total analysis time of 2 minutes. The mass spectrometer was operated in positive ion mode, with a capillary voltage of 3.0 kV, a drying gas flow rate of 11 L min$^{-1}$, and a drying gas temperature of 350 °C. The nebulizer pressure was set to 40 psi. The fragmentor voltage was set at 140 V. Data were acquired in the mass range of 100–1000 m/z. All data acquisition and processing were performed using Agilent MassHunter software (Agilent Technologies).

## Liquid chromatography-tandem mass spectrometry
LC-MS/MS was performed using a Shimadzu 30 A HPLC system equipped with an autosampler, binary pump, and column oven, coupled with an AB 4600 mass spectrometer (AB SCIEX, Framingham, MA, USA) equipped with a Turbo Ion Spray source. The mobile phases consisted of 0.1% formic acid in distilled water (A) and acetonitrile containing 0.1% formic acid (B) run at a flow rate of 0.4 mL min$^{-1}$. Chromatography was performed using an ACE C4 column (5 μm, $100 \times 2.1$ mm) with a gradient elution. The column temperature was set at 75 °C. The gradient elution condition was set as follows: t = 0 min, A = 100%, B = 0%; t = 2.5 min, A = 100%, B = 0%; t = 5 min, A = 5%, B = 95%; t = 6.5 min, A = 5%, B = 95%; t = 6.6 min, A = 100%, B = 0%; t = 8.0 min, A = 100%, B = 0%. The Electrospray Ionization was set in positive ion mode: the voltage value was 5500 V, gas 1 and gas 2 were set at 50 psi, and the curtain gas at 35 psi. The scan range was set from 80 to 550 Da. The collision energy was set to 30. The mass transition of isoniazid was 138 (m/z) to 121 (m/z)[62]. In the first-stage, mass spectrometry (MS1) was used to obtain precursor ions; and in the second stage, the precursor ion with m/z 138 was further analyzed by mass spectrometry (MS2). Data were collected to determine the quantity of the substance with m/z 121 by MS2. All data acquisition and processing were performed using Analyst and PeakView (SCIEX). The raw spectrum for LC-MS/MS were provided in Supplementary Data 2.

## Growth-complementation assay
The *Msm* wild type strain (WT-*Msm*), *Ms*Rv1273c/72c knockout strain (Δ*Ms*Rv1273c/72c) and the complemented strain containing pMV261-*Ms*Rv1273c/72c (Δ+*Ms*Rv1273c/72c) were separately plated on Luria Agar (LA) supplemented with 100 μg mL$^{-1}$ carbenicillin. Single colonies were picked and cultured in LB liquid medium containing 100 μg mL$^{-1}$ carbenicillin until the optical density at 600 nm (OD$_{600}$) reached 0.6–0.8. The cultures were then diluted to an OD$_{600}$ of 0.01 in 100 mL of LB medium supplemented with 0.1% (w/v) Tween80. For the experimental groups, 0.15 μg mL$^{-1}$ ethambutol or 0.05 μg mL$^{-1}$ isoniazid was added into the medium and cultured in a 37 °C shaker. Cell growth was determined by measurement of OD$_{600}$ in triplicate.

## Molecular dynamics (MD) simulation
To simulate the motion progress from $\text{IF}^{asym-2}$ to $\text{IF}^{asym-3}$, the starting model was constructed using the $\text{IF}^{asym-2}$ structure with ATP removed from the degenerate site. The system was built with the CHARMM-GUI webserver[63] using CHARMM36m[64] force field and TIP3P water model. Protein was embedded into the lipid bilayer consisting of 50% 1-palmitoyl-2-oleoyl-sn-glycero-3-phosphocholine (POPC) and 50% 1-palmitoyl-2-oleoyl-sn-glycero-3-phosphoethanolamine (POPE). The upper layer has 136 POPC molecules and 136 POPE molecules while the lower layer has 131 POPC molecules and 131 POPE molecules. The structure described above was solvated in a cubic water box containing 0.5 M NaCl. The simulation box dimensions are about 136 Å × 136 Å × 197 Å with 348,028 atoms, which include 86,313 water molecules. Energy minimization, NVT and NPT equilibration were performed in six steps, then, 100 ns production was performed by GROMACS 2024.1[65] at 310.15 K for three independent runs, with trajectories saved at 10 ps intervals, and with default parameters provided by CHARMM-GUI webserver. The distance was calculated between the mass center of all the Cα atoms in partial NBD of *Ms*Rv1273c (residues G418 to A519) and the mass center of all the Cα atoms in partial NBD of *Ms*Rv1272c (residues S380 to D473 and residues P546 to G628). Plots of energy minimization, NVT and NPT equilibration (Supplementary Fig. 17) were performed using qtgrace (https://sourceforge.net/projects/qtgrace/). Visualization was performed using R-4.3.2 (https://www.r-project.org/) and PyMOL. The initial and final configurations as well as an intermediate configuration (t = 36 ns) similar to $\text{IF}^{asym-3}$ by one MD simulation were provided as Supplementary Data 3, 4, 5.

## Reporting summary
Further information on research design is available in the Nature Portfolio Reporting Summary linked to this article.

## Data availability
All data needed to evaluate the conclusions are present in the main text or Supplementary Materials, and available from the Corresponding Author upon request. The cryo-EM maps of *Ms*Rv1273c/72c have been deposited in the Electron Microscopy Data Bank (EMDB) under accession codes EMD-37450 ($\text{IF}^{apo}$ state); EMD-37451 (AMPPNP-bound $\text{IF}^{asym-1}$ state); EMD-38626 (ATP | ADP-bound $\text{IF}^{asym-2}$ state); EMD-38627 (ADP-bound $\text{IF}^{asym-3}$ state (ATP 37 °C treated)); EMD-38628 (ADP-bound $\text{IF}^{asym-3}$ state (ADP 4 °C treated)); EMD-62611 (ATP-bound **Occ** state); EMD-60789 (ADP-bound $\text{IF}^{asym-3}$ **(peptidisc)** state (ATP 37 °C treated)); EMD-60790 (ADP-bound $\text{IF}^{asym-3}$ **(peptidisc)** state (ADP 4 °C treated)); and EMD-60791 [https://www.ebi.ac.uk/pdbe/entry/emdb/EMD-37491] (ATP | ADP+Vi-bound **Occ (Vi)** state). The atomic coordinates of *Ms*Rv1273c/72c have been deposited in the Protein Data Bank (PDB) under accession codes 8WCW ($\text{IF}^{apo}$ state); 8WCX (AMPPNP-bound $\text{IF}^{asym-1}$ state); 8XSR (ATP | ADP-bound $\text{IF}^{asym-2}$ state); 8XSS (ADP-bound $\text{IF}^{asym-3}$ state (ATP 37 °C treated)); 8XST (ADP-bound $\text{IF}^{asym-3}$ state (ADP 4 °C treated)); 9KWI (ATP-bound **Occ** state); 9IQE (ADP-bound $\text{IF}^{asym-3}$ **(peptidisc)** state (ATP 37 °C treated)); 9IQF (ADP-bound $\text{IF}^{asym-3}$ **(peptidisc)** state (ADP 4 °C treated)); and 9IQG (ATP | ADP+Vi-bound **Occ (Vi)** state). The input files of MD simulation are available on Github [https://github.com/qifeng0000001/Rv1272_73MD]. The source data underlying Figs. 1a, c, f-g, and Supplementary Fig. 1a-e, 1h are provided as a Source Data file. Source data are provided with this paper.

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

## Acknowledgements

We thank the staff from the Bio-EM Facility of ShanghaiTech University, for assistance during cryo-EM data collection. We also thank Dr. Jiakang Chen at the Analytical Chemistry Platform at Shanghai Institute for Advanced Immunochemical Studies, ShanghaiTech University for assistance with mass spectrometry analysis. This work was supported by grants from National Key R&D Program of China (grant No. 2021YFA1300900 to J.L.; 2022YFC2302900 to H.Y.), National Natural Science Foundation of China (grant No. 32394010 to Z.R.), Major Project of Guangzhou National Laboratory (grant No. GZNL2024A01024 to J.L.), Shanghai Municipal Commission of Science and Technology (grant No. 22ZR1441600 to J.L.), Shenzhen Clinical Research Center for Tuberculosis (grant No. 20210617141509001 to Y.Z.) and the Shanghai Frontiers Science Center for Biomacromolecules and Precision Medicine, ShanghaiTech University.

## Author contributions

J.L., Z.R., H.Y. supervised the study; J.Y., Y.L. purified the proteins; J.Y., J.L., Y.L., T.H., Y.G. collected and processed cryo-EM data and built the structure model; Y.L., J.Y., Z.C., J.P., Y.S., L.Y. performed biochemical and MD simulation experiments; J.L., J.Y., Y.L., C.Z., Z.C., X.Y., S.L., H.Y. and Z.R. analyzed the structure and discussed the results; J.L., J.Y., Y.L., Y.G., Y.Z., X.C., L.G., H.Y. and Z.R. prepared and wrote the manuscript.

## Competing interests

The authors declare no competing interests.
