## [Transparent Peer Review file · Nature Communications]

Structure and mechanism of a mycobacterial isoniazid efflux pump MsRv1273c/72c with a degenerate nucleotide-binding site

Corresponding Author: Professor Jun Li

Version 0:

Reviewer comments:

Reviewer #1

(Remarks to the Author)

Yu et al. present cryo-EM structures of the mycobacterial ABC transporter MsRv1272c/Rv1273c, a homolog of *Mycobacterium tuberculosis* (Mtb) Rv1272c/Rv1273c. Resistance towards the currently available drugs to treat tuberculosis (TB) has increased the urgency to identify new drug targets and drugs to treat TB. The authors suggest the heterodimeric ATP-binding cassette transporter Rv1272c/Rv1273c as a potential target, however prior to this study there was no information available about the structure and mechanism of this transporter. In this study, the authors aimed to capture the transporter in several different states. Interestingly, this is a heterodimeric transporter that has only one functional ATP-binding site. The authors present four nice high-resolution cryo-EM structures with well-resolved TMDs, NBDs and densities for the bound nucleotides. The authors show three structures in the inward-facing (IF) conformation – an apo state, an AMPPNP-bound structure that represents a state prior to ATP hydrolysis, given that AMPPNP cannot be hydrolyzed and an ADP-bound structure that represents a state after ATP hydrolysis. They also resolved a structure in the occluded state by introducing an EtoQ mutation in the consensus ATP-binding site.

While this seems to be a very interesting transporter to study, overall, the interpretation of the presented data is speculative and the results are incomplete and sometimes questionable. Furthermore, there is no functional data that supports the claims that the authors make. The study has potential, but a more comprehensive structural and functional analysis is needed.

My comments below are ordered by appearance in the text, not necessarily by importance.

Abstract

In 33-35: Speculative statement as there is no structural or functional data presented to support this.

Introduction

- The introduction does not make it clear what the focus of the study is and does not provide compelling evidence for why this study is necessary. There is also not enough information on Rv1272c/Rv1273c. For example, what is known about the topology of this heterodimer? What kind of functional analysis has been reported in the literature? Also, since it is suggested that this transporter could be a potential drug target, it might be helpful to include some information on the current drugs used for TB treatment and what their targets are.

- In 54 and In 59: I would suggest more appropriate references are used instead of ref.1 and 2. For example, for ref.1, recent WHO data will be a better citation than a review.

- In 65-67: This should be rephrased to accurately summarize what was written in ref.9, which is that the conformational changes induced by binding and hydrolysis of ATP result in conformational changes of the TMDs that are likely transmitted through the coupling helices.

- In 70-94: This section needs to be rewritten to properly summarize and cite the papers referenced.

- In 90: the manuscript states that Rv1272c/Rv1273c is predicted to form a heterodimer. The reference cited (ref. 20) is a review and therefore, not a direct reference. Furthermore, Rv1272c has been hypothesized to function as a homodimer (PMID: 29360453). Can the authors comment on that? Perhaps this should also be included/discussed in the introduction.
- In 95: Unclear what “this transporter” refers to. Is this in reference to the potential heterodimer?
- In 96-98: What kind of assays have been performed to show the multi-drug resistance role of the transporter? The manuscript states that this transporter has been “unequivocally identified as a multidrug transporter,” however I could not find support for this statement in the review cited (ref.20). My understanding is that this hypothesis is based on sequence analysis, but if that is not the case, more references and clear description should be included. Also, the references for the drugs listed discuss Rv1273c only.
- In 99: Ref. 26 does not include the information cited. Perhaps the wrong reference was listed and maybe the authors meant to refer to PMID: 29050768, instead. Additionally, it is unclear why the SNP is mentioned. Where is this mutation? How is it relevant for this study?
- The introduction does not make it clear why the authors chose the homolog Msm. Additionally, there should be some information about what inward-facing, occluded and outward facing states are.
- In 107: The authors claim that they present two turnover states. However, none of the samples prepared meet the criteria for turnover conditions. Turnover conditions imply that all ingredients needed for a transporter to go through its full transport cycle are present, including a substrate.
- In 108-109: As an extension of the above comment, the manuscript speculates on a mechanism; however, the results presented do not support the proposed mechanism.

Results

- The results section is mixed with discussion, although there is also a separate discussion section. Both sections should be rewritten for clarity and to eliminate repetition. Furthermore, each structure should be discussed separately, which will also improve the flow. Note that it is also not clear from the purification subsection of the Results that these transporters were purified in detergent.
- In 117: Choice of nomenclature is a bit confusing since it is not immediately clear that MsRv1273c-Rv1272c implies that both half-transporters are from Msm.
- In 120: What do the authors mean by “functional ATPase”? It is important to note that ATPase activity does not necessarily mean a functional transporter.
- ATPase assays: It is unclear how the ATPase activity assays were performed. Was this done in detergent or proteoliposomes? Ideally data should be shown for both. Have the authors performed these assays in the presence of vanadate? Does vanadate inhibit the transporter? Statistics were not reported and it is unclear what statistical analysis was performed.
- In 123: For clarity, it would be good to specify that AMPPNP is an analog of ATP that cannot be hydrolyzed.
- In 126-128: The observation that there is ADP but no vanadate at the consensus site and no ATP at the degenerate site goes against what is expected when ATP and vanadate are added to the sample, given that inhibition by vanadate is well-characterized in the literature. Is it possible that perhaps ADP was used by mistake instead of ATP for sample preparation? This would also explain the absence of the nucleotide at the degenerate site.
- In 129: The references to the figures should be moved to the appropriate location.
- In 143: This is the first mention of “gating loop” in the text, other than the abstract. Does such loop exist in other transporters? Why is it called “gating” loop? The authors may consider including something about this.
- In 144: Fig. 1B. does not clearly show what is stated in the text.
- In 150 and In 154: Contradiction in the statements. On In 150 it is suggested that residues 1-16 are resolved, while In 154 states that residues 1-9 are flexible.
- In 155-157: The interpretation of what is written in the referenced manuscript (Hohl et al) is incorrect. The original manuscript does not suggest that the “loop restricts the conformation of the TM helices”. Hohl et al state that:

“The N terminus of TM288 is 25 residues longer than the one of TM287 and is partially defined in the crystal structure starting from residue 10 (Supplementary Fig. 7). It folds into an extended N-terminal elbow helix and interacts as a coiled structure with ICL4 of the membrane domain of TM288. This interaction appears to restrict the movement of ICL4 (and of NBD1 connected to it) with respect to the other transmembrane helices of TM288 and might therefore further strengthen the interaction between the NBDs.”

- In 160: Could the authors clarify in the text the structure of which mutant they solved? What was the consideration behind the number of residues chosen to delete for this mutant? The Methods section should include information about these different mutants. It is unclear from the manuscript why the authors thought that deleting the first 16 residues of the gating loop would influence the overall structure of the transporter and what they expected to see, given that they already specify that the first several residues are disordered. Furthermore, why did the authors expect that these deletions would have an effect on the ATPase activity?
- In 166: How can the authors deduce that the “gating loop” does not modulate transport without having performed any transport assays?
- In 168-170: The hypothesis that the gating loop might help the substrate enter the cavity is speculative and not supported by the data presented here. It might be more appropriate to make this suggestion in the conclusion as future experiment, rather than use it as an absolute statement. The caption of Fig. 4 should also be edited accordingly.
- In 173: “since the substrate(s) of MsRv1273c-Rv1272c are yet to be identified” – the authors implied in the introduction (In 94-97) that long-chain fatty acids and three drugs are substrates of the transporter. The two statements are contradictory.
- In 185-187: This is another speculative statement not supported by data. Furthermore, it raises the question of how exactly where these “predicted” OF models generated? This is not described anywhere in the Methods.
- In 214: Why did the authors choose the D497N mutations and what did they hypothesize they will see?
- In 257-259: Why do the authors believe that ATP can be “squeezed out” during the cryo-EM sample preparation but AMPPNP cannot? Cryo-EM sample preparation will not cause a nucleotide to be “squeezed out”, however, if we were to assume that this were to happen, then why would the authors consider this structure in their further analysis of the mechanism of the transporter if they believe it is an artifact of sample preparation?
- In 267-270: It is unclear what the authors mean here and why MsRv1273c-Rv1272c is compared to a substrate-bound CFTR.
- In 270-272: This statement is also unclear. What do the authors mean by “Although ATP at the consensus site contacts both NBDs, this state does not exist physiologically”? Why do the authors think this is the case?
- In 276-283: Again, please note that none of the structures were determined under turnover conditions.
- In 308-309 and In 316. The manuscript alludes to weak ATP binding at the consensus site. It is unclear how this conclusion was drawn based on the apo structure.
- In 357: reference should be included for the alternating-access transport model

Discussion

- In 390-391: unclear what “more variable in structure and conformation” means
- In 402-403: This is a very vague statement. Can the authors elaborate what they mean by “specific mechanism of transport” for heterodimeric transporters with degenerate site? Could the authors provide references for this?

Methods

- This section should be expanded to include information about all samples generated, including the different mutants studied. There should also be adequate information about the composition of each sample used for cryo-EM studies. For example, for the AMPPNP-bound and ADP-bound samples, there is no mention of magnesium being added. The cryo-EM grid preparation should also include information about the concentration of each sample. Was this fresh sample?
- The cryo-EM image processing section should be expanded to include enough details and information so that the reader can get a comprehensive understanding of how the data was processed and how decisions were made at each stage of processing. Ideally, this should be done for each sample, although it may seem repetitive. There should also be description and workflow for the low-res mutant sample.
- Which cryoSPARC version was used? Which CTF estimation routine was chosen?
- Was there any postprocessing done - for example, per particle motion correction and ctf refinement? Were masks used?
- The model building and refinement section also needs to be expanded to include more details. Which map exactly was used for the building of the initial model? Was the backbone built by hand, or was a backbone tracing software used or another starting model used (ex. AlphaFold)? What about unresolved regions in the maps? This would be a good place to point out segments for each map that were unresolved and therefore not built.

Reviewer #2

(Remarks to the Author)

The Authors conducted structural studies on a heterodimeric ABC exporter, Rv1273-Rv1272 found in *Mycobacterium smegmantis* applying different conditions to gain mechanistic insights into the transport cycle of the membrane protein. As both Rv1272c and Rv1273c play a crucial role in the physiology of the pathogen, the medical relevance is inevitably high. In the current study, the Authors present four structures of the transporter; an inward-facing apo structure, two inward-facing structures with AMPPNP or ADP, and one nucleotide-occluded structure harboring the E-Q nonhydrolytic mutation. Based on careful analysis of the structural models, the Authors conclude a possible transport cycle for the heterodimer consisting of 6 states with two major transport cycles. While the work bears high significance to the field of ABC transporters, there are several concerns, which should be addressed before publication. Most importantly, I especially lack biochemical and functional experiments, which could support/modify the Authors conclusion(s). Such data would be vital to discuss the transport mechanism in greater depth.

Main comments:

- 1) I am puzzled by the structure of IF(turnover-1) with AMPPNP. The Authors state that the reason for having a tight binding at the degenerate site and a loose connection at the consensus site with AMPPNP is that the degenerate site binds the nucleotide first, acting as a scaffold to promote the dimerization at the other site. As a non-hydrolysable/poorly hydrolysable ATP analog, AMPPNP often locks ABC transporters in the occluded / outward-facing states and acts as a 'non-hydrolytic' mutation, by the disruption and simplification of the transport cycle into a reversible IF-OF transition. As the microenvironments of the two composite ATP binding sites are immensely different, another possible explanation for the IF(turnover-1) state is that AMPPNP cannot bind at the consensus site as strongly as it fits into the degenerate site. Is there any evidence that the binding of AMPPNP at the consensus site of Ms1273c/1272c is similar to that of ATP? It is also quite surprising that the stability of such a 'transition state' between the IF and OF conformations is high enough to capture it by cryoelectron microscopy. Techniques, such as single-molecule FRET could be suitable to lead to a conclusion about the extent of NBD dimer formation/dissociation at the composite sites, separately.
 - 2) The E-Q nonhydrolytic mutation abruptly ATP hydrolysis, promoting a tight NBD dimer and restricting the conformation of the TMDs into an outward-facing conformation for most ABC proteins. The Authors state that the reason for not observing the OF state is its instability. However, it may need additional discussion, since it is quite surprising that none of the prehydrolytic conditions (AMPPNP / E-Q) lead to an OF conformation, as all the reported structures are in the IF conformation. Did the Authors attempt to stabilize the OF conformation in other ways? Identification of the substrate might help in this issue and would strengthen the physiology and translational science of the transporter as well. Given the recent literature around Rv1273c and Rv1272c, the development of a transporter assay could be useful to investigate the substrate (e.g. long chain fatty acids) of the heterodimeric ABC transporter.
 - 3) As written above, the Authors state that ATP binds to the degenerate site first, and acts as a glue, followed by the binding of the second ATP molecule at the consensus site (if the reason for the loose consensus site is indeed the mechanism the Authors suggest). However, regarding the IF(turnover-2) structure, the Author states that ATP is squeezed out from the degenerate site as the NBDs rotate to each other. This would mean that ADP, besides destabilizing the dimer leading to the dissociation, also causes somehow the loss of the ATP at the degenerate site (through a conformational change?), which is quite unlikely in my opinion. Thus there might be a contradiction between IF(turnover-1) and (turnover-2). Indeed, for other ABC proteins with a degenerate site (e.g. CFTR), ATP remains bound for several transport cycle. How can the Authors explain the controversy between IF(turnover-1) and IF(Turnover-2)?
 - 4) Is there any reason (other than the better purity/higher yield) that the Authors characterized the rather non-pathogenic Ms transporter and not the pathogenic Mtb heterodimer? While the SEC chromatogram of Ms Rv1273c/Rv1272c looks evidently better, the ~70% sequence identity can often mean radically different microenvironments, which can interfere with the translational prospects toward rational drug design. Even a single difference in the amino acid sequence can drastically decrease the efficacy of the effector molecule, not to mention the desired specificity toward other ABC proteins. From this point of view, the relevance of the Mtb transporter structure and transport cycle would be higher, further increasing the medical importance of the present study. Did the Authors try to investigate the structure of the Mtb heterodimer as well?
- ### Minor comments and questions
- 5) The Authors might want to introduce the implication of AMPPNP and vanadate trapping in the introduction/results sections of the manuscript for the broader readership. It could help in understanding the conclusions the Authors made.
 - 6) The methods section for the grid preparation is not well-defined. Were Mg²⁺ present in all cases? It is reported added at the grid preparation of the E-Q-occluded structure, but it is certainly added for the If(turnover-2) state, as Mg is necessary for hydrolysis, and for IF(turnover-1) state, as Mg²⁺ is seen in the structure. Incubation time and temperature were not reported in all cases.
 - 7) Standard deviation describes the distribution, and standard error of the mean (S.E.M.) describes how accurate the determined mean is. Therefore, it is more fortunate to display S.E.M., rather than std. deviation in such cases.
 - 8) The Authors do not provide p values for the ATPase assay results. Are the determined values (especially for the E-Q vs the E-Q and D-N cases) significantly different?

9) Does Rv1273c and/or Rv1272c form homodimers?

10) In Figure 4, the only truly irreversible step is the hydrolysis of ATP, therefore, the Authors should indicate reversibility in some cases.

11) The authors might want to include a deeper explanation of the important residues in ATP hydrolysis at the NBD interface (i.e. Asp552 or the role of the switch histidine). An additional schematic figure could help the reader to understand the role of important residues, also subject of this study.

Version 1:

Reviewer comments:

Reviewer #1

(Remarks to the Author)

The new version of the manuscript provides in great detail the structural features of MsRv1273c/72c. It is clear that the authors have spent considerable time revising their manuscript and also acquiring new structural data. However, there are still many concerns that remain:

-It is still unclear why this particular transporter is a good target for inhibitor design against TB. It is also unclear why in general an ATP-binding transporter (and in particular, the ATP binding site) would be good target for inhibitors, when there are so many molecular machines that use ATP and would be affected by such an inhibitor.

-The authors discuss designing drugs specific to the degenerate site of the transporter from *Mycobacterium tuberculosis*. They also discuss the intricacies of the site and how small changes can affect it. However, for their research, they used analogs from *Mycobacterium smegmatis* and they show in Fig. 2a that the residues lining the degenerate NBS are different. This seems contradictory to the authors' goals.

-If Rv1273c/Rv1272c is indeed linked to drug resistance, then why were the drugs (isoniazid, ethambutol, etc) not tested by the authors? In their comments to the reviewers, the authors say that they tested the ATPase activity of several potential substrates and saw no change in activity. The authors should note that this result alone is not sufficient to draw conclusions and they will need to design and perform proper transport assays.

-Regarding the lack of vanadate – the authors suggest that the amount of vanadate they added to the sample for cryo-EM was not sufficient and that is why they did not see vanadate in their EM density map. If the authors believe this to be true, why did they not repeat the experiment with the proper amount of vanadate?

-The authors captured a state or states in which the degenerate site is 'clamped,' but there is no indication of nucleotide presence in the NBS. This raises suspicion and requires further experiments. Can the authors explain why they think such a conformation is physiologically relevant? It is highly doubtful that the NBDs can semi-dimerize without the nucleotide being present in the NBS. This suggests that the authors may have trapped an artificial state, and could be perhaps even an artefact of the detergent used.

-The proposed transport mechanism is still very speculative and not based on facts.

Reviewer #2

(Remarks to the Author)

The Authors answered all of my questions and concerns.

Version 2:

Reviewer comments:

Reviewer #1

(Remarks to the Author)

Thank you to the authors for the revised version and additional experiments. Unfortunately, there are still many concerns that remain. The manuscript can benefit from a rewrite to address the following:

- Proper rephrasing of the citations in the introduction
- Reorganization of the Results section
- Results and discussion are mixed, although there is also a separate Discussion section
- Authors jump from one topic to another and often it is difficult to follow their logic and hence, evaluate their science
- Section headings do not always correspond to what is discussed
- There is a lot of repetition throughout the text

- Text needs to be proofread for sentence structure and grammar mistakes
- More details are needed in the Methods section. For example, for the ATPase assay, it is unclear what system it was performed in. It is also not sufficient to say "as described previously" especially when the reference itself says the same thing and links to yet another reference. Methods should include enough detail so that they can be repeated by someone else.

In this new version, the authors present ATPase activity assay and transport assays. They did not see any change in ATPase activity in the presence of isoniazid, yet they conclude that the transporter acts as a drug efflux pump of isoniazid. The transport assay is also very confusing. It is unclear from the information presented, what the control was in this experiment.

Some suggested edits and concerns are listed below:

Ln 29 "for their transport" should be "of their transport mechanism"

Ln 31 "isoniazid efflux pump" -> implies that this was already known when it is something authors tested here

Ln 37 "a heterodimeric ABC transporter" -> please specify that you are talking for one particular ABC transporter and not all ABC transporters with a degenerate NBS

Ln 38 "asymmetry" -> which asymmetry are you referring to here? Between the MsRv1273c/72c or the active vs degenerate NBS?

Ln 142 – 146: Could you comment on how you would interpret the fact that you see no effect of the drugs on the rate of ATP hydrolysis?

Ln 147: "if the drugs had move inside the liposome when applied outside" -> please rephrase. Perhaps something along the lines of: "...to determine whether isoniazid and ethambutol could be imported by MsRv1273c/72c into the proteoliposomes."

Ln 151: "resistance of" -> should be "resistance to"

Ln 151: "whilst its effect on ethambutol" -> unclear what you are referring to / poor scientific language

Ln 152 – 153: Based on the results shown, it is not clear why the authors think that the transporter is a drug efflux pump for isoniazid. The ATPase activity seems to be unaffected by isoniazid. Furthermore, the cell-based transport assay shows diminished cell growth in the presence of isoniazid, but there is no cell death. The authors do not discuss this result.

Ln 178: "conserved Trp246" -> conserved with respect to what? Which species is it conserved in?

Ln 189: "which is extremely long" -> not very quantitative or scientific; unnecessary in the text

Ln 193-205: Out of place - unclear how this part fits into a section titled "Structure in the IFapo state"

Ln 208: "To capture MsRv1273c/72c in other states during transport..." -> needs to be rewritten to reflect what was done. These are not turnover conditions, so no states are being captured during transport. Instead, the authors try to capture a pre- and post-hydrolysis state as they discuss further in the text.

Ln211: It is unclear why the authors chose to use AMPPNP for their studies. What are the properties of this molecule that the authors were interested in? In particular, what additional information to that provided by EtoQ and vanadate were the authors hoping to obtain?

Ln 211 – 212: Please describe what you visually see for your AMPPNP state before you go on to discuss another structure.

Ln 220 – 240: The effect of an EtoQ mutation in the NBS is well established in the field. Perhaps the authors should discuss their experiment design and results in context to that. It is also unclear from the text why the authors pursued the D497N mutant.

Ln 242 - 243: "We then solved the cryo-EM structure in the ATP-bound Occ state...": Please note that the Occ state is what you observed based on your solved structure. This could be phrased better to indicate that your EtoQ mutant in the presence of ATP yielded a 3.1 Å structure that in the Occ state and then you can describe what this is characterized by.

Ln 248 – 249: The authors might want to include references to support their statement.

Ln 258: Here the text jumps back to AMPPNP, which was originally mentioned in Ln 211 – 212. The authors may want to reorganize the flow of information so that it is easy to follow and not repetitive.

Ln 262 – 279: This is now a completely different experiment and discussion within the AMPPNP section; the relevance is unclear.

Ln 281 – 285: Based on what is shown in the cited Figure, it is unclear what this discussion is about.

Ln 285: "...the ability of AMPPNP mediating NBD dimerization at consensus site" -> should read: the ability of AMPPNP to mediate NBD dimerization at the consensus site

Ln 301 "is not ideally bound" -> poor word choice, unclear what do you mean by "ideally"

Ln 304: Based on the section heading it is unclear which sample the authors are discussing.

Ln 310 – 311: The information here seems completely out of place.

Ln 320 – 328: The information here was already discussed in Ln 208 – 219.

Ln 325 – 328: It is irrelevant to discuss the sample that did not work. Please only keep the sample that had enough vanadate, which produced results consistent with the literature (i.e. bound vanadate where you expect it).

Ln 361 – 362: The sentence here needs to be rewritten in a grammatically correct way to convey the intended message.

Ln 362 – 364: The logic is unclear.

Ln 697: Here and throughout the rest of this section, "UN" should be "NU"

For the cryo-EM methods: (1) It will be good to mention exclusion criteria used after heterogeneous refinements. Also, if you performed any of the following, please include that as well – Local Refinement, Global CTF Refinement, Local CTF Refinement, Per-particle motion correction, use of any masks and at which stage. (2) Please note that the Supplementary Figures (workflows) for this section poorly reflect the actual processing done.

Reviewer #2

(Remarks to the Author)

The authors added new functional experiments to further support their results. While the manuscript has greatly improved, I have a few comments / questions.

The Authors did not provide any detail on the quality control of reconstitution into vesicles. Did they assess the integrity of vesicles and the efficacy of reconstitution? Upon detergent removal, membrane proteins might aggregate instead of being reconstituted into the liposomes. Can the Authors provide any data (negative stain EM / cryoelectron microscopy) / explanation on this issue? Also, the protein concentration of 20 mg/ml seems to be a bit high, especially for reconstituted proteins in liposomes. If it is not a typo (i.e. 2 mg/ml), what is the rationale behind it?

Reconstitution into proteoliposomes usually results in both the outside-out and the inside-out configurations. While the Authors adjusted the concentration of the liposome-reconstituted transporter, only the activity of the outside-out configuration can be measured (since ATP cannot enter the liposomes), which, assuming non-directed reconstitution (~50-50% distribution) would result in a significant increase in the ATPase rate. Do the Authors have any insight on this issue?

The proteoliposome-based transport-assay suggests that isoniazid can be transported by the heterodimer transporter, while ethambutol cannot be detected inside the vesicles. The methods section describes 3 tested concentrations of the drugs, while they show only one condition for each and forgot to specify the condition itself. Did the Authors observe concentration dependency? I am wondering, why the authors did use ethyl acetate which is known to be a less effective extracting agent for liposomes. Did the Authors/others test the distribution of proteoliposomes in the EtOAc/water system? Furthermore, the authors might want to consider using a system with LC-MS readout to quantify the amount of transported drugs.

While the Authors might want to tone further down the translational perspectives of the study due to the lack of physiological/functional data and the subtle differences between the two orthologs, I still think that the manuscript is suitable for publication, since the structural data is convincing, and the study is scientifically sound.

Reviewer #1 (Remarks to the Author):

Yu et al. present cryo-EM structures of the mycobacterial ABC transporter MsRv1272c/Rv1273c, a homolog of *Mycobacterium tuberculosis* (Mtb) Rv1272c/Rv1273c. Resistance towards the currently available drugs to treat tuberculosis (TB) has increased the urgency to identify new drug targets and drugs to treat TB. The authors suggest the heterodimeric ATP-binding cassette transporter Rv1272c/Rv1273c as a potential target, however prior to this study there was no information available about the structure and mechanism of this transporter. In this study, the authors aimed to capture the transporter in several different states. Interestingly, this is a heterodimeric transporter that has only one functional ATP-binding site. The authors present four nice high-resolution cryo-EM structures with well-resolved TMDs, NBDs and densities for the bound nucleotides. The authors show three structures in the inward-facing (IF) conformation – an apo state, an AMPPNP-bound structure that represents a state prior to ATP hydrolysis, given that AMPPNP cannot be hydrolyzed and an ADP-bound structure that represents a state after ATP hydrolysis. They also resolved a structure in the occluded state by introducing an EtoQ mutation in the consensus ATP-binding site.

While this seems to be a very interesting transporter to study, overall, the interpretation of the presented data is speculative and the results are incomplete and sometimes questionable. Furthermore, there is no functional data that supports the claims that the authors make. The study has potential, but a more comprehensive structural and functional analysis is needed.

Response: Thank you for your comments, questions and feedback. To improve our manuscript, we have collected additional Cryo-EM data to capture additional intermediate states during transport and have also provided new functional data to support our conclusions. The organization of paper has been optimized and more analysis has been added.

For example, we have performed an ATPase activity assay to show that the transporter is both active in detergent, DDM, and liposome environments. Through this assay we also proved that the addition of vanadate to the cryo-EM sample preparation is not

enough for inhibition. This could explain why we did not observe vanadate in the cryo-EM map of the ADP-bound **IF**^{turnover-2} (now **IF**^{asym-5}) state. To remove the possibility of an anomalous role for vanadate, we prepared new cryoEM samples in the presence of only ATP, incubated at 4 °C (**IF**^{asym-2}) and 37 °C (**IF**^{asym-3}), respectively, and only ADP, incubated at 4 °C (**IF**^{asym-4}) and then collected the cryo-EM data. All the solved structures showed **IF** conformations as a partial NBD dimer. **IF**^{asym-2} contains ATP at the degenerate site while no ATP/ADP is found there in the **IF**^{asym-3} and **IF**^{asym-4} structures. It is thus possible that ATP is released as ADP at the higher temperature after slow hydrolysis. In addition, we have also made changes to the text according to the reviewer's suggestions. We have added more information about Rv1273c/72c and properly summarized the backgrounds of ABC transporter in the Introduction section. We have reorganized the Results section. Each structure has been analyzed separately to improve the flow. More in depth discussion has been added in the Discussion section. The Methods section has been expanded to include more information about the experiments.

We hope these improvements will be sufficient to support our claims and be satisfied by the reviewer.

My comments below are ordered by appearance in the text, not necessarily by importance.

Abstract

In 33-35: Speculative statement as there is no structural or functional data presented to support this.

Response: We agree, it may require the identification of substrate which is currently not known to completely reveal the roles of the extending loop and helical hairpin structure. Considering the word limit of abstract, we have deleted such statement. The original sentence “The structures exhibit specific asymmetrical features such as an N-terminal gating loop and a periplasmic helical hairpin only in *MsRv1272c*, which help the substrate enter and release from the central cavity” has been changed as follows: “**Here, we report cryo-electron microscopy structures for the mycobacterial ABC transporter *MsRv1273c/72c* that show specific asymmetrical features including an N-terminal**

extending loop and a periplasmic helical hairpin only found in *MsRv1272c*"

Introduction

- The introduction does not make it clear what the focus of the study is and does not provide compelling evidence for why this study is necessary. There is also not enough information on Rv1272c/Rv1273c. For example, what is known about the topology of this heterodimer? What kind of functional analysis has been reported in the literature? Also, since it is suggested that this transporter could be a potential drug target, it might be helpful to include some information on the current drugs used for TB treatment and what their targets are.

Response: We have added the relevant information suggested by the reviewer in the Introduction section, including the domain composition and sequence identity of Rv1273c/72c, the type of functional analysis reported in the literature, current drugs used for TB treatment and their targets, et al.

For example, the following sentences have been added in the Introduction section:

“The current recommended treatment for TB involves a combination of four antibiotics: isoniazid (INH; targeting to InhA²), rifampicin (RIF; targeting to RNA polymerase³), pyrazinamide (PZA; target unknown) and ethambutol (EMB; targeting to EmbA/B/C⁴), which were all discovered nearly 70 years ago”

“Their coding proteins Rv1273c and Rv1272c are the TMD-NBD fusion proteins”

“Rv1273c shows 28% sequence identity with Rv1272c³² and it is suggested to be a homodimeric multidrug transporter with a wide substrate range as well as a probable contributor to biofilm formation”

“Further studies are needed to investigate whether Rv1273c interacts with Rv1272c”

As to why this study is necessary, we have stated in detail in the Introduction section and we will summary it by three aspects here, thus we have made it clear what the focus of our study.

(1) Limitation of current treatment of TB and new targets and drugs to treat TB are urgently needed. One possible class of proteins for drug discovery are the ATP-binding cassette (ABC) transporters, especially those with degenerate sites.

(2) The consequences of impaired degenerate NBS in the heterodimeric ABC

transporter remain unclear with respect to the mechanism for the transport cycle, which seems distinct from the homodimeric transporter. Therefore, more direct structural evidence is required to explain the exact role of the degenerate site during the process of transport.

(3) Rv1272c and Rv1273c are potential targets for the treatment of TB and for mitigating the effects of drug resistance. However, little is known about the molecular mechanism of how Rv1273c and Rv1272c functions as transporter proteins.

- In 54 and In 59: I would suggest more appropriate references are used instead of ref.1 and 2. For example, for ref.1, recent WHO data will be a better citation than a review.

Response: We have replaced the two references by the following two:

- 1 Organization, W. H. GLOBAL TUBERCULOSIS REPORT. (2023).
- 2 Akhtar, A. A. & Turner, D. P. J. The role of bacterial ATP-binding cassette (ABC) transporters in pathogenesis and virulence: Therapeutic and vaccine potential. *Microb Pathogenesis* 171 (2022).

- In 65-67: This should be rephrased to accurately summarize what was written in ref.9, which is that the conformational changes induced by binding and hydrolysis of ATP result in conformational changes of the TMDs that are likely transmitted through the coupling helices.

Response: The original sentence “The motional energy gained from the ATP-dependent association and dissociation at the NBDs is transmitted to the TMDs through coupling helices (CpHs), resulting in conformational changes that orient a substrate binding cavity either to the inside or the outside of the cell membrane” has been re-written as follows: "The binding and hydrolysis of ATP induces association and dissociation of the NBDs. This results in conformational changes of the TMDs that are likely transmitted through the coupling helices (CpHs)¹³. Consequently, the substrate binding cavity in the center of TMD will open either towards the cytoplasmic side (inward facing, **IF**) or towards the periplasmic side (outward facing, **OF**) of the cell membrane or be occluded (**Occ**) from both sides"

- In 70-94: This section needs to be rewritten to properly summarize and cite the papers referenced.

Response: The paragraph has been rewritten to properly summarize and cite the papers referenced. It now reads as follows:

“Transporters with two active consensus nucleotide-binding sites (NBSs) are called canonical transporters, while heterodimeric ABC exporters frequently have a degenerate NBS containing non-canonical residues that strongly impair ATP hydrolysis¹⁴. The proposed models of transport cycle for the canonical transporters are not compatible with an ABC transporter harboring a degenerate site, because these models (a) require the hydrolysis of two ATPs, (b) an obligate alternation of ATP hydrolysis between the two NBSs, or (c) impose complete NBD separation¹⁴. These ‘degenerate’ transporters must therefore have established a new mechanism, in which the degenerate NBS sustains active substrate translocation across a biological membrane¹⁴. Mutagenesis data suggest that hydrolysis in the consensus NBS is sufficient for all necessary conformational changes to complete a transport cycle, but subsequent hydrolysis in the degenerate site would further favor NBD opening¹⁵. Indirect evidence suggests that ATP might serve as a scaffold acting like a glue in the degenerate site, keeping this NBS in a closed conformation over several iterations of the transport cycles¹⁶⁻²⁰. Though several biochemical and structural investigations have been undertaken for this type of ABC exporter including TM278/288²¹⁻²³, TmrAB^{24,25}, TAP²⁶ and CFTR²⁷⁻²⁹, the consequences of having an impaired NBS remain unclear with respect to the mechanism for the transport cycle. Therefore, more direct structural evidence is required to explain the exact role of the degenerate site during the process of transport. Since the degenerate sites of different transporters often have unique substitutions which might have evolved multiple times³⁰, this site may prove to be the better target for the design of highly specific drugs^{15”}

- In 90: the manuscript states that Rv1272c/Rv1273c is predicted to form a heterodimer. The reference cited (ref. 20) is a review and therefore, not a direct reference. Furthermore, Rv1272c has been hypothesized to function as a homodimer (PMID: 29360453). Can the authors comment on that? Perhaps this should also be included/discussed in the introduction.

Response: We have modified the introduction as follows: “Their coding proteins, Rv1273c and Rv1272c, are the TMD-NBD fusion proteins and they are predicted to form a heterodimeric ABC exporter³¹. However, Rv1272c has also been hypothesized to function as a homodimer since it enhances the transport of radiolabeled long-chain fatty acids when expressed in *Escherichia coli* (*E. coli*)³². Rv1273c shows 28% sequence identity with Rv1272c³² and it is proposed as a homodimeric multidrug transporter with a wide substrate range as well as a probable contributor to biofilm formation³³. Further studies are needed to investigate whether Rv1273c interacts with Rv1272c.”

- In 95: Unclear what “this transporter” refers to. Is this in reference to the potential heterodimer?

Response: “this transporter” has been replaced by “Rv1272c and Rv1273c”. The original sentence “In addition, this transporter is also associated with the development of resistance of the drugs isoniazid²³, ethambutol²⁴ and bedaquiline²⁵, and as a result has been unequivocally identified as a multidrug transporter²⁰” has been revised as follows: “In addition, by using sequence analysis, Rv1272c and Rv1273c have been annotated as multidrug transporters³¹. Expression and mutation analysis of drug resistant clinical isolates of *Mtb* have shown Rv1273c is associated with the development of resistance towards drugs such as isoniazid³⁵, ethambutol³⁶ and bedaquiline³⁷”

- In 96-98: What kind of assays have been performed to show the multi-drug resistance role of the transporter? The manuscript states that this transporter has been “unequivocally identified as a multidrug transporter,” however I could not find support for this statement in the review cited (ref.20). My understanding is that this hypothesis is based on sequence analysis, but if that is not the case, more references and clear description should be included. Also, the references for the drugs listed discuss Rv1273c only.

Response: Expression and mutation analysis of the drug resistant clinical isolates from

Mtb have been used to show Rv1273c plays a role in drug resistance. Based on sequence analysis, Rv1272c and Rv1273c are classified as belonging to sub-family 6 of ABC transporters [31]. According to this reference the family is annotated as having the function of being multidrug transporters. Thus, the sentence in the text has been revised: “In addition, by using sequence analysis, Rv1272c and Rv1273c have been annotated as multidrug transporters³¹. Expression and mutation analysis of drug resistant clinical isolates of *Mtb* have shown Rv1273c is associated with the development of resistance towards drugs such as isoniazid³⁵, ethambutol³⁶ and bedaquiline³⁷”

- In 99: Ref. 26 does not include the information cited. Perhaps the wrong reference was listed and maybe the authors meant to refer to PMID: 29050768, instead. Additionally, it is unclear why the SNP is mentioned. Where is this mutation? How is it relevant for this study?

Response: It was the wrong reference cited. We have now corrected it by citing PMID: 29050768. Common SNPs (S118G, I175T) in Rv1273c have been found in several XDR clinical isolates of *Mtb* rather than the drug susceptible strains, these may be associated with an alternative mechanism of drug resistance in XDR-TB strains caused by Rv1273c. The sentence has been revised: “An analysis of extensively drug-resistant (XDR) clinical isolates of *Mtb* revealed common single nucleotide polymorphisms (SNPs) in Rv1273c such as S118G and I175T. These may be associated with an alternative mechanism of drug resistance in XDR-TB strains caused by Rv1273c³⁸”

- The introduction does not make it clear why the authors chose the homolog *Msm*. Additionally, there should be some information about what inward-facing, occluded and outward facing states are.

Response: We have revised the sentence to add information: “*MsRv1273c* (MSMEG_5008) and *MsRv1272c* (MSMEG_5009) from *Mycobacterium smegmatis* (*Msm*) are the homologs of Rv1273c and Rv1272c, respectively, sharing 69 and 75% sequence identities with their *Mtb* counterparts. We have focused on the *Msm* homologs because they were able to be successfully expressed and purified while this could not be achieved for the *Mtb* proteins.”

New information on the inward-facing, occluded and outward facing states is included in the first paragraph of the Introduction section: “Consequently, the substrate binding cavity in the center of the TMD will open either towards the cytoplasmic side (inward facing, **IF**) or towards the periplasmic side (outward facing, **OF**) of the cell membrane or be occluded (**Occ**) from both sides”

- In 107: The authors claim that that they present two turnover states. However, none of the samples prepared meet the criteria for turnover conditions. Turnover conditions imply that all ingredients needed for a transporter to go through its full transport cycle are present, including a substrate.

Response: Thank you for the explanation of turnover conditions. We have changed the expression of turnover conformation to intermediate asymmetric conformation in the sentence. We also avoid the use of turnover in other places in the text.

- In 108-109: As an extension of the above comment, the manuscript speculates on a mechanism; however, the results presented do not support the proposed mechanism.

Response: We have added new cryo-EM structures for intermediate states and biochemical results of inhibitory effect of vanadate in cryo-EM sample. These data and corresponding analysis provide more support for the proposed mechanism. The sentence has been revised: “With these data, the mechanism of degenerate NBS mediated transport by a heterodimeric ABC exporter can be proposed”

Results

- The results section is mixed with discussion, although there is also a separate discussion section. Both sections should be rewritten for clarity and to eliminate repetition. Furthermore, each structure should be discussed separately, which will also improve the flow. Note that it is also not clear from the purification subsection of the Results that these transporters were purified in detergent.

Response: The results and Discussion sections have been rewritten as suggested. Each structure has now been discussed separately. A comment on the detergent has now been

mentioned in the purification subsection of Results as follows: “Rv1273c and Rv1272c, as well as MsRv1273c and MsRv1272c, were tandemly expressed in *Msm* cells and then purified in the detergent, digitonin, as a complex with a Flag-tag or 6×His-tag fused at the C-terminus of Rv1272c and MsRv1272c, respectively.”

- In 117: Choice of nomenclature is a bit confusing since it is not immediately clear that MsRv1273c-Rv1272c implies that both half-transporters are from *Msm*.

Response: “*MsRv1273c-Rv1272c*” has been replaced by “*MsRv1273c/72c*” throughout the text.

- In 120: What do the authors mean by “functional ATPase”? It is important to note that ATPase activity does not necessarily mean a functional transporter. &

- ATPase assays: It is unclear how the ATPase activity assays were performed. Was this done in detergent or proteoliposomes? Ideally data should be shown for both. Have the authors performed these assays in the presence of vanadate? Does vanadate inhibit the transporter? Statistics were not reported and it is unclear what statistical analysis was performed.

Response: We have added statistics (mean \pm S.E.M., n = 3) and revised the sentence as follows: “To prove that *MsRv1273c/72c* can undertake ATP hydrolysis, an ATPase activity assay was performed. The results show that it has basal ATPase activity of 113.3 ± 13.9 nM P_i min⁻¹ mg⁻¹ and 98.8 ± 3.7 nM P_i min⁻¹ mg⁻¹ in detergent and proteoliposomes, respectively (mean \pm S.E.M., n = 3).” Also see figure below.

We have performed the activity assay in the presence of vanadate. The inhibitory effect is concentration dependent (see figure below). We apologize that we didn't include this information in the original manuscript when describing the cryo-EM sample preparation for IF^{turnover-2} state (IF^{asym-5}). The condition has been added in the Methods section: “For the ADP-bound IF^{asym-5} state, 10 mM ATP, 4 mM MgCl₂ and 2 mM Na₃VO₄ were incubated with the protein at 37 °C for 5 min before grid preparation.”

To explain why the vanadate is not observed in the structure, we performed an activity assay with protein at 1/10 concentration (10 μM) of that used for the cryo-EM study. When compounds were also assayed at 1/10 concentration (1 mM ATP, 1 mM ADP and 0.2 mM vanadate), there is only ~25% inhibition. Even 2 mM vanadate could only inhibit ~60% activity (see figure below). This suggests that 2 mM vanadate is not sufficient to fully bind to protein in the cryoEM sample. Thus, it is possible that NBSs is not occupied by vanadate. This finding has been added to the text.

- In 123: For clarity, it would be good to specify that AMPPNP is an analog of ATP that cannot be hydrolyzed.

Response: The sentence has been revised as follows: “Since AMPPNP is a non-hydrolysable analog of ATP, it can stabilize the NBD dimer at the dimer interface and lock a transporter in the **O_{cc}** or **O_F** state”

- In 126-128: The observation that there is ADP but no vanadate at the consensus site and no ATP at the degenerate site goes against what is expected when ATP and vanadate are added to the sample, given that inhibition by vanadate is well-characterized in the literature. Is it possible that perhaps ADP was used by mistake instead of ATP for sample preparation? This would also explain the absence of the nucleotide at the degenerate site.

Response: The condition is indeed ATP and vanadate. The primary purpose is to induce NBD closure to obtain the **O_{cc}** or **O_F** state. Unexpectedly, we captured the **II^{turnover-2}** state (**II^{asym-5}**). Through verification by the activity assay (see figure below), we found that the amount of vanadate added is not sufficient to inhibit protein when preparing the cryo-EM sample.

To further explain the absence of the nucleotide at the degenerate site, we have solved cryo-EM structures in three conditions, in which NBDs are all partially dimerized (see figure below). (1) When ADP was incubated in the sample at 4 °C (**IF^{asym-4}**), we found only the consensus site bound with ADP. (2) When ATP was incubated in the sample at 4 °C (**IF^{asym-2}**), we found ADP and ATP bound to the consensus and degenerate sites, respectively. Therefore, ATP was hydrolyzed to ADP at the consensus site, but the ATP at the degenerate site was unaltered. (3) When ATP was incubated in the sample at 37 °C (**IF^{asym-3}**), we found only the consensus site bound ADP. This means ATP is likely released as ADP from the degenerate site after hydrolysis. This is because the degenerate site is impaired for activity but it can still hydrolyze ATP, but much slower than the consensus site. Thus, based on these intermediate states, we can propose how the conformation of the transporter changes between **IF^{apo}** and **Occ** states, ultimately supporting our proposed mechanism.

The above analysis has been added in the text.

- In 129: The references to the figures should be moved to the appropriate location.

Response: Agree, this has been corrected. The sentence has been revised as follows:

“The cryo-EM structure was solved for this experiment and showed an ADP-bound IF^{asym-5} state at 3.4 Å resolution (Supplementary Fig. 8 and 10d)”

- In 143: This is the first mention of “gating loop” in the text, other than the abstract. Does such loop exist in other transporters? Why is it called “gating” loop? The authors may consider including something about this.

Response: Similar loops exist in other transporters such as TM287/288, it was stated: “The N terminus of TM288 is 25 residues longer than the one of TM287 and is partially defined in the crystal structure starting from residue 10 (Supplementary Fig. 7). It folds into an extended N-terminal elbow helix and interacts as a coiled structure with ICL4 of the membrane domain of TM288” in the paper (Hohl M, et al. Nat Struct Mol Biol. 2012. PMID: 22447242). We found that this loop inserts into the gap/gate between TM4 and TM6 of MsRv1272c, where it may affect substrate entering the central cavity. However, this may require further studies to confirm this idea. To modify our claim, we have changed its name to “extending loop” in the text, which is based on the statement for a similar loop in TM287/288.

- In 144: Fig. 1B. does not clearly show what is stated in the text.

Response: Fig. 1B (now Fig. 1e) has been revised by addition of a zoom-in view in the left to clearly show the details.

- In 150 and In 154: Contradiction in the statements. On In 150 it is suggested that residues 1-16 are resolved, while In 154 states that residues 1-9 are flexible.

Response: The sentence has been revised as follows: “The N-terminal extending loop (including residues 1-16, but only 10-16 are observed in the IF^{apo} structure) of *MsRv1272c* wraps around TM6 ...”

- In 155-157: The interpretation of what is written in the referenced manuscript (Hohl et al) is incorrect. The original manuscript does not suggest that the “loop restricts the conformation of the TM helices”. Hohl et al state that:

“The N terminus of TM288 is 25 residues longer than the one of TM287 and is partially defined in the crystal structure starting from residue 10 (Supplementary Fig. 7). It folds into an extended N-terminal elbow helix and interacts as a coiled structure with ICL4 of the membrane domain of TM288. This interaction appears to restrict the movement of ICL4 (and of NBD1 connected to it) with respect to the other transmembrane helices of TM288 and might therefore further strengthen the interaction between the NBDs.”

Response: The sentence has been revised as follows: “In that structure (PDB code: 3QF4), it is proposed that this loop restricts the movement of ICL4 (and of NBD1 connected to it) with respect to the other TM helices and might therefore further strengthen the interaction between the NBDs”

• In 160: Could the authors clarify in the text the structure of which mutant they solved? What was the consideration behind the number of residues chosen to delete for this mutant? The Methods section should include information about these different mutants. It is unclear from the manuscript why the authors thought that deleting the first 16 residues of the gating loop would influence the overall structure of the transporter and what they expected to see, given that they already specify that the first several residues are disordered. Furthermore, why did the authors expect that these deletions would have an effect on the ATPase activity?

Response: We have deleted residues 1-16 from the extended loop. The revised sentence now reads “To verify this, we made a deletion of the whole extending loop (Δ 1-16) and solved the low-resolution (6.93 Å) cryo-EM map of this truncated transporter”

We have also included additional information on the purification, cryoEM sample preparation and ATPase activity assay of the mutants in the Methods section. For example:

“All mutants of *MsRv1273c/72c* were generated by the TaKaRa MutanBEST Kit using the DNA sequence of the wild-type protein as the template. Expression plasmids were prepared in the same way as for the wild-type transporter. Mutated proteins were expressed and purified following the same protocol as the wild-type protein except that DDM was used during the entire purification procedure for the *MsRv1273c/72c*^{E553Q} mutant.”

“For the extending-loop truncated structure, the *MsRv1273c/72c* ^{Δ 1-16} mutant was incubated with 5 mM AMPPNP and 4 mM MgCl₂ at 4 °C for 30 min before grid preparation.”

“For the *MsRv1273c/72c* ^{Δ 1-16} mutant dataset, 1,094 micrographs were selected and 2,472,133 particles were extracted. After cleaning by 2D classification, 331,600 particles were used for Ab-Initio reconstruction. After several rounds of heterogeneous

refinement, 116,495 particles were refined using UN refinement to generate the final cryo-EM map with 6.93 Å resolution.”

“4.68 µg purified protein complex or the mutants was incubated in a 20 µL reaction volume containing 50 mM Tris (pH 8.0), 500 mM NaCl, 1 mM ATP and 4 mM MgCl₂ for 10 min at 37 °C”

Since AMPPNP is an analog of ATP which could induce an **OF** or **Occ** state by mediating the full closure of two NBDs (evidenced by the structures of MsbA (PDB code: 3B60) and Sav1866 (2ONJ)), we then solved the AMPPNP-bound structure. However, the **IF^{turnover-1}** state (now is renamed as **IF^{asym-1}**) was captured unexpectedly. There are gaps between TM4 and TM6 of *MsRv1272c* and between NBDs at the consensus site in this state. There could be two possible reasons to induce this conformation. One is the lack of an extending loop and the other is nucleotide. First, it is proposed that the insertion of extending loop blocks the closure between TM4 and TM6 when AMPPNP is added. However, the gaps still exist in the AMPPNP-bound structure when the extending loop is deleted (Supplementary Fig. 11a). This means the extending loop does not affect the conformational state. Besides that, we observed that an **Occ** state was induced in the ATP-bound *MsRv1273c/72c^{E553Q}* and the extending loop was squeezed out from the TM region (Supplementary Fig. 11c). This also provides additional supporting evidence that the extending loop does not affect the conformational state when ATP is added, instead, replacing AMPPNP by ATP could change conformations.

ATPase activity could be affected by the dimerization of NBDs which is coupled by the conformation of TMDs. If the extending loop blocks conformational changes in the TM region, then ATPase activity will be affected. Thus, we tested the activity for different truncations at the extending loop. The results showed no apparent changes for activity by these mutants compared to the wildtype (Supplementary Fig. 11d). This further supports the hypothesis that the extending loop does not affect the conformational state of the transporter. All these results together show that the structures of the **IF^{turnover-1}** (**IF^{asym-1}**) state and **Occ** state are induced by the nucleotides rather than the extending loop.

The above analyses have been added in the text.

- In 166: How can the authors deduce that the “gating loop” does not modulate transport without having performed any transport assays?

Response: the original sentence “These data suggest that the gating loop does not modulate transport by altering the gap between TM4 and TM6” has been revised: “These data suggest that the extending loop does not modulate function by affecting conformational changes”

- In 168-170: The hypothesis that the gating loop might help the substrate enter the cavity is speculative and not supported by the data presented here. It might be more appropriate to make this suggestion in the conclusion as future experiment, rather than use it as an absolute statement. The caption of Fig. 4 should also be edited accordingly.

Response: To replace the statement, we have revised the sentence as follows: “Since the extending loop is inserted and points into the central cavity in the IF^{apo} state (Fig. 1f), we hypothesize that this loop might help the substrate enter the cavity.” The following statement has also been added in the Discussion section: “Further evidence will be required in future studies to understand the precise roles of the extending loop and helical hairpin” The caption of Fig.4 (now is Fig. 5) has also been revised: “Entry and release of substrate is proposed to require the extending loop and ECD of *MsRv1272*”

- In 173: “since the substrate(s) of *MsRv1273c*-*Rv1272c* are yet to be identified” – the authors implied in the introduction (ln 94-97) that long-chain fatty acids and three drugs are substrates of the transporter. The two statements are contradictory.

Response: The statement in line 173 has been deleted in the revised manuscript. It has previously been reported that transport of long-chain fatty acids and three drugs could be affected by *Rv1272c* and *Rv1273c*. However, there is no direct evidence that *MsRv1273c/72c* could transport these as substrates. Here, we have tested the ATPase activity of several potential substrates including oleic acid, mycolic acid, isoniazid, ethambutol, rifampicin and bedaquiline. None of them affected activity.

- In 185-187: This is another speculative statement not supported by data. Furthermore, it raises the question of how exactly where these “predicted” OF models generated? This is not described anywhere in the Methods.

Response: We have now added a description for the generation of the OF models in the text. The statement has been revised as follows: “To generate the OF model of *MsRv1273c/72c*, we superposed the two wings of TM helices in the IF^{apo} structure onto the OF structure of TmrAB (PDB ID: 6RAH). In the two OF models (Supplementary Fig. 12a), the ECD moves as a rigid body together with either of the wings. From that alignment we could propose that the helical hairpin structure may interact with substrate when it exits from the central cavity, since the ECD still covers over the substrate exit location” We have also mentioned it to be a topic of future study in the Discussion section: “Further evidence will be required in future studies to understand the precise roles of the extending loop and helical hairpin”

- In 214: Why did the authors choose the D497N mutations and what did they hypothesize they will see?

Response: Asp497 is located at the corresponding position as Glu in the Walker B motif of NBD. Since the conserved Glu has been verified to be crucial for ATPase activity, we wanted to know if Asp is important for the ATPase activity considering the side chain of Asp is shorter than Glu. Therefore, we choose to test the D497N mutation. We could see that this mutation only slightly reduced the activity and the NBD of *MsRv1273c* containing Asp497 was recognized as a degenerate NBD. We have rewritten this paragraph to more clearly present our motivation for the experiment and analysis of the mutation results: “Sequence alignment with other ATPases shows that Glu553 in the Walker B motif of *MsRv1272c* is the residue that activates the attacking water molecule during ATP hydrolysis⁴¹. In *MsRv1273c* this residue is replaced by Asp497 (Fig. 2a). Another conserved residue is His584 in the Switch-loop of *MsRv1272c* which plays a central role in stabilizing the transition state⁴². It is replaced by Gln528 in *MsRv1273c* (Fig. 2a). Thus, *MsRv1273c* appears to have a degenerate site while *MsRv1272c* appears to contain a consensus site. To verify this, we performed an ATPase activity assay for the transporter and its mutants. The result showed that the

E553Q mutation at the consensus site significantly diminished ATP hydrolysis, but still with residual activity (Fig. 2b). This suggests that Glu553 is crucial for ATPase activity of the transporter while Asp497 could still slowly hydrolyze ATP, even though the side chain is shorter than Glu, and may not be ideal for catalysis. In contrast, the D497N mutant at the degenerate site did not show a significant loss of ATPase activity compared to the wild-type transporter (Fig. 2b). This is because Glu553 is still fully functional at the other site. In addition, the D497N/E553Q double mutant had little activity (Fig. 2b). Together these results indicate that the consensus site is functional with normal ATPase activity while the degenerate site exhibits very low activity”

- In 257-259: Why do the authors believe that ATP can be “squeezed out” during the cryo-EM sample preparation but AMPPNP cannot? Cryo-EM sample preparation will not cause a nucleotide to be “squeezed out”, however, if we were to assume that this were to happen, then why would the authors consider this structure in their further analysis of the mechanism of the transporter if they believe it is an artifact of sample preparation?

Response: We think that the $\mathbf{IF}^{\text{turnover-2}}$ ($\mathbf{IF}^{\text{asym-5}}$) state induced by adding ATP and vanadate is inadequate to explain why ATP at the degenerate site is disappeared. To avoid the confusion of the possible presence of vanadate and to resolve the mechanism of transport, we solved three new cryo-EM structures by incubating sample with ADP at 4 °C, ATP at 4 °C, ATP at 37 °C, respectively before data collection. All the solved structures showed \mathbf{IF} asymmetric conformations with NBDs partially dimerized (named as $\mathbf{IF}^{\text{asym-4}}$, $\mathbf{IF}^{\text{asym-2}}$, $\mathbf{IF}^{\text{asym-3}}$, respectively). Among them, $\mathbf{IF}^{\text{asym-3}}$ and $\mathbf{IF}^{\text{asym-4}}$ states are similar to the previous $\mathbf{IF}^{\text{asym-5}}$ state (Supplementary Fig. 16b). In the first structure ($\mathbf{IF}^{\text{asym-4}}$) we found ADP does not bind to the degenerate site. In the second structure ($\mathbf{IF}^{\text{asym-2}}$) we found ATP binds to the degenerate site before hydrolysis occurs, this is consistent with the finding that AMPPNP also binds to the degenerate site in the $\mathbf{IF}^{\text{turnover-1}}$ ($\mathbf{IF}^{\text{asym-1}}$) structure. In the third structure ($\mathbf{IF}^{\text{asym-3}}$) we found no nucleotide at the degenerate site. This is because ATP has been hydrolyzed to ADP at the degenerate site when preparing the sample at 37 °C, and as seen in the $\mathbf{IF}^{\text{asym-4}}$ structure, ADP cannot strongly bind to the distorted degenerate site and it is very easy to release it from this site. This analysis has been added in the text to replace the original statement.

- In 267-270: It is unclear what the authors mean here and why MsRv1273c-Rv1272c is compared to a substrate-bound CFTR. &
- In 270-272: This statement is also unclear. What do the authors mean by “Although ATP at the consensus site contacts both NBDs, this state does not exist physiologically”? Why do the authors think this is the case?

Response: The CFTR structure also adopts an asymmetric conformation when the NBDs are partially dimerized. However, its dimer interface is located at the consensus site rather than degenerate site. Besides that, this is not a state that exists physiologically, because elexacaftor is not an endogenous molecule in the cell. We compared our structures with other proteins including CFTR to show that we have captured the intermediate states of the ABC transporter distinct from previously reported conformations. Thus, these are very novel findings. The text has been revised: “An elexacaftor-bound ABC transporter CFTR $\Delta 508$ mutant (PDB code: 8EIG)⁴⁸ also has a “cracked-open” NBD dimer structure partially held together by the presence of ATP at the consensus site. This contrasts with the *MsRv1273c/72c* structures that dimerize at the degenerate site (Supplementary Fig. 13e). Because elexacaftor is not endogenous to the cell, this is not a conformation likely to exist physiologically.”

- In 276-283: Again, please note that none of the structures were determined under turnover conditions.

Response: Thanks. The expression of “turnover” has been replaced by “intermediate”. The expression of “**IF**turnover” has been replaced by “**IF**asym” in the text.

- In 308-309 and In 316. The manuscript alludes to weak ATP binding at the consensus site. It is unclear how this conclusion was drawn based on the apo structure.

Response: We agree and have deleted corresponding analysis and revised the statement in the text: “For example, the fragment containing a part of the D-loop, a part of the Walker B motif and the connecting residues (⁵⁵⁴AT⁵⁵⁵), which we assigned as the D-WalkerB loop (⁵⁵²DEATSSVD⁵⁵⁹), adopts a classical S-shaped conformation when the

NBD binds nucleotide (Fig. 4b and Supplementary Fig. 13h). However, the loop is adjusted to form an L-shaped conformation in the IF^{apo} state (Fig. 4b and Supplementary Fig. 13g). The catalytic residue, Glu553, points in the opposite direction to the consensus NBS. Meanwhile, Asp552 of the Walker B motif also shifts away from the NBS (Fig. 4c). On the other hand, the locations of $\alpha 7$ and $\alpha 8$ have changed and a gap is created to accommodate the side chain of Glu553. Glu553 is further stabilized through interactions with Arg585 and Thr588 (Fig. 4c). Meanwhile, His584 of the Switch-loop linking to $\alpha 8$ is also pulled away from the NBS (Fig. 4c). Thus, the consensus NBS is distorted in the IF^{apo} state.”

- In 357: reference should be included for the alternating-access transport model

Response: The following reference has been added:

Beis, K. Structural basis for the mechanism of ABC transporters. *Biochem Soc T* 43, 889-893 (2015).

Discussion

- In 390-391: unclear what “more variable in structure and conformation” means

Response: We mean that when compared to the common half structure of type IV ABC transporter (such as MsbA), *MsRv1272c* has many additional structures such as the N-terminal extending loop and ECD and has more conformational changes such as changes at the D-WalkerB loop and X-signature loop. These are function related. The sentence has been revised as follows: “*MsRv1273c* bears the degenerate site and the classical fold of half an ABC transporter, while *MsRv1272c* contains more structural features compared to the canonical type IV ABC transporter.”

- In 402-403: This is a very vague statement. Can the authors elaborate what they mean by “specific mechanism of transport” for heterodimeric transporters with degenerate site? Could the authors provide references for this?

Response: We have used an inappropriate word. The expression “specific mechanism of transport” has been revised as “mechanism of transport”. The sentence has been

revised: “Because of the existence of degenerate site, the heterodimeric ABC transporters have a mechanism of transport that is different from the homodimeric counterparts.”

Methods

- This section should be expanded to include information about all samples generated, including the different mutants studied. There should also be adequate information about the composition of each sample used for cryo-EM studies. For example, for the AMPPNP-bound and ADP-bound samples, there is no mention of magnesium being added. The cryo-EM grid preparation should also include information about the concentration of each sample. Was this fresh sample?

Response: The methods section has been expanded to include the information suggested. For example:

“For cryo-EM grid preparation, aliquots (3 μL) of fresh protein samples at 10 mg mL^{-1} after purification were immediately applied to H_2/O_2 glow-discharged holey carbon grids (Quantifoil Cu R1.2/1.3).”

“For the AMPPNP-bound $\text{IF}^{\text{asym-1}}$ state, the protein was incubated with 5 mM AMPPNP and 4 mM MgCl_2 at 4 $^\circ\text{C}$ for 30 min before grid preparation. For the ATP-bound **Occ** state, the *MsRv1273c/72c*^{E553Q} mutant was incubated with 4 mM ATP and 4 mM MgCl_2 at 4 $^\circ\text{C}$ for 30 min before grid preparation. For the ATP|ADP-bound $\text{IF}^{\text{asym-2}}$ state, the protein was incubated with 10 mM ATP and 4 mM MgCl_2 at 4 $^\circ\text{C}$ for 30 min before grid preparation. For the ADP-bound $\text{IF}^{\text{asym-3}}$ state, the protein was incubated with 10 mM ATP and 4 mM MgCl_2 at 37 $^\circ\text{C}$ for 5 min before grid preparation. For the ADP-bound $\text{IF}^{\text{asym-4}}$ state, 10 mM ADP and 4 mM MgCl_2 were incubated with the protein at 4 $^\circ\text{C}$ for 30 min before grid preparation. For the ADP-bound $\text{IF}^{\text{asym-5}}$ state, 10 mM ATP, 4 mM MgCl_2 and 2 mM Na_3VO_4 were incubated with the protein at 37 $^\circ\text{C}$ for 5 min before grid preparation. For the extending-loop truncated structure, the *MsRv1273c/72c* ^{$\Delta 1-16$} mutant was incubated with 5 mM AMPPNP and 4 mM MgCl_2 at 4 $^\circ\text{C}$ for 30 min before grid preparation.”

- The cryo-EM image processing section should be expanded to include enough details and information so that the reader can get a comprehensive understanding of how the

data was processed and how decisions were made at each stage of processing. Ideally, this should be done for each sample, although it may seem repetitive. There should also be description and workflow for the low-res mutant sample.

Response: The cryo-EM image processing section has been expanded to include more details and information of the preparation of each sample including the low-res mutant sample: “For the dataset of **IF^{apo}** state, 4,863 micrographs were selected after those images that exhibited defects in the Thon rings due to excessive drift, ice contamination, or astigmatism were discarded. 3,566,457 particles were automatically extracted using a box size of 320 pixels and subjected to several rounds of reference-free 2D classification to discard bad particles, yielding a stack of 378,470 particles which were used for Ab-Initio reconstruction to generate 3D models as references to perform heterogeneous refinement. After several rounds of heterogeneous refinement, 139,918 particles were refined using non-uniform (NU) refinement to generate the final cryo-EM map with an estimated average resolution of 3.1 Å according to the gold-standard Fourier shell correlation cutoff of 0.143⁵⁵. All the other datasets were processed in the same pipeline. For the **IF^{asym-1}** state dataset, 3,206 micrographs were selected and 3,132,968 particles were extracted. After cleaning by 2D classification, 353,371 particles were used for Ab-Initio reconstruction. After several rounds of heterogeneous refinement, 132,058 particles were refined using NU refinement to generate the final cryo-EM map with 3.09 Å resolution. For the **Occ** state dataset, 3,073 micrographs were selected and 2,751,005 particles were extracted. After cleaning by 2D classification, 474,336 particles were used for Ab-Initio reconstruction. After several rounds of heterogeneous refinement, 190,634 particles were refined using NU refinement to generate the final cryo-EM map with 3.13 Å resolution. For the **IF^{asym-2}** state dataset, 6,663 micrographs were selected, and 2,436,646 particles were extracted. After cleaning by 2D classification, 888,523 particles were used for Ab-Initio reconstruction. After several rounds of heterogeneous refinement, 549,764 particles were refined using NU refinement to generate the final cryo-EM map with 2.93 Å resolution. For the **IF^{asym-3}** state dataset, 6,176 micrographs were selected and 4,860,724 particles were extracted. After cleaning by 2D classification, 451,589 particles were used for Ab-Initio reconstruction. After several rounds of heterogeneous refinement, 166,490 particles were refined using NU refinement to generate the final

cryo-EM map with 3.33 Å resolution. For the **IF^{asym-4}** state dataset, 3,411 micrographs were selected and 2,366,288 particles were extracted. After cleaning by 2D classification, 350,776 particles were used for Ab-Initio reconstruction. After several rounds of heterogeneous refinement, 155,852 particles were refined using NU refinement to generate the final cryo-EM map with 3.10 Å resolution. For the **IF^{asym-5}** state dataset, 1,094 micrographs were selected and 2,472,133 particles were extracted. After cleaning by 2D classification, 331,600 particles were used for Ab-Initio reconstruction. After several rounds of heterogeneous refinement, 116,495 particles were refined using NU refinement to generate the final cryo-EM map with 3.40 Å resolution. For the *MsRv1273c/72c^{Δ1-16}* mutant dataset, 1,094 micrographs were selected and 2,472,133 particles were extracted. After cleaning by 2D classification, 331,600 particles were used for Ab-Initio reconstruction. After several rounds of heterogeneous refinement, 116,495 particles were refined using NU refinement to generate the final cryo-EM map with 6.93 Å resolution.”

The workflow of data process for the low-res mutant sample (Supplementary Fig. 9) has been added.

- Which cryoSPARC version was used? Which CTF estimation routine was chosen?

Response: This information has been added as follows: “CTF estimation was performed using the “Patch CTF Estimation” program in cryoSPARC (v4.0.1).”

- Was there any postprocessing done - for example, per particle motion correction and ctf refinement? Were masks used?

Response: No postprocessing was performed. No masks were used.

- The model building and refinement section also needs to be expanded to include more details. Which map exactly was used for the building of the initial model? Was the backbone built by hand, or was a backbone tracing software used or another starting model used (ex. AlphaFold)? What about unresolved regions in the maps? This would be a good place to point out segments for each map that were unresolved and therefore not built.

Response: The model building and refinement section has been expanded to include more details: “The AlphaFold2 predicted models of *MsRv1273c* (AF-A0R271-F1) and *MsRv1272c* (AF-A0R272-F1) were used as the initial models, and they were rigid body fitted into the cryo-EM map of *MsRv1273c/72c* complex in the **IF^{apo}** state and adjusted manually using Coot. The structure was refined in real space using PHENIX⁵⁶ with secondary structure and geometry restraints to prevent over-fitting. The final atomic model was evaluated using MolProbity⁵⁷. The structure in the **IF^{apo}** state was further used as the initial model for other states. The processes of model building and refinement for these states are similar to the **IF^{apo}** state except that ATP, ADP, AMPPNP and Mg²⁺ were fitted into the corresponding cryo-EM map according to the additional non-protein density.”

Supplementary Table 2 has been added to summarize the composition of the models including ligands and fragments that have been resolved or are unresolved.

Structure model	Subunit	Residue built	Residue not built	Residue built (from vector)	ligand built

IF^{apo}	MsRv1273c	1-571	572-578	-1-0	--
	MsRv1272c	9-624	1-8, 625	--	--
IF^{asym-1}	MsRv1273c	1-572	573-578	-1-0	AMPPNP, Mg
	MsRv1272c	6-625	1-5	626-630	AMPPNP, Mg
IF^{asym-2}	MsRv1273c	1-572	573-578	-1-0	ATP, Mg
	MsRv1272c	13-625	1-12	626-628	ADP, Mg
IF^{asym-3}	MsRv1273c	1-572	573-578	-1-0	--
	MsRv1272c	13-625	1-12	626-631	ADP, Mg
IF^{asym-4}	MsRv1273c	1-572	573-578	-1-0	--
	MsRv1272c	13-625	1-12	626-631	ADP, Mg
IF^{asym-5}	MsRv1273c	1-572	573-578	-5-0	--
	MsRv1272c	6-625	1-5	626-631	ADP
Occ	MsRv1273c	1-572	573-578	-1-0	ATP, Mg
	MsRv1272c	14-625	1-13	626-629	ATP, Mg

Reviewer #2 (Remarks to the Author):

The Authors conducted structural studies on a heterodimeric ABC exporter, Rv1273-Rv1272 found in *Mycobacterium smegmantis* applying different conditions to gain mechanistic insights into the transport cycle of the membrane protein. As both Rv1272c and Rv1273c play a crucial role in the physiology of the pathogen, the medical relevance is inevitably high. In the current study, the Authors present four structures of the transporter; an inward-facing apo structure, two inward-facing structures with AMPPNP or ADP, and one nucleotide-occluded structure harboring the E-Q nonhydrolytic mutation. Based on careful analysis of the structural models, the Authors conclude a possible transport cycle for the heterodimer consisting of 6 states with two major transport cycles. While the work bears high significance to the field of ABC transporters, there are several concerns, which should be addressed before publication. Most importantly, I especially lack biochemical and functional experiments, which could support/modify the Authors conclusion(s). Such data would be vital to discuss

the transport mechanism in greater depth.

Response: Thank you for the positive comments. We have added new biochemical and functional experiments including ATPase activity assay for the protein in the proteoliposomes and validation of the inhibitory effect of vanadate. Moreover, we have solved three more structures to capture the intermediate states during transport. We have also added discussion in the text based on our responses to the reviewer's comments. These data and analyzes add support to our original conclusions and discuss the transport mechanism in greater depth. We hope these efforts will address all the concerns from the reviewer.

Main comments:

1) I am puzzled by the structure of IF(turnover-1) with AMPPNP. The Authors state that the reason for having a tight binding at the degenerate site and a loose connection at the consensus site with AMPPNP is that the degenerate site binds the nucleotide first, acting as a scaffold to promote the dimerization at the other site. As a non-hydrolysable/poorly hydrolysable ATP analog, AMPPNP often locks ABC transporters in the occluded / outward-facing states and acts as a 'non-hydrolytic' mutation, by the disruption and simplification of the transport cycle into a reversible IF-OF transition. As the microenvironments of the two composite ATP binding sites are immensely different, another possible explanation for the IF(turnover-1) state is that AMPPNP cannot bind at the consensus site as strongly as it fits into the degenerate site. Is there any evidence that the binding of AMPPNP at the consensus site of Ms1273c/1272c is similar to that of ATP? It is also quite surprising that the stability of such a 'transition state' between the IF and OF conformations is high enough to capture it by cryoelectron microscopy. Techniques, such as single-molecule FRET could be suitable to lead to a conclusion about the extent of NBD dimer formation/dissociation at the composite sites, separately.

Response: We agree with the reviewer's comment. The microenvironments of the two composite ATP binding sites are different, this may mean that AMPPNP cannot bind at the consensus site as strongly as it fits into the degenerate site. Thus, the ability of

AMPPNP to mediate NBD dimerization at the consensus site is reduced compared to that at the degenerate site. Since AMPPNP is an analog of ATP, it should replicate how ATP binds before it gets hydrolyzed. The binding mode of AMPPNP at the consensus site of *MsRv1272c* superposed well with that of ATP in the **Occ** state (see figure below). However, when comparing the **IF^{turnover-1}** (**IF^{asym-1}**) and **Occ** structures, we further deduced that the ability of AMPPNP mediating NBD dimerization at consensus site is less than ATP at the consensus site, because ATP can induce full closure of NBDs at consensus site. Thus, the statement “the reason for having tight binding at the degenerate site and loose connection at the consensus site with AMPPNP is that the degenerate site binds the nucleotide first, acting as a scaffold to promote the dimerization at the other site” is not appropriate here and we have deleted it in the text. Replacement analysis has been added to the text: “This means that AMPPNP mediates the closure of NBDs at the degenerate site whereas the NBDs are still separated at the consensus site (Fig. 3b). In other words, the ability of AMPPNP mediating NBD dimerization at consensus site is reduced compared with that at degenerate site. Since AMPPNP is an analog of ATP, it should bind in a similar fashion as ATP. As expected, we observed that the binding mode of AMPPNP at consensus site in the **IF^{asym-1}** structure superposes well with that of ATP in the **Occ** structure (Supplementary Fig. 13j). However, when comparing the two structures, we can further deduce that the ability of AMPPNP mediating NBD dimerization at consensus site is less than ATP at the consensus site, noting that ATP can induce full closure of NBDs at consensus site.”

In order to prove the extent of NBD dimer formation/dissociation at the composite sites in solution, there are three possible methods: (1) double electron–electron resonance (DEER) (Hutter et al. Nat Commun. 2019;10(1):2260), (2) smFRET (Roy R, et al. Nat

Methods. 2008. PMID: 18511918) and (3) a disulfide-bond forming (DSBF) assay (Hohl M, et al. Nat Struct Mol Biol. 2012. PMID: 22447242). All these methods require the Cys to be removed from the sample and incorporation of a new pair of Cys residues at the ideal position for detection of spin labels, dye labels or cross-linking. Since DEER requires special equipment, this experiment was not possible for us to carry out at this time. As for smFRET, it can detect distance changes as small as 3 Å when the labeled dyes are separate at a distance larger than 30 Å. We are not sure which position is suitable for labeling since there is no reference about detecting NBD partial dimer using smFRET. Thus, we should try many mutations but there is also a risk that the mutants cannot be expressed and purified in time for resubmission of the paper. Therefore, we chose the DSBF assay to perform these experiments and analysis (Hohl M, et al. Nat Struct Mol Biol. 2012. PMID: 22447242).

First, we mutated all the five Cys residues in *MsRv1273c* (C71S, C304S, C342S, C382S and C560S) one-by-one, then we introduced a pair of Cys residues around each NBS (A466C⁷³ and D166C⁷² for the consensus site, D524C⁷² and G107C⁷³ for the degenerate site) at similar positions according to the paper for TM287/288 (Hohl M, et al. Nat Struct Mol Biol. 2012. PMID: 22447242). We then expressed and purified the mutated samples, but with low yield. Due to the limited time to setup the experiment, we have only obtained preliminary results so far. Our results (see figure below) showed that AMPPNP could enhance the formation of the disulfide bond between C107 and C524 indicating NBD closure at the degenerate site, while there are no significant changes at the consensus site. However, the reproducibility of this experiment was not good and the position of Cys seemed not suitable for our sample. This is because there was a high degree of cross-linking when the mutated protein was purified, which may have interfered with the interpretation of the results. Though we added 2 mM DTT (even tried 20 mM) before experiment, the cross-linking cannot be completely removed. Therefore, more effort will be required to optimize this experiment and the current data cannot be included in the text.

On the other hand, cryo-EM proved useful for capturing intermediate states during transport (Hofmann et al. Nature. 2019;571(7766):580-583.) In our study, it could also demonstrate the extent of NBD dimer formation/dissociation at the composite sites. Through data processing, we only found one state ($\mathbf{IF}^{\text{asym-1}}$) when the sample was treated with AMPPNP. We did not find the \mathbf{IF}^{apo} or \mathbf{Occ} state. This means the majority of our sample is trapped in the $\mathbf{IF}^{\text{asym-1}}$ state when adding AMPPNP, which could represent the conformation of our sample in solution.

2) The E-Q nonhydrolytic mutation abruptly ATP hydrolysis, promoting a tight NBD dimer and restricting the conformation of the TMDs into an outward-facing conformation for most ABC proteins. The Authors state that the reason for not observing the OF state is its instability. However, it may need additional discussion, since it is quite surprising that none of the prehydrolytic conditions (AMPPNP / E-Q) lead to an OF conformation, as all the reported structures are in the IF conformation. Did the Authors attempt to stabilize the OF conformation in other ways? Identification of the substrate might help in this issue and would strengthen the physiology and

translational science of the transporter as well. Given the recent literature around Rv1273c and Rv1272c, the development of a transporter assay could be useful to investigate the substrate (e.g. long chain fatty acids) of the heterodimeric ABC transporter.

Response: In the study of TmrAB (Hofmann S, et al Nature 2019;571(7766):580-583), the authors suggest that in the resulting **OF** state, the transporter can transit between **OF** and **Occ** conformations until the release of P_i from the canonical site weakens the interactions between the NBDs. This could also be the case for our transporter. In another study for TM287/288 (Hutter CAJ, et al Nat Commun. 2019;10(1):2260), the authors found that the mechanical force of the firmly sealed extracellular gate (closed D-lock) is required to dissociate the NBDs after ATP hydrolysis in order to reset the transporter to its **IF** state. Thus, on the contrary, when the force of forming the NBD dimer is not strong enough to overcome the tension/interactions at the extracellular gate, the transporter will be stabilized in the **Occ** state. This could be the reason that our structure is stabilized at the **Occ** state rather than **OF** state. It lacks energy to break the D-locks. These points have been added to the Discussion section.

We agree with the reviewer to use substrate to induce the **OF** state, because binding of substrate in the central cavity probably makes the extracellular gate easy to open. However, we did not identify its substrate by the ATPase activity assay. Though we have tested several compounds such as oleic acid, mycolic acid, isoniazid, ethambutol, rifampicin and bedaquiline, none of them enhances the activity. The transporter assay seems to be difficult to setup because potential substrates need to be labelled for measurement. Hence, the identification of substrate will be a future study. During revision of the paper, we have also tried to mutate the D-locks since mutation of D-locks will shift the equilibrium towards the **OF** state. Unfortunately, we failed to obtain the mutant sample (D61A^{MsRv1272c}&D43A^{MsRv1273c} double mutant) after expression and purification.

3) As written above, the Authors state that ATP binds to the degenerate site first, and acts as a glue, followed by the binding of the second ATP molecule at the consensus site (if the reason for the loose consensus site is indeed the mechanism the Authors

suggest). However, regarding the IF(turnover-2) structure, the Author states that ATP is squeezed out from the degenerate site as the NBDs rotate to each other. This would mean that ADP, besides destabilizing the dimer leading to the dissociation, also causes somehow the loss of the ATP at the degenerate site (through a conformational change?), which is quite unlikely in my opinion. Thus there might be a contradiction between IF(turnover-1) and (turnover-2). Indeed, for other ABC proteins with a degenerate site (e.g. CFTR), ATP remains bound for several transport cycle. How can the Authors explain the controversy between IF(turnover-1) and IF(Turnover-2)?

Response: Given the concern raised by the reviewer, we did further analysis and we thought our statement “first binding at the degenerate site followed by the binding of the second ATP molecule at the consensus site” may be inappropriate and as a result we have deleted it. This is because there is the possibility that ATP can bind at both NBSs simultaneously. However, due to the microenvironments potentially being different, their binding affinities maybe different. To explain the loss of ATP at the degenerate site, we did more cryo-EM studies to capture the transporter in the **IF^{asym-2}** state when ATP was incubated at 4 °C, in the **IF^{asym-3}** state when ATP was incubated at 37 °C, in the **IF^{asym-4}** state when ADP was incubated at 4 °C. **IF^{asym-3}** and **IF^{asym-4}** states are similar to the **IF^{turnover-2}** (renamed as **IF^{asym-5}**) state (Supplementary Fig. 16b). From the **IF^{asym-4}** state, we found ADP could induce NBD dimerization at the degenerate site but cannot bind to this distorted site. Therefore, it is easily released from this site due to its weak affinity. From the **IF^{asym-2}** state, we found ATP at the degenerate site as expected, which is also consistent with the **IF^{turnover-1}** (renamed as **IF^{asym-1}**) structure that AMPPNP binds at the degenerate site. However, ATP is hydrolyzed to ADP at the consensus site and binds there as ADP. From the **IF^{asym-3}** state, ATP is missing at the degenerate site. A higher temperature will accelerate ATP hydrolysis at both NBSs given that the degenerate site has impaired but not total loss of ATPase activity. When ATP is hydrolyzed to ADP at the degenerate site, it will be easily released as observed in the **IF^{asym-4}** state. Thus, this analysis better explains the underlying reasons for the conformational changes that occur. We hope that the conjectural comments in the original version of the manuscript have been revised satisfactorily in the new text.

4) Is there any reason (other than the better purity/higher yield) that the Authors

characterized the rather non-pathogenic *Ms* transporter and not the pathogenic *Mtb* heterodimer? While the SEC chromatogram of *Ms* Rv1273c/Rv1272c looks evidently better, the ~70% sequence identity can often mean radically different microenvironments, which can interfere with the translational prospects toward rational drug design. Even a single difference in the amino acid sequence can drastically decrease the efficacy of the effector molecule, not to mention the desired specificity toward other ABC proteins. From this point of view, the relevance of the *Mtb* transporter structure and transport cycle would be higher, further increasing the medical importance of the present study. Did the Authors try to investigate the structure of the *Mtb* heterodimer as well?

Response: We firstly aimed to study the structure and mechanism of *Mtb*Rv1273c/72c, however, it could not be successfully purified for cryo-EM studies. Indeed, the sample quality was not even good enough to do the basic sample screening. Thus, we focused on the study of *Ms*Rv1273c/72c.

We believe the sequence identity is high enough between *Msm* and *Mtb* transporters such that the conclusions arrived at for the *Msm* sample will largely hold true for *Mtb*, especially with regard to efforts that could be placed in inhibitor/drug design. Even though the microenvironments could be slightly different, we can generate models of the *Mtb* transporter in different states including the intermediate asymmetric states. Revisions have been added to the text: “Even though the localized environments could be different between the *Mtb* and *Msm* heterodimers, the overall folds and functional sites are expected to be conserved. Thus, our structures solved here represent accurate templates for anti-TB drug design. Potentially, the unique structural features in Rv1273c/72c such as the extending loop and ECD could be blocked by designed inhibitors. Alternatively, the conformational changes important for transport could be inhibited by targeting the D-WalkerB loop and X-signature loop of *Ms*Rv1272c. In addition, we can generate models of *Mtb* Rv1273c/72c in different intermediate states and use these as templates for inhibitor design. Thus, these new structures provide a solid framework for the development of novel anti-TB drugs.”

Minor comments and questions

5) The Authors might want to introduce the implication of AMPPNP and vanadate trapping in the introduction/results sections of the manuscript for the broader readership. It could help in understanding the conclusions the Authors made.

Response: This has been included in the results sections: “To capture *MsRv1273c/72c* in other states during transport, we tried several approaches to induce NBD closure. Since AMPPNP is a non-hydrolysable analog of ATP, it can stabilize the NBD dimer at the dimer interface and lock a transporter in the **Occ** or **OF** state. Therefore, we determined the cryo-EM structure in the AMPPNP-bound **IF^{asym-1}** state at 3.1 Å resolution (Supplementary Fig. 3 and 10b). However, its NBDs are only semi-closed. Next, vanadate was used in an attempt to trap the transporter in a conformation immediately after ATP hydrolysis. In such state, ADP together with vanadate should stabilize the NBD dimer. The cryo-EM structure was solved for this experiment and showed an ADP-bound **IF^{asym-5}** state at 3.4 Å resolution (Supplementary Fig. 8 and 10d). Its NBDs are also in a semi-closed state and no vanadate was observed”

6) The methods section for the grid preparation is not well-defined. Were Mg²⁺ present in all cases? It is reported added at the grid preparation of the E-Q-occluded structure, but it is certainly added for the If(turnover-2) state, as Mg is necessary for hydrolysis, and for IF(turnover-1) state, as Mg²⁺ is seen in the structure. Incubation time and temperature were not reported in all cases.

Response: Mg²⁺ was added into the sample in all the cases. Incubation time and temperature have also been added in the methods: “For the AMPPNP-bound **IF^{asym-1}** state, the protein was incubated with 5 mM AMPPNP and 4 mM MgCl₂ at 4 °C for 30 min before grid preparation. For the ATP-bound **Occ** state, the *MsRv1273c/72c*^{E553Q} mutant was incubated with 4 mM ATP and 4 mM MgCl₂ at 4 °C for 30 min before grid preparation. For the ATP|ADP-bound **IF^{asym-2}** state, the protein was incubated with 10 mM ATP and 4 mM MgCl₂ at 4 °C for 30 min before grid preparation. For the ADP-bound **IF^{asym-3}** state, the protein was incubated with 10 mM ATP and 4 mM MgCl₂ at 37 °C for 5 min before grid preparation. For the ADP-bound **IF^{asym-4}** state, 10 mM ADP and 4 mM MgCl₂ were incubated with the protein at 4 °C for 30 min before grid preparation. For the ADP-bound **IF^{asym-5}** state, 10 mM ATP, 4 mM MgCl₂ and 2 mM

Na_3VO_4 were incubated with the protein at 37 °C for 5 min before grid preparation. For the extending-loop truncated structure, the *MsRv1273c/72c* $^{\Delta 1-16}$ mutant was incubated with 5 mM AMPPNP and 4 mM MgCl_2 at 4 °C for 30 min before grid preparation.”

7) Standard deviation describes the distribution, and standard error of the mean (S.E.M.) describes how accurate the determined mean is. Therefore, it is more fortunate to display S.E.M., rather than std. deviation in such cases.

Response: We have now used S.E.M instead of S.D. for figures Fig. 1a, Fig. 2b, Supplementary Fig. 11d and 16a.

8) The Authors do not provide p values for the ATPase assay results. Are the determined values (especially for the E-Q vs the E-Q and D-N cases) significantly different?

Response: The p values have now been provided for the ATPase assay results in Fig 2b and Supplementary Fig 11d. ***, $p < 0.001$; ns, not significant.

9) Does Rv1273c and/or Rv1272c form homodimers?

Response: Previous reports predicted each of them can form a homodimer but there is a lack of direct evidence. These have been included in the Introduction section: “However, Rv1272c has also been hypothesized to function as a homodimer since it enhances the transport of radiolabeled long-chain fatty acids when expressed in *Escherichia coli* (*E. coli*)³². Rv1273c shows 28% sequence identity with Rv1272c³² and it is proposed as a homodimeric multidrug transporter with a wide substrate range as well as a probable contributor to biofilm formation³³. Further studies are needed to

investigate whether Rv1273c interacts with Rv1272c.”

Based on our results, we only proved the existence of the heterodimer.

10) In Figure 4, the only truly irreversible step is the hydrolysis of ATP, therefore, the Authors should indicate reversibility in some cases.

Response: We think besides ATP hydrolysis, the binding of ATP to induce NBD dimer at the composite NBSs is also irreversible, which should require ATP hydrolysis to break the NBD dimer. If this is not the case, **OF/Occ** state will be able to return to **IF^{apo}** state without consuming energy, which is unreasonable. The figure has been revised.

11) The authors might want to include a deeper explanation of the important residues in ATP hydrolysis at the NBD interface (i.e. Asp552 or the role of the switch histidine). An additional schematic figure could help the reader to understand the role of important residues, also subject of this study.

Response: the information has been added in the text: “Sequence alignment with other ATPases shows that Glu553 in the Walker B motif of *MsRv1272c* is the conserved catalytic residue that activates the attacking water molecule during ATP hydrolysis⁴¹. In *MsRv1273c* this residue is replaced by Asp497 (Fig. 2a). Another conserved residue is His584 in the Switch-loop of *MsRv1272c* which plays a central role in stabilizing the

transition state of the reaction⁴²” The location of E522 and H584 in *MsRv1272c*, D497 and Q528 in *MsRv1273c* has been highlighted in Fig. 2e and 2f.

Reviewer #1 (Remarks to the Author):

The new version of the manuscript provides in great detail the structural features of MsRv1273c/72c. It is clear that the authors have spent considerable time revising their manuscript and also acquiring new structural data. However, there are still many concerns that remain:

-It is still unclear why this particular transporter is a good target for inhibitor design against TB. It is also unclear why in general an ATP-binding transporter (and in particular, the ATP binding site) would be good target for inhibitors, when there are so many molecular machines that use ATP and would be affected by such an inhibitor.

Response: Since Rv1273c is capable of affecting the cell wall architecture and lipid composition, inhibitors targeting this transporter should be able to treat TB. This is similar to how ethambutol and isoniazid work by targeting cell wall biosynthesis. Secondly, Rv1273c promotes mycobacterial intracellular survival within macrophages. Inhibition of this transporter will then allow *Mtb* to be more easily killed in macrophages. Lastly, this transporter plays an important role in drug resistance and appears to be an isoniazid efflux pump (see below). The combination of an inhibitor that targets this transporter and other drugs with different targets should therefore increase anti-TB activity of those drugs, because they will not be pumped out by *Mtb*.

Though there are many ATP-binding transporters with different functions, they are also different in structures, conformations and mechanisms of action. The unique features in our transporter such as the substrate binding pocket, N-terminal extending loop, the periplasmic helical hairpin and the intermediate conformations can be targeted for specific drug design, not affecting other ATP-binding transporters. The ATP binding site in our structure is not the same as other ATP-binding transporters. Rv1273c/72c has a specific degenerate site with substitutions of Glu in the Walker B motif and His in the Switch loop by an Asp and Glu, respectively. These degenerate site residues are different compared to other ATP-binding transporters. Furthermore, the distorted consensus and degenerate site in the intermediate state could be trapped by specific inhibitors. In addition, the residues in the surrounding space of the ATP binding site are different from other ATP-binding transporters and could also be used for the development of specific inhibitors and drugs (see figure below).

In the context of specific drug discovery, the ATP binding site of protein kinases (this is a large family) is targeted by many approved drugs such as Imatinib (PDB code: 2HYY), Palbociclib (PDB

code: 7N7O) and Erlotinib (PDB code: 1M17). The success of these drugs provides further evidence that the ATP binding site of our transporter can be targeted for specific drug development.

The last paragraph in the Discussion section has been revised to include the above statements: "Rv1273c/72c is a good target for new drugs to treat TB. This is because its role is critical in cell wall biosynthesis, critical to pathogen survival in the host cell and the immune response, and in drug resistance. In particular, we have shown that *MsRv1273c/72c* is a drug efflux pump for isoniazid. *MsRv1273c/72c* shares 71% overall sequence identity with *Mtb Rv1273c/72c*. Even though the localized environments could be different between the *Mtb* and *Msm* heterodimers, the overall folds, intermediate conformations and functional sites are expected to be conserved. In addition, we can generate models of *Mtb Rv1273c/72c* in different intermediate states and use these as templates for inhibitor design. Thus, our structures solved here represent accurate templates for anti-TB drug design. Though there are many ATP-binding transporters with different functions, they are also different in structures, conformations and mechanisms of action. Potentially, the unique structural features in Rv1273c/72c such as the extending loop and ECD could be blocked by designed inhibitors, not affecting other ATP-binding transporters. Alternatively, the conformational changes important for transport could be inhibited by targeting the D-WalkerB loop and X-signature loop of Rv1272c, especially in the distorted degenerate and consensus sites. The residues in the surrounding space of the ATP binding site are different from other ATP-binding transporters and could also be used for the development of specific inhibitors and drugs. Thus, our new structures provide a solid framework for the development of novel anti-TB drugs."

-The authors discuss designing drugs specific to the degenerate site of the transporter from *Mycobacterium tuberculosis*. They also discuss the intricacies of the site and how small changes can affect it. However, for their research, they used analogs from *Mycobacterium smegmatis* and they show in Fig. 2a that the residues lining the degenerate NBS are different. This seems

contradictory to the authors' goals.

Response: The different residues in Walker B motif and Switch loop in Fig.2a include L494 and S527 in *MsRv1273* (they are V and T in *Mtb Rv1273*, respectively), however, they are far away from ATP (15.0 Å and 10.6 Å), not solvent accessible and not directly involved in the binding of nucleotide in the degenerate site (see figure below). In addition, both substituted residues are similar to those in *MsRv1273*, thus we can expect that they will not affect the structure of degenerate site and not affect ATP binding. Overall, *MsRv1273c/72c* and *Mtb Rv1273c/72c* share 71% sequence identity, another indicator that the overall structures are similar enough to allow effective rational drug design based only on the *Msm* structure. To clarify our earlier statement, we can use the conserved residues between *MsRv1273c/72c* and *Mtb Rv1273c/72c* for drug design, which are also directly involved in nucleotide binding. We suggest to target the distorted degenerate site in the **IF^{asym}** state, which also exists in *Mtb Rv1273c/72c* but is distinct compared to other proteins. Therefore, our structures of *MsRv1273c/72c* are suitable templates for drug design. The similar statements have been incorporated in the last paragraph in the Discussion section.

-If Rv1273c/Rv1272c is indeed linked to drug resistance, then why were the drugs (isoniazid, ethambutol, etc) not tested by the authors? In their comments to the reviewers, the authors say that they tested the ATPase activity of several potential substrates and saw no change in activity. The authors should note that this result alone is not sufficient to draw conclusions and they will need to design and perform proper transport assays.

Response: Based on the reviewer's suggestion, we have now performed a transport assay for isoniazid and ethambutol in proteoliposomes and detected, using mass spectrometry, the appearance of these drugs inside the liposomes. Our results showed that the wild-type protein is able to transport isoniazid into the liposomes while the D497N/E553Q double mutant cannot when ATP is provided

(Supplementary Fig. 1e). However, our transporter did not pump ethambutol in this assay (Supplementary Fig. 1f). To further support these results, we performed a growth assay for different *Msm* strains in the presence or absence of the drugs. We found that deletion of *MsRv1273c/72c* makes *Msm* more sensitive to isoniazid while no difference is observed for ethambutol (Supplementary Fig. 1g and 1h). This result confirmed that *MsRv1273c/72c* is indeed linked to the resistance of isoniazid but may not be linked to the resistance of ethambutol.

The following has been added in the Results section: " Since *Rv1273c* and *Rv1272c* are potentially involved as a resistance mechanism for drugs such as isoniazid and ethambutol, we tested their effects on ATP hydrolysis by the *MsRv1273c/72c* complex. However, neither drug affected ATPase activity (Supplementary Fig. 1c and 1d). Next, we established a transport assay using proteoliposomes to detect if the drugs had move inside the liposome when applied outside. Mass spectrometry showed that *MsRv1273c/72c* is able to transport isoniazid while ethambutol was not detected (Supplementary Fig. 1e and 1f). Consistent with this result, *Msm* growth in culture experiments showed *MsRv1273c/72c* is directly involved the resistance of isoniazid *in vitro* whilst its effect on ethambutol is negligible (Supplementary Fig. 1g and 1h). Thus, our data showed that *MsRv1273c/72c* forms a heterodimeric transporter, and is a drug efflux pump of isoniazid."

The following has been added in the Methods section: "**Liposome-based transport assay.** Wild type protein complex or the mutant was inserted into proteoliposomes as described above. Liposomes without inserted protein were used as a negative control. 50 μM or 100 μM or 500 μM of isoniazid or ethambutol, 5 mM ATP and 4 mM MgCl_2 were added into the reaction mix. The mixture was incubated at 37 $^\circ\text{C}$ for 30 min. Proteoliposomes were then collected by centrifugation at 230,000 g for 30 min. The supernatant was discarded, and the pellet was resuspended in Buffer A, followed by another centrifugation at 230,000 g for 30 min. This washing step was repeated three times. The pellet was resuspended in 200 μL Buffer A mixed with 200 μL ethyl acetate. The mixture was vortexed and then allowed to stand for 3 min. The upper layer was aspirated and evaporated to dryness under a stream of warm air. The residue was reconstituted in 200 μL methyl cyanide and filtered through a 0.22 μm filter for mass spectrometry. 100 ng mL^{-1} of either isoniazid (0.73 μM) or ethambutol (0.36 μM) was used as the standard for comparison with mass spectrometry. High-resolution mass spectra (HRMS) were recorded on an Agilent 6230 mass spectrometer using ESI. All experiments were performed in triplicate.";

"Growth-complementation assay. The *Msm* wild type strain (WT-*Msm*), *MsRv1273c/72c* knockout strain (Δ *MsRv1273c/72c) and the complemented strain containing pMV261-*MsRv1273c/72c* (Δ +*MsRv1273c/72c) were separately plated on Luria Agar (LA) supplemented with 100 $\mu\text{g mL}^{-1}$ carbenicillin. Single colonies were picked and cultured in LB liquid medium containing 100 $\mu\text{g mL}^{-1}$ carbenicillin until the optical density at 600 nm (OD_{600}) reached 0.6-0.8. The cultures were then diluted to an OD_{600} of 0.01 in 100 mL of LB medium supplemented with 0.1% (w/v) Tween80. For the experimental groups, 0.15 $\mu\text{g mL}^{-1}$ ethambutol or 0.05 $\mu\text{g mL}^{-1}$ isoniazid was added into the medium and cultured in a 37 $^{\circ}\text{C}$ shaker. Cell growth was determined by measurement of OD_{600} in triplicate."**

Supplementary Fig. 1c, 1d, 1e, 1f, 1g and 1h have been added to the manuscript.

-Regarding the lack of vanadate – the authors suggest that the amount of vanadate they added to the sample for cryo-EM was not sufficient and that is why they did not see vanadate in their EM density map. If the authors believe this to be true, why did they not repeat the experiment with the proper amount of vanadate?

Response: We have performed the cryo-EM experiment after incubation of 20 mM vanadate with the protein sample. The structure was solved at 2.7 Å resolution (Supplementary Fig. 12). As expected, with a high concentration of vanadate, the structure was successfully trapped in the occluded state rather than the **IF^{asym-5}** state. In this structure, ADP-Vi was bound at the consensus site while ATP was bound at the degenerate site (Supplementary Fig. 19e and 19f).

The following has been added in the Results section: "The cryo-EM structures were solved for this experiment and showed an ADP-bound **IF^{asym-5}** state at 3.4 Å resolution after treatment with 2 mM vanadate (Supplementary Fig. 8 and 13d) and an ATP|ADP+Vi-bound **Occ (Vi)** state at 2.7 Å resolution after treatment with 20 mM vanadate (Supplementary Fig. 12 and 13j).";

"Next, we increased the concentration of vanadate to 20 mM for cryo-EM and obtained the **Occ (Vi)** state structure as expected, which is similar to the above-mentioned **Occ** structure (Supplementary Fig. 12, 19e and 19f). This confirmed that the NBSs were not occupied by sufficient vanadate in the **IF^{asym-5}** structure."

The following has been added in the Methods section: "For the ATP|ADP+Vi-bound **Occ (Vi)** states, 10 mM ATP, 4 mM MgCl₂ and 20 mM Na₃VO₄ were incubated with the protein at 37 °C for 5 min before grid preparation."

Fig. 5 has been revised.

Supplementary Fig. 12 has been added.

Supplementary Fig. 19e and 19f have been added.

-The authors captured a state or states in which the degenerate site is 'clamped,' but there is no indication of nucleotide presence in the NBS. This raises suspicion and requires further experiments. Can the authors explain why they think such a conformation is physiologically relevant? It is highly doubtful that the NBDs can semi-dimerize without the nucleotide being present in the NBS. This suggests that the authors may have trapped an artificial state, and could be perhaps even an artefact of the detergent used.

&

-The proposed transport mechanism is still very speculative and not based on facts.

Response: We do not believe it is an artificial state. First, our purified protein is in the \mathbf{IF}^{apo} state after purification in detergent and it still has ATPase activity. In this state, the NBDs at the degenerate site are separated and allow nucleotide binding. This means the initial conformation is correct and conformational change is able to be induced by ATP (because NBD dimerization is required for ATP binding and hydrolysis). Thus, the purified protein in detergent is not an artefact. The $\mathbf{IF}^{asym-3/4/5}$ state is only induced by addition of nucleotide (ATP/ADP) into the purified sample. As we know, the two nucleotides are either the substrate or product of the transporter, and either induces physiological relevant conformations. An artificial state is unlikely to be induced by functional molecules. Thus, the $\mathbf{IF}^{asym-3/4/5}$ state represents a nucleotide-induced physiological conformation. On the other hand, it is possible that there are contacts between NBDs without nucleotide being present in the NBS. For example, both the C-terminal end of NBDs in our \mathbf{IF}^{apo} structure (Fig. 1b) and in the structure of TM287/288 (PDB code: 4Q4H; see Fig. 1B in the paper Hohl M, et al. Proc Natl Acad Sci U S A. 2014. PMID: 25030449) are involved in NBD partial dimerization. These cases support our observation that the $\mathbf{IF}^{asym-3/4/5}$ state has the NBDs semi-dimerized without the nucleotide being present in the NBS.

To further confirm this conclusion we eliminated detergent, by reconstituting our protein in either a nanodisc or peptidisc. Only the peptidisc sample (still active for ATPase activity, Fig. 1a) could be used for cryoEM. We were able to solve cryo-EM structures after addition of ATP (37 °C) or ADP (4 °C). Both captured the $\mathbf{IF}^{\text{asym-3/4/5}}$ state as expected (Supplementary Fig. 19c and 19d). Note that in the two new structures, the size and shape of peptidisc wrapping around the TM region is different from that in detergent (Supplementary Fig. 10 and 11). This confirmed that this state is not an artefact of the detergent used. Additionally, no densities for detergents were found in all the structures in the $\mathbf{IF}^{\text{asym-3/4/5}}$ state which might affect conformational changes or NBD dimerization.

The following has been added in the Results section: "However, more studies are required to determine how the nucleotide is released from this site. Here, the $\mathbf{IF}^{\text{asym-3/4/5}}$ state is only induced by addition of nucleotide (ATP/ADP) into the purified sample. As we know, the two nucleotides are either the substrate or product of the transporter, and either induces physiological relevant conformations. An artificial state is unlikely to be induced by functional molecules. Thus, the $\mathbf{IF}^{\text{asym-3/4/5}}$ state represents a nucleotide-induced physiological conformation. On the other hand, it is possible that there are contacts between NBDs without nucleotide being present in the NBS. For example, both the C-terminal end of NBDs in our \mathbf{IF}^{apo} structure and the *apo* structure of TM287/288 (PDB code: 4Q4H)²³ are involved in NBD partial dimerization. To rule out the effect of detergent on the induction of this state, we reconstituted the protein sample in a peptidisc and then solved structures with the same treatments as $\mathbf{IF}^{\text{asym-3}}$ and $\mathbf{IF}^{\text{asym-4}}$ states. These new structures (here we named $\mathbf{IF}^{\text{asym-3}}$ (**peptidisc**) (Supplementary Fig. 10 and 13f) and $\mathbf{IF}^{\text{asym-4}}$ (**peptidisc**) (Supplementary Fig. 11 and 13h)) adopt the same conformation as the $\mathbf{IF}^{\text{asym-3/4/5}}$ state (Supplementary Fig. 19c and 19d). This suggests that this conformation is induced by the nucleotide and it is not an artefact of the detergent used."

The following has been added in the Methods section: "To remove detergent and prepare protein reconstructed in peptidisc, lyophilized NSPr (Nter-FAEKFKAEVVKDYFAKFWDPAAEKLKEAVKDYFAKLWD-Cter) (Genscript, purity >98%) was solubilized in Buffer A at room temperature to a final concentration of 1 mg mL⁻¹ (Buffer B) and kept on ice for use⁵². After rinsing the Ni-NTA beads with Buffer A containing 0.02% (w/v) DDM and 30 mM imidazole, 10 mL Buffer B was added to the beads and incubated for 10 min on ice. The recombinant protein complex was eluted from the beads with Buffer B supplemented with 500 mM imidazole. The eluted sample was concentrated and applied to a Superose 6 column pre-

equilibrated with Buffer B supplemented with 5 mM DTT. Finally, the main peak fractions were pooled and concentrated to 2 mg mL⁻¹ for further studies or stored at -80 °C.";

"For the ADP-bound **IF**^{asym-3} state and **IF**^{asym-3} (**peptidisc**) state, the protein was incubated with 10 mM ATP and 4 mM MgCl₂ at 37 °C for 5 min before grid preparation. For the ADP-bound **IF**^{asym-4} state and **IF**^{asym-4} (**peptidisc**) state, 10 mM ADP and 4 mM MgCl₂ were incubated with the protein at 4 °C for 30 min before grid preparation."

Fig. 1a has been revised.

Supplementary Fig. 10 and 11 have been added.

Supplementary Fig. 19c and 19d have been added.

The $IF^{asym-3/4/5}$ state and other states are captured only with different treatments of nucleotides or temperatures. Thus, these states could be physiological related separate pictures during substrate transport. We have not been able to reveal exactly how the nucleotide is released from the degenerate site in the $IF^{asym-3/4/5}$ state, we only discussed a possible mechanism. We suggest it may be released as ADP after slow hydrolysis. However, to determine this will require further studies by capturing more intermediate states.

To support the idea that the IF^{asym-2} state is a state before $IF^{asym-3/4/5}$ and $IF^{asym-3/4/5}$ can be induced by IF^{asym-2} when ATP in the degenerate site is released, we performed a molecular dynamics simulation starting from the IF^{asym-2} state without ATP and calculated the changes of distance between the two NBDs. During 100 ns simulation, the distance similar to that observed in the $IF^{asym-3/4/5}$ state appeared after 25 ns (Supplementary Fig. 17d). This implies that the transition from IF^{asym-2} to $IF^{asym-3/4/5}$ can occur.

The following has been added to the text: "To support this proposal, we performed a molecular dynamics (MD) simulation starting from the IF^{asym-2} structure by deleting ATP in the degenerate site, then we inspected the conformations and analyzed the changes of the distance between the two NBDs close to the consensus site during 100 ns simulation (Supplementary Fig. 17d). We found that the NBDs keep interactions at the degenerate site and the $IF^{asym-3/4/5}$ like state could be induced from the IF^{asym-2} state if ATP in the degenerate site is released."

The following has been added to the Methods section: "**Molecular Dynamics (MD) Simulation.** To simulate the progress from IF^{asym-2} to $IF^{asym-3/4/5}$, the starting model was constructed using the IF^{asym-2} structure with ATP removed from the degenerate site. The system was built with the CHARMM-GUI webserver⁶⁰ using the CHARMM36m⁶¹ force field and TIP3P water model.

Protein was embedded into the lipid bilayer consisting of 50% 1-palmitoyl-2-oleoyl-sn-glycero-3-phosphocholine (POPC) and 50% 1-palmitoyl-2-oleoyl-sn-glycero-3-phosphoethanolamine (POPE). The upper layer has 136 POPC molecules and 136 POPE molecules while the lower layer has 131 POPC molecules and 131 POPE molecules. The structure described above was solvated in a cubic water box containing 0.5 M NaCl. Energy minimization, NVT and NPT equilibration were performed in six steps, 100 ns production was performed by GROMACS 2024.1⁶² at 310.15 K with default parameters provided by the CHARMM-GUI webserver. The distance was calculated between the mass center of all the C α atoms in partial NBD of *MsRv1273c* (residues G418 to A519) and the mass center of all the C α atoms in partial NBD of *MsRv1272c* (residues S380 to D473 and residues P546 to G628). Visualization was performed using R-4.3.2 and PyMOL."

Supplementary Fig. 17d has been added.

To tone down our expression for the transport mechanism, we revised the text: "(7) ATP is then released from the distorted degenerate site by an unknown mechanism, possibly in the form of ADP after slow hydrolysis.";

"To complete the full suite of structures to explain the mechanism requires additional intermediate states to be determined. Further evidence is needed to explain how step (7) occurs to induce the conversion from the $\text{IF}^{\text{asym-2}}$ state to the $\text{IF}^{\text{asym-3/4/5}}$ state (Fig. 5)."

Fig. 5 has been modified by including a grey arrow and a question mark for the transition from the $\text{IF}^{\text{asym-2}}$ state to the $\text{IF}^{\text{asym-3/4/5}}$ state.

Reviewer #2 (Remarks to the Author):

The Authors answered all of my questions and concerns.

Reviewer #1 (Remarks to the Author)

Thank you to the authors for the revised version and additional experiments. Unfortunately, there are still many concerns that remain. The manuscript can benefit from a rewrite to address the following:

- Proper rephrasing of the citations in the introduction

Response The text in the introduction section has been significantly rewritten. We have tried to more faithfully respect the commentary in the selected citations.

- Reorganization of the Results section. Results and discussion are mixed, although there is also a separate Discussion section. Authors jump from one topic to another and often it is difficult to follow their logic and hence, evaluate their science. Section headings do not always correspond to what is discussed

Response The Results have been reorganized such that only one structural state is described in each subsection. Discussion in the results has been moved to the discussion section or removed from text. Each subsection has been revised such that the headings are appropriate. For example, the text on the IF^{asym-3} state being a physiological state has been moved into the discussion. Discussion about the predicted OF model in the subsection of the IF^{apo} structure has been removed. In the section about the Occ state, we have removed mention of the IF^{asym-5} structure. In the section about IF^{asym-1} state, we deleted the discussion about the extending loop by using the truncated Δ 1-16 structure. The title of subsection “Structure in the ADP-bound IF^{asym-3/4/5} state” has been changed to “Structure in the ADP-bound IF^{asym-3} state”, thus we only analyze one structure in the IF^{asym-3} state.

All unrelated data have been removed, including the IF^{asym-5} structure, the Δ 1-16 truncated structure, and corresponding results of ATPase activity assay for the N-terminal truncation and inhibition of vanadate.

- There is a lot of repetition throughout the text

Response We have tried to remove as much repetition as possible (e.g., mention about the AMPPNP-bound IF^{asym-1} structure and ADP-bound IF^{asym-5} structure in different places).

- Text needs to be proofread for sentence structure and grammar mistakes

Response The text has been polished by all authors, including several who are fluent in English.

- More details are needed in the Methods section. For example, for the ATPase assay, it is unclear what system it was performed in. It is also not sufficient to say “as described previously” especially when the reference itself says the same thing and links to yet another reference. Methods should include enough detail so that they can be repeated by someone else.

Response The methods section has been revised to include all details. The ATPase assay has been revised: “The ATPase activity assay was performed as described previously⁴¹. The released P_i from ATP hydrolysis reacts with malachite green reagent to form a stable dark green complex, whose presence is detected by measuring absorbance at 620 nm. The intensity of the color is directly proportional to the amount of P_i generated, and thus, to the ATPase activity in the sample. For the assay, 4.68 μg of purified protein complex or the mutants was incubated in a 20 μL reaction volume containing 50 mM Tris (pH 8.0), 500 mM NaCl, 1 mM ATP and 4 mM MgCl₂ for 10 min at 37 °C. The reaction was stopped by mixing 100 μL aliquots with activated malachite green ammonium molybdate for 2 min. Samples were subsequently incubated at room temperature for 30 min with 24% (w/v) sodium citrate after which the absorbance at 620 nm was measured using a SpectraMax iD3 multifunction reader (Molecular Devices). Samples without ATP were used as the negative controls and subtracted as background. ATPase activity is represented as the amount of P_i produced by 1 mg of protein per minute. All experiments were performed in triplicate.”

In this new version, the authors present ATPase activity assay and transport assays. They did not see any change in ATPase activity in the presence of isoniazid, yet they conclude that the transporter acts as a drug efflux pump of isoniazid. The transport assay is also very confusing. It is unclear from the information presented, what the control was in this experiment.

Response The conclusion that the transporter acts as a drug efflux pump of isoniazid is based on the results from the transport assay, not the ATPase activity. Since isoniazid is not a physiological substrate of the transporter, it is possible that the drug will not have strong binding interactions with the transporter thus the ATPase activity is not increased.

As to the control, the setup of transport assay is stated in the Methods section. “Liposomes without inserted proteins were used as the negative control.”, “0.1 μg mL⁻¹ of either isoniazid (0.73 μM) or ethambutol (0.36 μM) were separately tested by mass spectrometry, which was used as the standard reference.” The D497N/E553Q double mutant was also used for comparison. Our results showed that only the wildtype protein can transport isoniazid, while the mutant cannot transport isoniazid. Therefore, the results from the transport assay provide convincing evidence that *MsRv1273c/72c* is a drug efflux pump of isoniazid. Furthermore, we showed that isoniazid is transported by *MsRv1273c/72c* in a dose-dependent manner (see response to reviewer #2).

In the Results section, we have added the following description to make the results of transport assay clearer: “Next, we established a transport assay to determine whether isoniazid or ethambutol could be imported into proteoliposomes containing *MsRv1273c/72c* (Fig. 1d). Mass spectrometry detected isoniazid inside the liposomes when the wildtype transporter was inserted into the liposomes, while there was no detectable isoniazid inside the liposomes when the ATPase inactive mutant was used (Fig. 1e). When ethambutol was tested, it was not detected inside the liposomes in any of the experiments (Supplementary Fig. S1g). These results confirmed that *MsRv1273c/72c* is able to transport isoniazid but not ethambutol. Further analysis by Liquid Chromatography-Tandem Mass Spectrometry (LC-MS/MS) showed that isoniazid is transported by *MsRv1273c/72c* in a dose-dependent manner (Fig. 1d and 1f).”

The figure legends of Fig. 1e and Supplementary Fig. S1g have been revised to clearly describe the results: “e Mass spectrometry was used to determine the contents inside the liposomes with wildtype *MsRv1273c/72c* (WT), or with the E553Q/D497N double mutant (EQ&DN) inserted, and liposomes without any protein added. Pure isoniazid (INH) was also measured as a standard. The red circle indicates the expected mass/charge of the drug.”

“g Mass spectrometry was used to determine the contents inside the liposomes with wildtype *MsRv1273c/72c* (WT), or with the E553Q/D497N double mutant (EQ&DN) inserted, and liposomes without any protein added. Pure ethambutol (EMB) was also measured as a standard. The red circle indicates the expected mass/charge of the drug.”

Fig. 1d has been added to help the reader to understand how the transport assay was performed:

Some suggested edits and concerns are listed below:

Ln 29 “for their transport” should be “of their transport mechanism”

Response The sentence has been revised: “However, the structural basis of their transport mechanism remains to be explained”.

Ln 31 “isoniazid efflux pump” -> implies that this was already known when it is something authors tested here

Response The sentence has been revised: “Here, we have determined mycobacterial *MsRv1273c/72c* to be an isoniazid efflux pump and determined several structures by cryo-electron microscopy showing specific asymmetrical features including an N-terminal extending loop and a periplasmic helical hairpin only found in *MsRv1272c*.”

Ln 37 “a heterodimeric ABC transporter” -> please specify that you are talking for one particular ABC transporter and not all ABC transporters with a degenerate NBS

Response The sentence has been revised: “These data provide new insights into the mechanism of this heterodimeric ABC transporter containing a degenerate NBS”.

Ln 38 “asymmetry” -> which asymmetry are you referring to here? Between the MsRv1273c/72c or the active vs degenerate NBS?

Response The asymmetry is between MsRv1273c and MsRv1272c and also between consensus NBS and degenerate NBS. We are not able to explain the “asymmetry” in detail due to the word limit for the abstract, thus the expression has been deleted: “These data provide new insights into the mechanism of this heterodimeric ABC transporter containing a degenerate NBS”. The “asymmetry” is discussed in detail in the first paragraph of the Discussion section.

Ln 142 – 146: Could you comment on how you would interpret the fact that you see no effect of the drugs on the rate of ATP hydrolysis?

Response Since isoniazid is not the physiological substrate of the transporter, it may only weakly bind to the central cavity and not be able to affect the conformation. This may be the reason that ATP hydrolysis is not affected. The sentence has been revised: “In this experiment, neither drug affected ATPase activity (Supplementary Fig. 1c and 1d), this may be due to their weak binding, which is commonly observed for a non-physiological substrate.”

Ln 147: “if the drugs had move inside the liposome when applied outside” -> please rephrase. Perhaps something along the lines of: “...to determine whether isoniazid and ethambutol could be imported by MsRv1273c/72c into the proteoliposomes.”

Response The sentence has been revised: “Next, we established a transport assay to determine whether isoniazid or ethambutol could be imported into proteoliposomes containing MsRv1273c/72c (Fig. 1d).”

Ln 151: “resistance of” -> should be “resistance to”

Response This has been corrected.

Ln 151: “whilst its effect on ethambutol” -> unclear what you are referring to / poor scientific language

Response The sentence has been revised with new expression: “However, no growth difference was observed in all these strains in the presence of ethambutol (Supplementary Fig. S1h), which suggests that MsRv1273c/72c is not sensitive to ethambutol.”

Ln 152 – 153: Based on the results shown, it is not clear why the authors think that the transporter is a drug efflux pump for isoniazid. The ATPase activity seems to be unaffected by isoniazid. Furthermore, the cell-based transport assay shows diminished cell growth in the presence of isoniazid, but there is no cell death. The authors do not discuss this result.

Response Our transport assay has shown that MsRv1273c/72c is a drug efflux pump for isoniazid

but not for ethambutol. As a control, both the liposomes alone and liposomes with the D497N&E553Q double mutant (inactive for ATP hydrolysis) cannot pump isoniazid. This is the direct evidence to conclude that our transporter is a drug efflux pump for isoniazid. Furthermore, we showed that isoniazid is transported by *MsRv1273c/72c* in a dose-dependent manner (see response to reviewer #2). Since isoniazid is not the physiological substrate of the transporter, it may only weakly bind to the central cavity and not be able to affect the conformation. This may be the reason why the rate of ATP hydrolysis is not changed. We also performed cell growth assays, which provides further evidence to show *MsRv1273c/72c* is a drug efflux pump for isoniazid.

The cell death is not observed in the growth assay because we used a low concentration of isoniazid ($0.05 \mu\text{g mL}^{-1}$) which only affects the growth rate. It is known that only a high concentration of isoniazid ($>1 \mu\text{g mL}^{-1}$) will cause cell death for the wild type *Msm* strain, thus, we can expect that increased cell death will be observed for the knockout strain at a high concentration of isoniazid since it is more sensitive for isoniazid. After all, our data have already shown the difference in growth, which has proved that *MsRv1273c/72c* contributes to the resistance of isoniazid. This then indirectly supports the finding that our transporter is a drug efflux pump.

The paragraph has been revised as follows: “Since *Rv1273c* and *Rv1272c* could potentially play a part in the resistance mechanism for drugs such as isoniazid and ethambutol, we first tested their effects on ATP hydrolysis using the *MsRv1273c/72c* complex. In this experiment, neither drug affected ATPase activity (Supplementary Fig. 1c and 1d), this may be due to their weak binding, which is commonly observed for a non-physiological substrate. Next, we established a transport assay to determine whether isoniazid or ethambutol could be imported into proteoliposomes containing *MsRv1273c/72c* (Fig. 1d). Mass spectrometry detected isoniazid inside the liposomes when the wildtype transporter was inserted into the liposomes, while there was no detectable isoniazid inside the liposomes when the ATPase inactive mutant was used (Fig. 1e). When ethambutol was tested, it was not detected inside the liposomes in any of the experiments (Supplementary Fig. S1g). These results confirmed that *MsRv1273c/72c* is able to transport isoniazid but not ethambutol. Further analysis by Liquid Chromatography-Tandem Mass Spectrometry (LC-MS/MS) showed that isoniazid is transported by *MsRv1273c/72c* in a dose-dependent manner (Fig. 1d and 1f). In agreement with this, *Msm* culture growth experiments showed that knocking out of *MsRv1273c/72c* resulted in an increase in growth inhibition when isoniazid was added, while complementation of *MsRv1273c/72c* in the knockout strain eliminated such a growth difference in the presence of isoniazid (Fig. 1g). These results confirm that *MsRv1273c/72c* is involved in resistance to isoniazid. However, no growth difference was observed in all these strains in the presence of ethambutol (Supplementary Fig. S1h), which suggests that *MsRv1273c/72c* is not sensitive to ethambutol. Thus, *MsRv1273c/72c* is a drug efflux pump for isoniazid, but not ethambutol. Therefore, its *Mtb* counterpart is a likely factor contributing to isoniazid resistance in *Mtb*.”

Ln 178: “conserved Trp246” -> conserved with respect to what? Which species is it conserved in?

Response: The sentence has been revised with clearer expression: “...and then interacts with the sidechain of Trp246 in TM4, which is conserved among the mycobacterial *Rv1272c* homologs

(Supplementary Fig. 12c)".

Supplementary Fig. 12c has been added:

Ln 189: "which is extremely long" -> not very quantitative or scientific; unnecessary in the text

Response: This sentence has been revised as the reviewer suggested: "The extra-cellular domain (ECD) is formed by the extracellular loop 1 (ECL1, residue 62-109) between TM1 and TM2."

Ln 193-205: Out of place - unclear how this part fits into a section titled "Structure in the IFapo state"

Response: This part has been deleted.

Ln 208: "To capture MsRv1273c/72c in other states during transport..." -> needs to be rewritten to reflect what was done. These are not turnover conditions, so no states are being captured during transport. Instead, the authors try to capture a pre- and post-hydrolysis state as they discuss further in the text.

Response: Because we have reorganized the Results section, this sentence has been deleted.

Ln211: It is unclear why the authors chose to use AMPPNP for their studies. What are the properties of this molecule that the authors were interested in? In particular, what additional information to that provided by EtoQ and vanadate were the authors hoping to obtain?

Response: Since AMPPNP is not able to induce the Occ state, we did not mention it in the "Structure in the ATP-bound Occ state" section. We only mentioned AMPPNP in the next "Structure in the AMPPNP-bound IF^{asym-1} state" section by using it as a non-hydrolysable analog of ATP to capture pre-hydrolytic state occurring before the Occ state. The EtoQ mutant or ATP-vanadate is often used to capture the Occ or OF state. The sentences have been revised as follows: "To capture a pre-hydrolytic state occurring before the Occ state, we treated the cryo-EM sample with AMPPNP, since it is a non-hydrolysable analog of ATP. The cryo-EM structure of MsRv1273c/72c in the AMPPNP-bound IF^{asym-1} state was then obtained (Fig. 4a, Supplementary Fig. 4 and 11b)."

Ln 211 – 212: Please describe what you visually see for your AMPPNP state before you go on to discuss another structure.

Response: We have now reorganized the Results section. The AMPPNP bound state is now not mentioned in this section, we only discuss the Occ state. The AMPPNP bound state is described in the next section on the AMPPNP-bound IF^{asym-1} structure.

Ln 220 – 240: The effect of an EtoQ mutation in the NBS is well established in the field. Perhaps the authors should discuss their experiment design and results in context to that. It is also unclear from the text why the authors pursued the D497N mutant.

Response: We have already mentioned in the text that the EtoQ mutation is required to trap the structure in the pre-hydrolytic state. However, to make it clearer, we have re-organized the text: “An approach to obtain the **Occ** state of ABC transporter is to inactivate the ATPase activity by making a glutamate to glutamine (EtoQ) mutation in the Walker B motif of NBD. This allows the trapping of the transporter in a pre-hydrolytic state⁴³. Based on this, we purified the E553Q mutant which had a significantly diminished ATPase activity and then added ATP before data collection. We then solved the cryo-EM structure of E553Q mutant in the presence of ATP at 3.1 Å resolution, which showed an ATP-bound **Occ** state (Fig. 3a, Supplementary Fig. 3 and 11h).”

We have stated in the text that Asp497 aligned with the conserved catalytic Glu in Walker B motif of NBDs from different ABC transporters. Since the catalytic residue is Asp497 in *MsRv1273c* instead of Glu, we wanted to test if this residue affects ATPase activity by making the D497N mutant. We have revised the text to explain more clearly: “However, the corresponding residues in *MsRv1273c* are replaced by Asp497 and Gln528, respectively (Fig. 1b). Thus, *MsRv1273c* should have a degenerate site impaired for ATPase activity while *MsRv1272c* has a functional consensus site with the classic enzymatic motifs. Based on this analysis, we performed a mutagenesis study and used an ATPase activity assay to verify the roles of the expected catalytic residues (e.g., E553 acts as a catalytic base?).”

Ln 242 - 243: “We then solved the cryo-EM structure in the ATP-bound Occ state...”: Please note that the Occ state is what you observed based on your solved structure. This could be phrased better to indicate that your EtoQ mutant in the presence of ATP yielded a 3.1 Å structure that in the Occ state and then you can describe what this is characterized by.

Response: The sentence has been revised as the reviewer suggested: “We then solved the cryo-EM structure of E553Q mutant in the presence of ATP at 3.1 Å resolution, which showed an ATP-bound **Occ** state.”

Ln 248 – 249: The authors might want to include references to support their statement.

Response: The following reference has been added:

Thomas, C. & Tampe, R. Structural and Mechanistic Principles of ABC Transporters. *Annu Rev Biochem* **89**, 605-636, doi:10.1146/annurev-biochem-011520-105201 (2020).

Ln 258: Here the text jumps back to AMPPNP, which was originally mentioned in Ln 211 – 212.

The authors may want to reorganize the flow of information so that it is easy to follow and not repetitive.

Response The flow of the results has been reorganized. The description of the AMPPNP bound structure was deleted from Ln 211-212, and is now only mentioned in this section.

Ln 262 – 279: This is now a completely different experiment and discussion within the AMPPNP section; the relevance is unclear.

Response These sentences has been deleted.

Ln 281 – 285: Based on what is shown in the cited Figure, it is unclear what this discussion is about.

Response The sentences have been revised: “However, in this state, the overall conformation is asymmetric, especially at the NBDs. Though both NBSs bind AMPPNP, the NBDs are partially dimerized at the degenerate site whereas there is a gap between NBDs at the consensus site (Fig. 4b). Thus, the ability of AMPPNP to mediate NBD dimerization at consensus site is reduced compared with that at degenerate site.”

Ln 285: “...the ability of AMPPNP mediating NBD dimerization at consensus site” -> should read: the ability of AMPPNP to mediate NBD dimerization at the consensus site

Response The sentence has been revised: “Thus, the ability of AMPPNP to mediate NBD dimerization at consensus site is reduced compared with that at degenerate site.”

Ln 301 “is not ideally bound” -> poor word choice, unclear what do you mean by “ideally”

Response The sentence has been revised: “These observations indicate that AMPPNP binds weakly at the degenerate site in the $\mathbf{IF}^{\text{asym-1}}$ state.”

Ln 304: Based on the section heading it is unclear which sample the authors are discussing.

Response We explain that the $\mathbf{IF}^{\text{asym-3}}$, $\mathbf{IF}^{\text{asym-4}}$, $\mathbf{IF}^{\text{asym-5}}$ structures are nearly identical and can be recognized as one state, though they were obtained with different treatments. To make it clear, we deleted the ATP+Vi treated $\mathbf{IF}^{\text{asym-5}}$ structure in the manuscript as suggested by the reviewer (see below). Meanwhile, we have re-defined the ATP (37 °C) treated $\mathbf{IF}^{\text{asym-3}}$ structure and ADP (4 °C) treated $\mathbf{IF}^{\text{asym-4}}$ structure to be $\mathbf{IF}^{\text{asym-3}}$ structures. In this section, we analyzed the ADP-bound $\mathbf{IF}^{\text{asym-3}}$ state using the ATP (37 °C) treated structure and this has been stated in the text. The section heading has been revised to now read: “**Structure in the ADP-bound $\mathbf{IF}^{\text{asym-3}}$ state**”.

Ln 310 – 311: The information here seems completely out of place.

Response This paragraph has been reorganized and the sentence has been moved to a more suitable place in the paragraph: “...In this structure, ADP binds at the consensus site (Fig. 4d) as a

hydrolyzed product after ATP hydrolysis. Interestingly, there is no nucleotide at the degenerate site even though we added ATP or ADP in the sample. The NBDs are also partially dimerized (Fig. 4e), but the conformation is different from the IF^{asym-1} state.....”.

Ln 320 – 328: The information here was already discussed in Ln 208 – 219.

Response: These sentences have been deleted from the text.

Ln 325 – 328: It is irrelevant to discuss the sample that did not work. Please only keep the sample that had enough vanadate, which produced results consistent with the literature (i.e. bound vanadate where you expect it).

Response: The IF^{asym-5} structure has been deleted from the text.

Ln 361 – 362: The sentence here needs to be rewritten in a grammatically correct way to convey the intended message.

Response: The sentence has been revised: “The consensus site is bound with an ADP, which originates from the rapid hydrolysis of ATP, while the degenerate site is bound with an ATP, which has not been hydrolyzed.”

Ln 362 – 364: The logic is unclear.

Response: This sentence has been deleted.

Ln 697: Here and throughout the rest of this section, “UN” should be “NU”

Response: These have been corrected.

For the cryo-EM methods: (1) It will be good to mention exclusion criteria used after heterogeneous refinements. Also, if you performed any of the following, please include that as well – Local Refinement, Global CTF Refinement, Local CTF Refinement, Per-particle motion correction, use of any masks and at which stage. (2) Please note that the Supplementary Figures (workflows) for this section poorly reflect the actual processing done.

Response: (1) We did not use Local Refinement, Global CTF Refinement, Local CTF Refinement, Per-particle motion correction or masks during cryo-EM data processing. The exclusion criteria used after heterogeneous refinements have been added in the methods: “They were used for Ab-Initio reconstruction to generate 3D models as references to perform heterogeneous refinement. The particles belonging to the correct initial map were subjected to several rounds of heterogeneous refinement. After that, 139,918 particles belonging to the map with best resolution were then refined using non-uniform (NU) refinement to generate the final cryo-EM map with an estimated average resolution of 3.1 Å according to the gold-standard Fourier shell correlation cutoff of 0.143⁵⁶. No mask was used for all the map refinement processes.”

(2) The workflows in Supplementary Figures are the actual processing flows we have performed, though each dataset produces different results in the intermediate processing steps. We have revised the presentations of workflows in Supplementary Fig. 2-10 to keep consistent with the descriptions of the data processing in the methods. However, the final results have not changed.

Reviewer #2 (Remarks to the Author):

The authors added new functional experiments to further support their results. While the manuscript has greatly improved, I have a few comments / questions.

The Authors did not provide any detail on the quality control of reconstitution into vesicles. Did they assess the integrity of vesicles and the efficacy of reconstitution? Upon detergent removal, membrane proteins might aggregate instead of being reconstituted into the liposomes. Can the Authors provide any data (negative stain EM / cryoelectron microscopy) / explanation on this issue?

Response: To perform quality control of reconstitution into vesicles, we first checked the sample by negative stain EM. The result showed that the proteoliposomes are integral and no protein aggregation is formed. Protein incorporation was evaluated by SDS-PAGE analysis. The results showed that about 97% of the proteins added appear in the proteoliposomes. This ratio was used to adjust protein amount in the ATPase assay and transport assay for the proteoliposomes.

The following has been added to the Methods section: “The integrity of vesicles was checked by negative stain EM to guarantee that all the proteoliposomes were integral and no protein aggregation was formed (Supplementary Fig. S1f). The efficacy of reconstitution was evaluated by SDS-PAGE showing about 97% of the added protein became incorporated into the proteoliposomes (Supplementary Fig. S1e). This ratio was used to adjust the amount of protein in the ATPase and transport assays.”

Supplementary Fig. S1e and S1f have been added:

Also, the protein concentration of 20 mg/ml seems to be a bit high, especially for reconstituted proteins in liposomes. If it is not a typo (i.e. 2 mg/ml), what is the rationale behind it?

Response: We are sorry for the mistake. 20 mg/ml is the concentration of lipids (the corresponding concentration of protein is about 0.2 mg/mL). The text has been corrected as follows: “Then, they were suspended in Buffer A and the lipid concentration was adjusted to 20 mg mL⁻¹ for further studies or stored at -80 °C.”

Reconstitution into proteoliposomes usually results in both the outside-out and the inside-out configurations. While the Authors adjusted the concentration of the liposome-reconstituted transporter, only the activity of the outside-out configuration can be measured (since ATP cannot enter the liposomes), which, assuming non-directed reconstitution (~50-50% distribution) would result in a significant increase in the ATPase rate. Do the Authors have any insight on this issue?

Response: We have updated this result by undertaking a new measurement by considering the 50-50% distribution of two configurations. The ATPase rate was increased as expected. The sentence in the text has been revised as follows: “The results showed that it has basal ATPase activity of 113.3 ± 13.9 nM P_i min⁻¹ mg⁻¹ (mean ± S.E.M., n = 3), 181.2 ± 2.3 nM P_i min⁻¹ mg⁻¹ and 147.9 ± 2.4 P_i min⁻¹ mg⁻¹ in detergent, proteoliposomes and peptidiscs, respectively (Fig. 1a).”

This statement has been added in the Methods section: “The concentration of the liposome-reconstituted transporter was further adjusted by considering the 50-50% distribution of two

configurations (outside-out and inside-out) in the proteoliposomes⁶¹. Only the activity of NBDs in the inside-out configuration can be measured.”

Fig. S1a has been updated:

The proteoliposome-based transport-assay suggests that isoniazid can be transported by the heterodimer transporter, while ethambutol cannot be detected inside the vesicles. The methods section describes 3 tested concentrations of the drugs, while they show only one condition for each and forgot to specify the condition itself. Did the Authors observe concentration dependency? I am wondering, why the authors did use ethyl acetate which is known to be a less effective extracting agent for liposomes. Did the Authors/others test the distribution of proteoliposomes in the EtOAc/water system? Furthermore, the authors might want to consider using a system with LC-MS readout to quantify the amount of transported drugs.

Response: The current assay is a qualitative analysis and the result is solid. By comparing with the proteoliposomes for inactive mutants and control of empty liposomes, we found that isoniazid can be transported by the heterodimer transporter, while ethambutol cannot be detected inside the vesicles. To detect if there is concentration dependency, we modified our sample processing procedure by adding methanol instead of ethyl acetate. This process can disrupt the proteoliposomes without discarding any components. Thus, the amount of isoniazid released can be correctly measured by the LC-MS/MS method. Our results showed that there is dose-dependency between the transport rate of isoniazid and its concentration supplied outside the proteoliposomes.

The following sentence has been added to the Results section: “Further analysis by Liquid Chromatography-Tandem Mass Spectrometry (LC-MS/MS) showed that isoniazid is transported by MsRv1273c/72c in a dose-dependent manner (Fig. 1d and 1f).”

The following has been added in the Methods section:

“To quantify the amount of isoniazid transported by proteoliposomes, isoniazid standards were dissolved in methanol at concentrations of 0.02, 0.2, 1, 10, and 50 $\mu\text{g mL}^{-1}$. The calibration curve was generated using Liquid Chromatography-Tandem Mass Spectrometry (LC-MS/MS). Next, 20 mg mL^{-1} lipid concentration of proteoliposomes, 5 or 50 or 500 $\mu\text{g mL}^{-1}$ isoniazid, 5 mM ATP and 4 mM MgCl_2 were mixed to prepare a reaction system with a total volume of 100 μL . The mixture

was incubated at 37 °C for 30 min. After that, the proteoliposomes were washed three times and the pellet was suspended in 100 µL of buffer A. Next, 900 µL of methanol was added to disrupt the proteoliposomes. The supernatant was collected by centrifugation at 12,000 rpm for 10 min before being passed through a 0.22 µm filter to prepare the samples for LC-MS/MS analysis. The transport rate is the amount of isoniazid transported by 1 mg of protein per minute. All experiments were performed in triplicate.”

“Liquid Chromatography-Tandem Mass Spectrometry

LC-MS/MS was performed using a Shimadzu 30A HPLC system equipped with an autosampler, binary pump, and column oven, coupled with an AB 4600 mass spectrometer (AB SCIEX, Framingham, MA, USA) equipped with a Turbo Ion Spray source. The mobile phases consisted of 0.1% formic acid in distilled water (A) and acetonitrile containing 0.1% formic acid (B) run at a flow rate of 0.4 mL min⁻¹. Chromatography was performed using an ACE C4 column (5 µm, 100 × 2.1 mm) with a gradient elution. The column temperature was set at 75 °C. The gradient elution condition was set as follows: t = 0 min, A = 100%, B = 0%; t = 2.5 min, A = 100%, B = 0%; t = 5 min, A = 5%, B = 95%; t = 6.5 min, A = 5%, B = 95%; t = 6.6 min, A = 100%, B = 0%; t = 8.0 min, A = 100%, B = 0%. The Electrospray Ionization was set in positive ion mode: the voltage value was 5500 V, gas 1 and gas 2 were set at 50 psi, and the curtain gas at 35 psi. The scan range was set from 80 to 550 Da. The collision energy was set to 30. The mass transition of isoniazid was 138 (m/z) to 121 (m/z)⁶². In the first-stage, mass spectrometry (MS1) was used to obtain precursor ions; and in the second stage, the precursor ion with m/z 138 was further analyzed by mass spectrometry (MS2). Data were collected to determine the quantity of the substance with m/z 121 by MS2.”

Fig. 1d and 1f have been added to help explain all of the steps:

While the Authors might want to tone further down the translational perspectives of the study due to the lack of physiological/functional data and the subtle differences between the two orthologs, I still think that the manuscript is suitable for publication, since the structural data is convincing, and the study is scientifically sound.

Response: We very much appreciate all the comments by the reviewer.